# Single-cell multiomics analysis reveals dynamic clonal evolution and targetable phenotypes in acute myeloid leukemia with complex karyotype

Chromosomal instability is a major driver of intratumoral heterogeneity (ITH), promoting tumor progression. In the present study, we combined structural variant discovery and nucleosome occupancy profiling with transcriptomic and immunophenotypic changes in single cells to study ITH in complex karyotype acute myeloid leukemia (CK-AML). We observed complex structural variant landscapes within individual cells of patients with CK-AML characterized by linear and circular breakage–fusion–bridge cycles and chromothripsis. We identified three clonal evolution patterns in diagnosis or salvage CK-AML (monoclonal, linear and branched polyclonal), with 75% harboring multiple subclones that frequently displayed ongoing karyotype remodeling. Using patient-derived xenografts, we demonstrated varied clonal evolution of leukemic stem cells (LSCs) and further dissected subclone-specific drug–response profiles to identify LSC-targeting therapies, including BCL-xL inhibition. In paired longitudinal patient samples, we further revealed genetic evolution and cell-type plasticity as mechanisms of disease progression. By dissecting dynamic genomic, phenotypic and functional complexity of CK-AML, our findings offer clinically relevant avenues for characterizing and targeting disease-driving LSCs.

Acute myeloid leukemia with complex karyotype (CK-AML) is typically characterized by three or more chromosomal aberrations and comprises 10–12% of patients with AML. The disease is associated with complex chromosomal rearrangements[1], ITH, therapy resistance and poor overall survival[2–4]. The molecular and cellular mechanisms underlying poor response to standard induction chemotherapy are poorly understood, although frequent *TP53* loss and extensive ITH as a result of genomic instability are believed to contribute to therapeutic failure[2,5]. Despite major clinical need, CK-AML has remained understudied at the genomic, molecular and cellular levels, largely because of technological limitations in analyzing ITH alongside widespread chromosomal complexity[6].

Single-cell genomic sequencing has emerged as a promising technique to investigate ITH through somatic copy-number profiling[7–10].

However, copy-number profiles do not capture the full karyotypic heterogeneity in malignancies with complex structural variant patterns, such as CK-AML, because copy-balanced and complex rearrangement structures remain typically unresolved in these malignancies[6,10,11]. In addition, the connections of cell genotype, epigenotype, phenotype and function remain underexplored in malignancies that exhibit extensive karyotypic complexity and genetic heterogeneity, such as CK-AML. Thus, the prevalence of genetic and nongenetic mechanisms driving disease progression and resistance remain underexplored[12].

In the present study, we extended the understanding of patterns of ITH during CK-AML evolution and exemplified the translational relevance of single-cell clonal evolution analyses. We harnessed two single-cell multiomics frameworks (single-cell nucleosome occupancy

✉e-mail: jan.korbel@embl.org; a.trumpp@dkfz-heidelberg.de

and genetic variation analysis (scNOVA[13])), based on single-cell template strand sequencing (Strand-seq[14]) and cellular indexing of transcriptomes and epitopes by sequencing (CITE-seq[15]), coupling single-cell transcriptomics with cell-surface, protein-level measurements—linking genotype and phenotype in eight patients with primary CK-AML and two longitudinally collected samples. We combined this single-cell characterization with functional xenotransplantation assays and ex vivo drug-sensitivity profiling.

## Results

### Genetic complexity drives karyotype heterogeneity in CK-AML

To gain insight into the evolution of genomic rearrangements and the resulting phenotypic complexity in CK-AML, we established a single-cell multiomics framework to study heterogeneity of structural variants together with nongenetic properties at single-cell resolution. We coupled scNOVA[13] with droplet-based CITE-seq[15], to reveal the scNOVA–CITE framework outlined in Fig. 1a (Supplementary Fig. 1a–c). To allow comprehensive insight into CK-AML genetic complexity, we generated Strand-seq libraries from bone marrow or peripheral blood cells of eight patients with primary CK-AML from diagnosis or salvage samples, five matched patient-derived xenografts (PDXs) and two matched relapse or refractory samples with 855 single-cell genomes sequenced overall (Fig. 1a and Supplementary Table 1). Each single-cell library was sequenced to a mean of 365,436 mapped nonduplicate read-pairs, amounting to ~0.017× coverage per cell (Supplementary Table 2 and Supplementary Fig. 1a).

Capitalizing on the Strand-seq data generated, we first focused on the eight diagnosis or salvage CK-AML samples. Performing structural variant detection with the single-cell tri-channel processing (scTRIP) method[16], we identified an average of 18.9 (±2.9 s.d.) chromosomal alterations per cell, including interstitial structural variants, terminal gains and losses, whole-chromosome aneuploidies, balanced structural variants and complex chromosomal rearrangements (Fig. 1b and Supplementary Table 3). In each patient with CK-AML, 3–12 chromosomes harbored at least one chromosomal alteration present at high cell fraction (>80%) (Fig. 1b and Supplementary Table 3), with CK282 exhibiting the highest number of alterations ($n = 50.3$, mean per single cell). Although chromosomes 5 and 12 were most frequently mutated at a high cell fraction (present in 5 out of 8 patients), chromosomes 10, 13, 19 and 22 did not show detectable high cell fraction aberrations in any patient (Fig. 1b and Supplementary Table 3). These data underscore the extensive karyotypic complexity of CK-AML.

Analysis of clonal structural variants present at high cell fractions revealed several instances of complex structural variant formation, highlighting considerable chromosomal instability of CK-AML. In patient CK282, the copy-number profiles of chromosomes 12 and 17 oscillated between three states and displayed islands of deletions

(dels), inversions (invs) and inverted duplications with at least 15 and 6 detected breakpoints, respectively (Fig. 1c,d). For both chromosomes, resolving the structural variants by chromosome-length haplotype revealed only a single rearranged homolog (Fig. 1c,d and Supplementary Fig. 2a), suggesting that the respective structural variant profiles resulted from chromothripsis[1,17,18]. By quantifying the co-segregation footprints of the directional reads using scTRIP[16], we identified 15 high-confidence translocations (Supplementary Table 4) that fused fragments of these complex rearrangements into both derivative and marker chromosomes—an observation verified by multiplex fluorescence in situ hybridization (M-FISH) and ultra-long DNA molecule optical genome mapping (OGM) (Fig. 1e and Supplementary Fig. 2b).

We also detected complex, clonal rearrangements affecting chromosomes commonly rearranged in AML. In patients HIAML85 and CK397, fragments from one 3q haplotype (H1) contained intrachromosomal rearrangements spanning the 3q arm in all cells. HIAML85 cells contained one large inversion, whereas CK397 cells harbored a complex intrachromosomal rearrangement involving at least three large inversions (Fig. 1f and Extended Data Fig. 1a). Reconstruction of the 3q arm using OGM confirmed both rearrangements, validating the Strand-seq-based data (Fig. 1f and Supplementary Fig. 3a). In patient HIAML85, the single inv(3)(q21.3q26.2) generated the oncogenic *RPN1*–*MECOM* fusion (Fig. 1f), commonly seen in 3q-rearranged AML[19,20]. In patient CK397, the kilobase-scale resolution provided by OGM identified 11 intrachromosomal fusions spanning the 3q arm with inv(3)(q21.3q26.2) and inv(3)(q26.2q29) also generating a *RPN1*–*MECOM* fusion (Fig. 1f and Supplementary Fig. 3b), further verified by RNA sequencing (RNA-seq; Supplementary Fig. 3c). In both patients the 3q rearrangement resulted in overexpression of *MECOM* (Extended Data Fig. 1b) and an H1-specific reduction in nucleosome occupancy in CK397 (Extended Data Fig. 1c). Hence, by leveraging the ability of Strand-seq to characterize structural variants in a haplotype-aware manner along each homolog, our data revealed balanced as well as complex intrachromosomal 3q rearrangements as driver events, resulting in overexpression of the poor prognosis oncogene *MECOM*[21].

To further quantify ITH using Strand-seq, we calculated the structural variant burden per CK-AML cell (ranging between 0 SV- and 63 SV-altered segments per cell as identified by scTRIP; Fig. 1g and Supplementary Table 3) and applied the standard deviation of the structural variant burden as a measure of intrapatient karyotype heterogeneity. CK282 had both the highest structural variant burden ($n = 50.3$, mean per single cell) and intrapatient karyotype heterogeneity (s.d. 9.3) followed by CK349 (s.d. 6.3) (Fig. 1g). By contrast, the two *MECOM*-overexpressing samples, CK397 and HIAML85, did not show extensive intrapatient karyotype heterogeneity (s.d. 0.5 and 0.3, respectively) despite CK397 exhibiting the third highest structural variant burden ($n = 22.0$, mean per single cell) (Fig. 1g). These data underscore that, although intrapatient karyotype heterogeneity is

**Fig. 1 | Complex chromosomal rearrangements drive karyotype heterogeneity in CK-AML. a**, Schematic study layout of single-cell multiomics profiling with scNOVA and CITE-seq, applied to eight samples from patients with primary CK-AML at initial sampling, five matching PDXs and two matching refractory or relapse samples. scNOVA was used to assess structural variant (SV) landscapes and nucleosome occupancy (NO). CITE-seq was applied to assess transcriptomes and cell-surface proteomes. Panel **a** created with BioRender.com. **b**, Karyotype heatmap of 542 single cells arranged using Ward's method for hierarchical clustering of structural variant genotypes in eight patients at initial sampling. **c**,**d**, Strand-specific read depth of a representative single cell from CK282 showing clustered deletions, inverted duplications and inversions along a single homolog chromosome 12 (**c**) and chromosome 17 (**d**), resulting from clonal chromothripsis. Reads denoting somatic structural variants, discovered using scTRIP, were mapped to the Watson (orange) or Crick (green) strand. Gray indicates single-cell IDs. **e**, Circos plot illustrating complex rearrangements and translocations involving multiple chromosomes, assessed by OGM from a PDX of CK282. Chromosomes (outside of the circular plot) and chromosomal

rearrangements are shown as arcs connecting the two relevant genomic regions in the middle. The data are represented as follows (starting from the outer ring): structural variants, copy-number variation and translocations. **f**, Chromosome view of 3q in HIAML85 and CK397 with mapping of segments by Strand-seq (top) and OGM (bottom) showing inversions spanning parts of the q arm. In Strand-seq, composite reads shown were taken from all informative cells in which reads could be phased (Watson–Crick or Crick–Watson configuration). The black vertical dotted lines indicate the breakpoint positions of inversions. In OGM, de novo genome maps (blue) are aligned to the reference genome (yellow) with gray lines showing connecting genomic segments. **g**, Karyotype heterogeneity in eight samples from patients with CK-AML based on structural variant burden (bottom) and its s.d. (top). Each gray dot represents a single cell in CK282 ($n = 76$), CK295 ($n = 41$), CK397 ($n = 70$), CK349 ($n = 91$), P9D ($n = 44$), HIAML47 ($n = 91$), D1922 ($n = 63$) and HIAML85 ($n = 66$); Point ranges were defined by minima = mean − 2× s.d., maxima = mean + 2× s.d., point = mean. Dup, duplication; InvDup, inverted duplication; Tra, translocation.

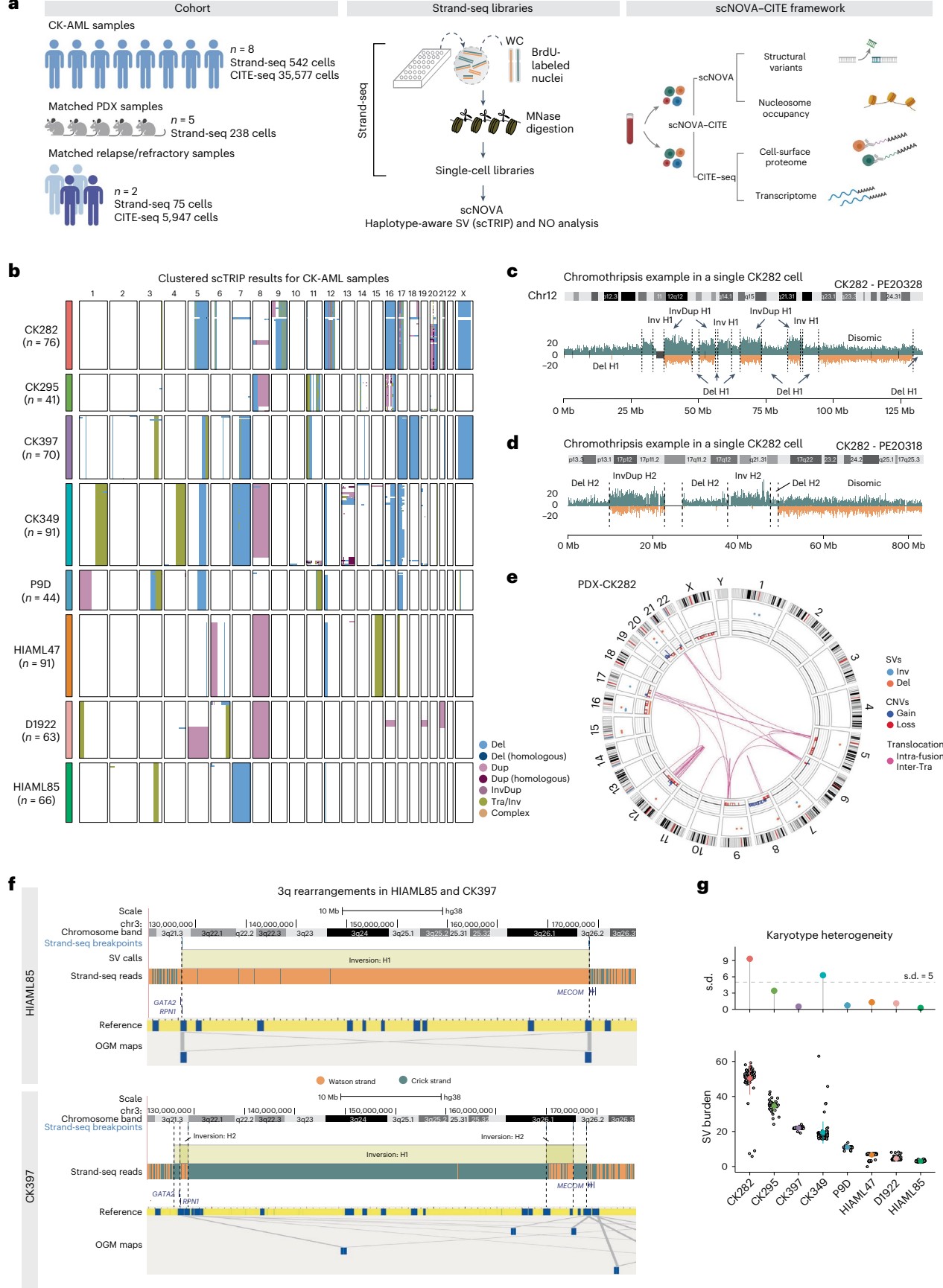

widespread in CK-AML, this is not necessarily linked to the overall structural variant burden in a patient, but instead reflects individual subclonal diversity levels.

## Different modes of clonal dynamics in CK-AML

To gain further insights into CK-AML subclonal evolution, we carried out a comprehensive analysis of structural variant subclonality for each diagnosis or salvage sample. We observed three distinct subclonal growth patterns: (1) monoclonal growth, (2) linear growth and (3) branched polyclonal growth (Fig. 2a). Two of eight cases exhibited monoclonal growth, whereby a single subclone was dominant at the time of sampling and only individual cells deviated from the main clone (Fig. 2b and Supplementary Table 3). In the remaining six cases, we identified oligo- or polyclonal growth, whereby multiple subclones were present. Of these, three showed linear and three branched growth patterns (Fig. 2a). As expected, the two samples with the highest intra-patient karyotype heterogeneity showed branched growth patterns (Fig. 1g).

In the two patients with monoclonal growth, structural variants were shared between all cells (excluding singleton events) affecting 3 chromosomes in patient HIAML85 and 12 chromosomes in patient CK397 (Fig. 2b and Supplementary Table 3). Both patients harbored inversions at 3q, generating the recurrent oncogenic *RPN1–MECOM* fusion described above (Fig. 1f and Supplementary Fig. 4a). By contrast, the three patients with linear growth (CK295, P9D and HIAML47) were characterized by a step-wise acquisition of structural variants (Fig. 2c). In each of the three patients a set of structural variants was present in virtually all cells (Fig. 2c) and thus probably originate from a common precursor AML cell. In all cases, we identified additional structural variants acquired in a step-wise manner, generating the dominant clone at the time of sampling (Fig. 2c and Supplementary Fig. 4b). Structural variants acquired later in disease evolution generally overlapped regions with known oncogenes and tumor suppressors, such as *MYC* at 8q, *CDKN1B* at 12p and *TP53* at 17p. Notably, one cell (1 out of 91 cells, 1.1%) in patient HIAML47 lacked detectable structural variants (Fig. 2c,d). As this patient progressed from a *JAK2*-mutant myeloproliferative neoplasm (MPN) or chronic myelomonocytic leukemia (CMML) to AML (Supplementary Table 1), this cell hints at the presence of residual MPN- or CMML-related blood cells at the time of CK-AML diagnosis. Collectively, these findings underscore the selective growth advantage gained by the acquisition of additional structural variants in a linear step-wise process, probably leading to a successively more aggressive malignancy.

The branched polyclonal growth cases (D1922, CK282 and CK349) harbored multiple subclones displaying differences in their karyotypical complexities (Fig. 2e and Supplementary Table 3). Similar to the linear growth samples, we identified a set of structural variants that were present in virtually all cells, indicative of a common precursor cell. In patient D1922, all cells harbored a polyploid chromosome 8 together

with translocation signatures at 1p and 6q, whereas, in patients CK349 and CK282, seven and ten chromosomes, respectively, carried both simple gains and losses as well as complex rearrangements (Fig. 2e). Among the branched polyclonal growth cases, patient D1922 had the lowest structural variant burden (*n* = 4.5, mean per single cell) and largely lacked complex rearrangements (Fig. 2e). We detected five subclones, referred to as SC1–SC5, that were characterized by distinct sets of whole-chromosome duplications, affecting five chromosomes (chromosomes 5, 16, 19, 20 and 21; Fig. 2e). In patient CK349, we classified cells into three main subclones, referred to as SC1, SC2 and SC3, each with distinct structural variant burdens (Fig. 2e). SC1 (81 out of 91 cells, 89%) represented the largest clone and harbored uniquely a chromosome 8 trisomy (Fig. 2e). By contrast, SC2 (5 out of 91 cells, 5.5%) and SC3 (5 out of 91 cells, 5.5%) carried a distinct set of rearrangements affecting chromosome 13 (Fig. 2e and Extended Data Fig. 2a,b). SC3 had additionally acquired a set of structural variants at chromosome 11, resulting in wave-like, copy-number profiles (discussed further below). Finally, patient CK282 showed the most abundant subclone diversity, represented by five distinct subclones and characterized by 6–59 structural variant-altered segments in each cell (Fig. 2e and Extended Data Fig. 3a). Three subclones, referred to as SC1, SC2 and SC3, showed a high level of genetic similarity, with the exception of structural variants identified on chromosomes 8 and 20 (Fig. 2e and Extended Data Fig. 3a,b). SC4 (19 out of 76 cells, 25.0%) lacked rearrangements on chromosome 20 but displayed several unique structural variants, including three duplications on chromosomes 9, 12 and 18, and one inversion on chromosome 17, respectively (Fig. 2e and Extended Data Fig. 3a). By comparison, SC5 (3 out of 76 cells, 3.95%) differed markedly from all other subclones and harbored a distinct and much smaller structural variant set, which almost entirely lacked complex rearrangements abundant in SC1–SC4 (Fig. 2e and Extended Data Fig. 3a), suggesting parallel evolution from a common precursor stem cell harboring an inversion at 3q.

Together, our single-cell assessment of subclonal growth patterns in CK-AML add new insight into the clonal dynamics in diagnosis or salvage CK-AML, and showcase that multiple clones can exist and expand simultaneously in CK-AML. A detailed description of the structural variants in all samples and subclones can be found in Supplementary Note 1.

## Single cells with excessive chromosomal instability

Beyond the assessment of subclonal growth patterns, our analysis of structural variants restricted to an individual cell revealed evidence for genomic regions subject to extensive chromosomal instability. As an example, we noted that chromosome 20 in CK282 subclones SC1, SC2 and SC3 displayed a classic breakage–fusion–bridge (BFB) event[16,22] with the typical inverted duplication and adjacent terminal deletion signature arising on the same haplotype, but with the length of the terminal deletion varying from cell to cell (Extended Data Fig. 4a,b and Supplementary Note 1). Likewise, all CK349 cells displayed deletions on

**Fig. 2 | CK-AML is characterized by different modes of clonal dynamics and ongoing instability. a**, Patterns of subclonal growth observed in patients with CK-AML at initial sampling. **b,c**, Manually curated clonal trees showing the hierarchy of somatic structural variant subclones discovered using scTRIP for samples showing monoclonal (**b**) and linear (**c**) growth. Each colored circle represents a subclone of genetically similar cells. The accumulated structural variants can be traced with solid lines toward the root. The size of the circle is proportional to the clonal population and the percentage within or next to each circle is the percentage of each clone among the total cells. **d**, Strand-specific read depths of chromosomes 6 (upper), 8 (middle) and 12 (bottom) in three representative single cells from HIAML47. The arrow on the clonal tree indicates the subclone represented. **e**, Manually curated clonal trees for samples showing branched polyclonal growth. Karyotype heterogeneity in the different subclones, which is based on structural variant burden (bottom) and its s.d. values (top), is shown next to the clonal trees. Each gray dot represents

a single cell. The structural variant burden between subclones was compared using two-tailed Wilcoxon's test (D1922: SC1 (*n* = 30), SC2 (*n* = 5) and SC3 (*n* = 17), SC4 (*n* = 7) and SC5 (*n* = 4); CK282: SC1 (*n* = 15), SC2 (*n* = 4), SC3 (*n* = 34), SC4 (*n* = 19) and SC5 (*n* = 3); CK349: SC1 (*n* = 5), SC2 (*n* = 5) and SC3 (*n* = 81)). Point ranges were defined by: minima = mean − 2× s.d.; maxima = mean + 2× s.d.; point = mean. **f**, Strand-specific read depth of four representative single cells from CK349 depicting different amplification statuses. DNA reads are colored as follows: Watson, orange; Crick, green. **g**, Model for the evolution of seismic amplification in CK349. Panel **g** created with BioRender.com. **h**, Two-color FISH of ring chromosome 11 from PDX of CK349 using 11p (green) and 11q (red) partial chromosome painting (pcp) probes. Scale bar, 10 μm. In **b–e**, the size of the circle is proportional to the clonal population. [a]Engraftment-driving subclone (Figs. 4 and 5). [b]Differing breakpoints affecting the same chromosome. CF, cell fraction; Cx, complex; Inter, interstitial; Ter, terminal.

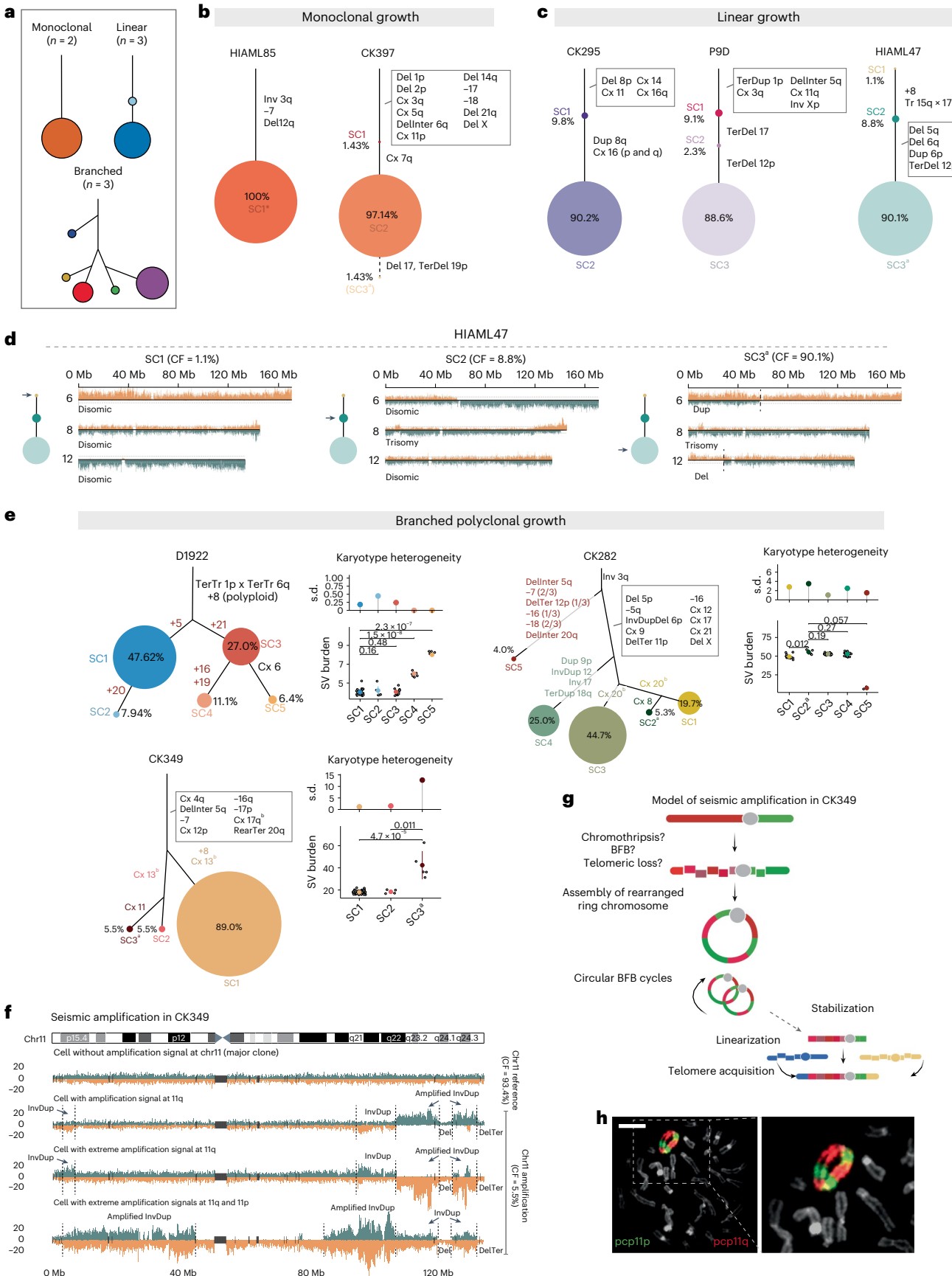

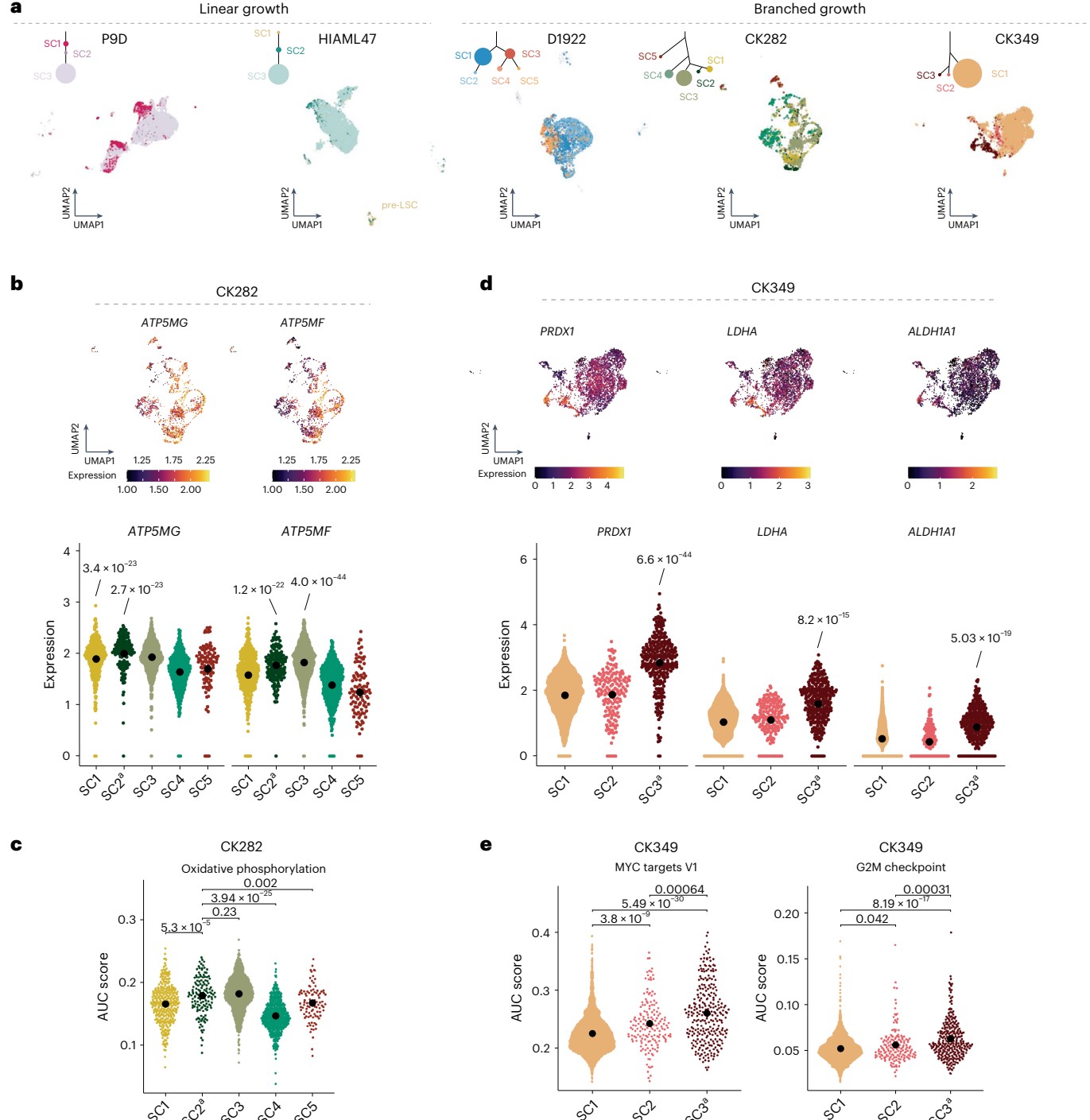

**Fig. 3 | Transcriptome provides mechanistic insight into subclonal architecture. a**, Weighted nearest neighbor-based UMAP plots of leukemic cells from CITE-seq data faceted by growth pattern. Cells are colored based on the subclones identified using scTRIP depicted above the UMAP in the clonal tree, with the size of the circle relative to the clonal population. Annotation of each cell was based on targeted SCNA recalling using CONICSmat. **b**, Expression of *ATP5MG* and *ATP5MF* in single cells and subclones in CK282 ($n$ = 95–796 single cells). **c**, Area under the curve (AUC) score for activity of oxidative phosphorylation-associated gene set for each cell in the different subclones ($n$ = 95–796 single cells). **d**, Expression of *PRDX1*, *LDHA* and *ALDH1A1* in single cells and subclones in CK349 ($n$ = 162–2,553 single cells). **e**, AUC score for activity MYC targets G2M checkpoint-associated gene sets for each cell in the different subclones ($n$ = 162–2,553 single cells). In **b**–**e**, beeswarm plots show the 95% confidence interval (CI) for the mean, gene expression comparisons show the $P_{adj}$ values from two-sided, pairwise Welch $t$-tests between subclones and AUC scores were compared using two-tailed Wilcoxon's test followed by Benjamini–Hochberg multiple correction testing. Expression levels of the individual genes in the score were calculated from normalized and variance-stabilized counts. [a]Engraftment-driving subclone (Figs. 4 and 5).

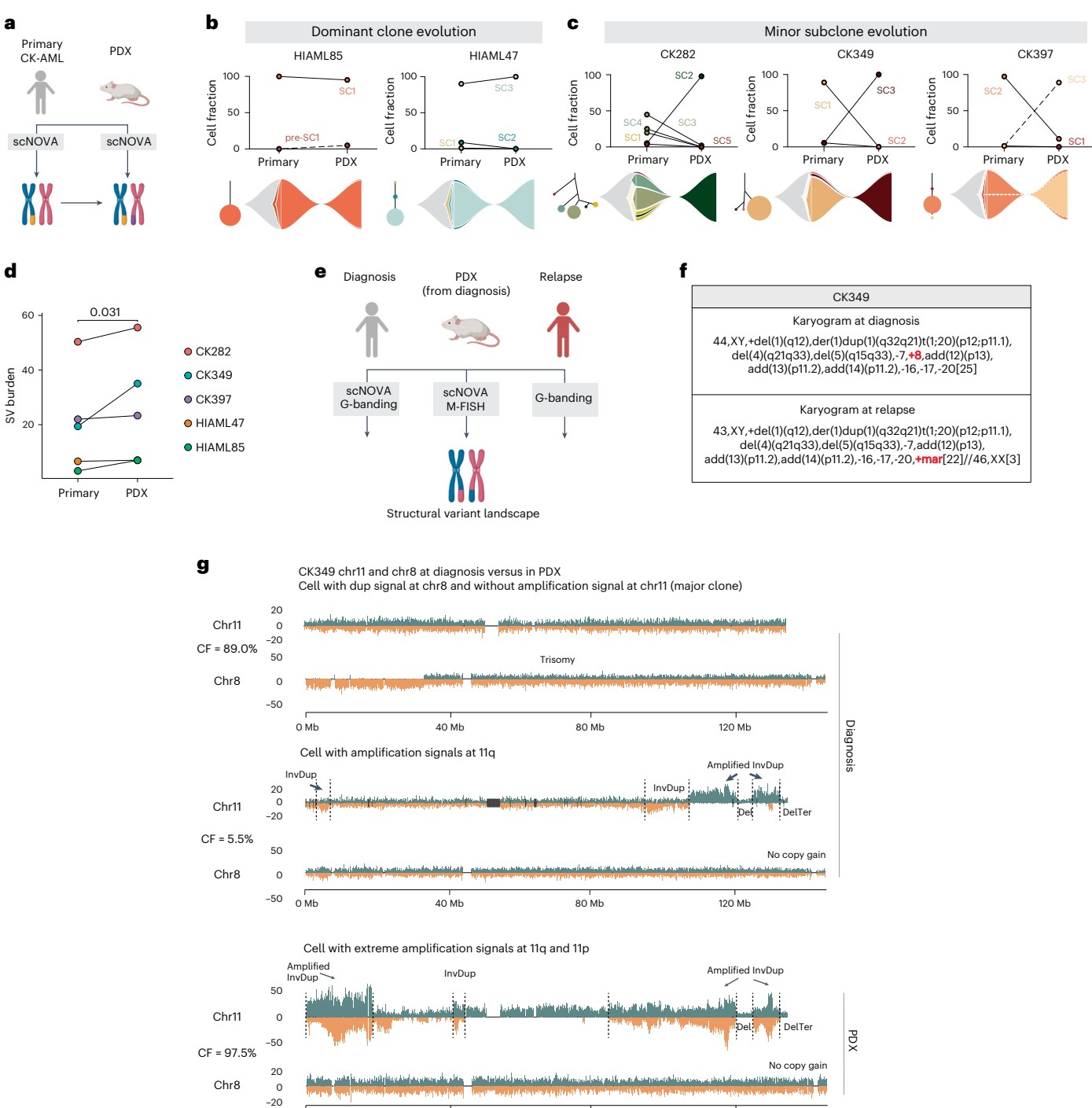

**Fig. 4 | Different clonal evolution patterns contribute to CK-AML reconstitution in mice. a**, Schematic of the structural variant landscape comparison between primary CK-AML samples and matched PDXs. **b,c**, CK-AML reconstitution is driven by dominant clone (**b**) or minor subclone (**c**). The cell fraction of subclones in the primary sample and the matching engrafted cells in the PDX model are shown. Lines connect different time points (initial sample versus PDX) of the same subclone (top). Fish plots (bottom) show the inferred clonal evolution patterns and the subclonal trees the hierarchies of somatic structural variant subclones in the primary samples, with the size of the circle relative to the clonal population. **d**, Mean structural variant burden in the primary CK-AML samples and matched PDX models. Each dot represents a sample. Structural variant burden between primary and PDX samples was

compared using one-tailed, paired Wilcoxon's test. **e**, Schematic of the structural variant landscape comparison across diagnosis, PDX and relapse samples from CK349. Panels **a** and **e** created with BioRender.com. **f**, G-banding karyograms of CK349 at diagnosis and at relapse. Structural variants differing between the two time points are highlighted in red. **g**, Depiction of two example CK349 cells at diagnosis and one in PDX with differing levels of amplification at chr11, based on no amplification at diagnosis (upper, major clone), marked amplification at diagnosis (middle, minor clone) and extreme amplification in PDX (lower, major clone). For each cell, the chr8 trisomy status is shown beneath, which scTRIP inferred to be mutually exclusive with chr11 amplification. Add, addition; Der, derivative; Mar, marker chromosome; t, translocation.

chromosome 17, with these events partially overlapping and presenting 15 unique, nonoverlapping breakpoints, pointing to persistent chromosomal instability involving this chromosome (Extended Data Fig. 4c,d and Supplementary Note 1).

We also detected subclone-specific chromosomal instability. The five cells comprising the SC3 of patient CK349 exhibited the highest degree of karyotype heterogeneity across all cells (Figs. 1g and 2e). These cells exhibited a diversity of complex rearrangements affecting chromosome 11, comprising amplifications at different genomic positions and reaching distinct copy-number levels, interrupted by nonamplified disomic and/or deleted segments (Fig. 2f and Extended Data Fig. 5a,b). Closer inspection of the amplified regions showed highly variable and oscillating copy-number states, which differed from one-off chromothripsis events that yield typically only two (or occasionally three) oscillating copy-number states (Fig. 1c,d and Supplementary Fig. 5a)[17,18]. These wave-like, copy-number events also differed from other amplification events that contained distinct structural variant breakpoints demarking a single copy-number state (Extended Data Figs. 2a and 3b and Supplementary Fig. 5b). Instead, these rearrangement patterns are indicative of the occurrence of seismic amplifications, a class of complex structural variants recently described in solid tumors from bulk whole-genome analysis[23,24]. Given the multistep rearrangement process involved in seismic amplifications[23,24], the unique breakpoints and amplification states observed in each cell with a high structural variant burden in CK349 may result from successive circular recombination events initiated on chromosome 11 (Fig. 2g). Indeed, M-FISH analysis of a PDX sample generated from CK349 revealed a large ring chromosome containing several copies of segments from 11p and 11q (Fig. 2h), confirming the presence of a circular DNA structure. This is likely to promote chromosomal instability and acquisition of intrapatient karyotype heterogeneity in patient CK349. Linearized marker chromosomes containing segments from chromosome 11 were likewise present (Extended Data Fig. 5c), suggesting stabilization of the seismic amplification process in a subset of cells. Our findings are notably consistent with, and hence validate, the previously proposed model of circular recombination[23,24], which our data reveal can act as a source of cell-to-cell DNA rearrangements fostering ITH in CK-AML. A detailed characterization of the chromosome 11 events can be found in Supplementary Note 2.

## Epigenetic and transcriptomic insight into patient subclones

The impacts of larger structural variants on the cell epigenome, transcriptome and cell-surface proteome in AML remain unexplored as a result of the current lack of appropriate genomic technologies. To address this gap, we harnessed the distinct multimodal, single-cell readouts accessible through scNOVA and CITE-seq. Capitalizing on the high-resolution structural variant breakpoint coordinates obtained from Strand-seq, we utilized the CONICSmat[25] computational method

to pursue targeted somatic copy-number alteration (SCNA) recalling in the CITE-seq data to integrate the single-cell readouts, thereby expanding the number of assessed single cells to 35,577 (Extended Data Fig. 6a and Methods). In five of six patients exhibiting polyclonal growth, we confidently assigned cells from the CITE-seq data to the corresponding subclones defined by scNOVA[13] (Fig. 3a, Extended Data Fig. 6b and Supplementary Fig. 6). We observed a marked correlation between Strand-seq and CITE-seq subclone detection (Spearman's $R = 0.7$, $P = 0.0003$; Extended Data Fig. 6b,c and Supplementary Fig. 6), suggesting that both single-cell techniques provide a similar representation of subclonal frequencies. Within each patient, integration of the CITE-seq data showed clustering of the cells mostly by genetic subclone, with each subclone exhibiting distinct transcriptomic and immunophenotypic profiles (Fig. 3a). This effect was most evident in patients with branched growth, suggesting stronger phenotypic differences between competing subclones.

Leveraging the SCNA recalling in the CITE-seq data, we were able to obtain further insight into each subclone identified using scTRIP. For example, in patient HIAML47, we rediscovered the presence of primitive myeloid cells lacking structural variants ($n = 77$ cells) (Fig. 3a and Extended Data Fig. 7a), confirming the presence of pre-LSCs also identified using Strand-seq (Fig. 2c). These pre-LSCs (SC1) showed upregulation of multiple interferon (IFN) response genes (for example, *IFITM1*, *IFITM2* and *IFITM3*) (Extended Data Fig. 7b,c and Supplementary Table 5), commonly upregulated in MPNs[26]. This was further recapitulated by pathway analysis whereby INFγ and INFα response gene sets, as well as the JAK–STAT signaling pathway, showed strongly enriched activity (Extended Data Fig. 7d), providing additional support for our hypothesis that the pre-LSCs represent residual persister cells of the preceding MPN or CMML disease rather than healthy hematopoietic stem or progenitor cells (HSPCs). By contrast, the dominating subclone harboring the most structural variants (SC3) in HIAML47 showed the lowest IFN and JAK–STAT signaling, but increased expression of cell cycle-associated genes (for example, *E2F3*, *EIF4E*, *EIF3H* and *EIF3J*) (Extended Data Fig. 7b,c and Supplementary Table 5). This was further reflected in the upregulation of the G2M checkpoint and mitotic spindle-associated gene signatures (Extended Data Fig. 7d). These findings are consistent with the selective growth advantage observed for this subclone.

We also gained insight into the molecular expression networks of patients displaying branched growth. Subclones from the same evolutionary branch typically expressed similar transcriptomic programs (Supplementary Note 3 and Supplementary Fig. 7). For example, in patient CK282, cells from SC1, SC2 and SC3 showed upregulation of genes involved in mitochondrial complex V (*ATP5MF*, *ATP5MG* and *ATP5MD*) (Fig. 3b and Supplementary Table 5) and enrichment of oxidative phosphorylation (Fig. 3c). In patient CK349, the transcriptomic data also reflected the extensive chromosomal instability observed

**Fig. 5 | Levering single-cell multiomics to dissect drug–response profiles of functional LSCs. a**, Schematic of the drug–response profiling using cell-surface proteins from CITE-seq data to capture distinct subclones by flow cytometry. Panel **a** created with BioRender.com. **b**, Heatmap showing differentially expressed cell-surface markers for subclones in CK282. **c**, Viabilities of blasts from three CK-AMLs after 24 h of ex vivo exposure with indicated conditions. The mean viabilities of two replicates are shown. **d**, Scatter plot of CD34 and GPR56 expression from HIAML47 CITE-seq data pre-gated to (pre)leukemic cells. **e**, FACS plot displaying expression of CD34 and GPR56 on untreated pregated leukemic cells in HIAML47. Engraftment-driving LSCs are highlighted in red. **f**, Viabilities of engraftment-driving LSCs and all blasts in HIAML47 after 24 h of ex vivo exposure with the indicated concentrations of venetoclax. Each dot represents a replicate and the line connects the mean viabilities of the two replicates. **g**, Scatter plot of CD45RA and CD49F expression from CK349 CITE-seq data pre-gated to leukemic cells. **h**, FACS plot displaying expression of CD45RA and CD49F on untreated pre-gated leukemic cells in CK349. Engraftment-driving

LSCs are highlighted in red. **i**, Viabilities of engraftment-driving LSCs and all blasts in CK349 after 72 h of ex vivo exposure with the indicated concentrations of cytarabine (Ara-C) and daunorubicin. **j**, Scatter plot of CD45RA and CD90 expression from CK282 CITE-seq data pre-gated to leukemic cells. **k**, FACS plot displaying expression of CD45RA and CD90 on untreated pre-gated leukemic cells in CK282. Engraftment-driving LSCs are highlighted in red. **l**, Viabilities of engraftment-driving LSCs and all blasts in CK282 after 24 h of ex vivo exposure with the indicated concentrations of A-1331852. Each dot represents a replicate and the line connects the mean viabilities of the two replicates. **m**, Viabilities of different CK282 populations after 24 h of ex vivo exposure with the indicated concentrations of standard chemotherapy regimens, as well as BH3 mimetics. The mean viabilities of two replicates are shown and engraftment-driving LSCs are highlighted in red. **n**, Fluorescence intensity of BCL-xL protein expression in different CK282 populations. Engraftment-driving LSCs are highlighted in red. Ex vivo viabilities were calculated as a fraction of viable cells compared with an untreated control. 5-AZA, azacitidine.

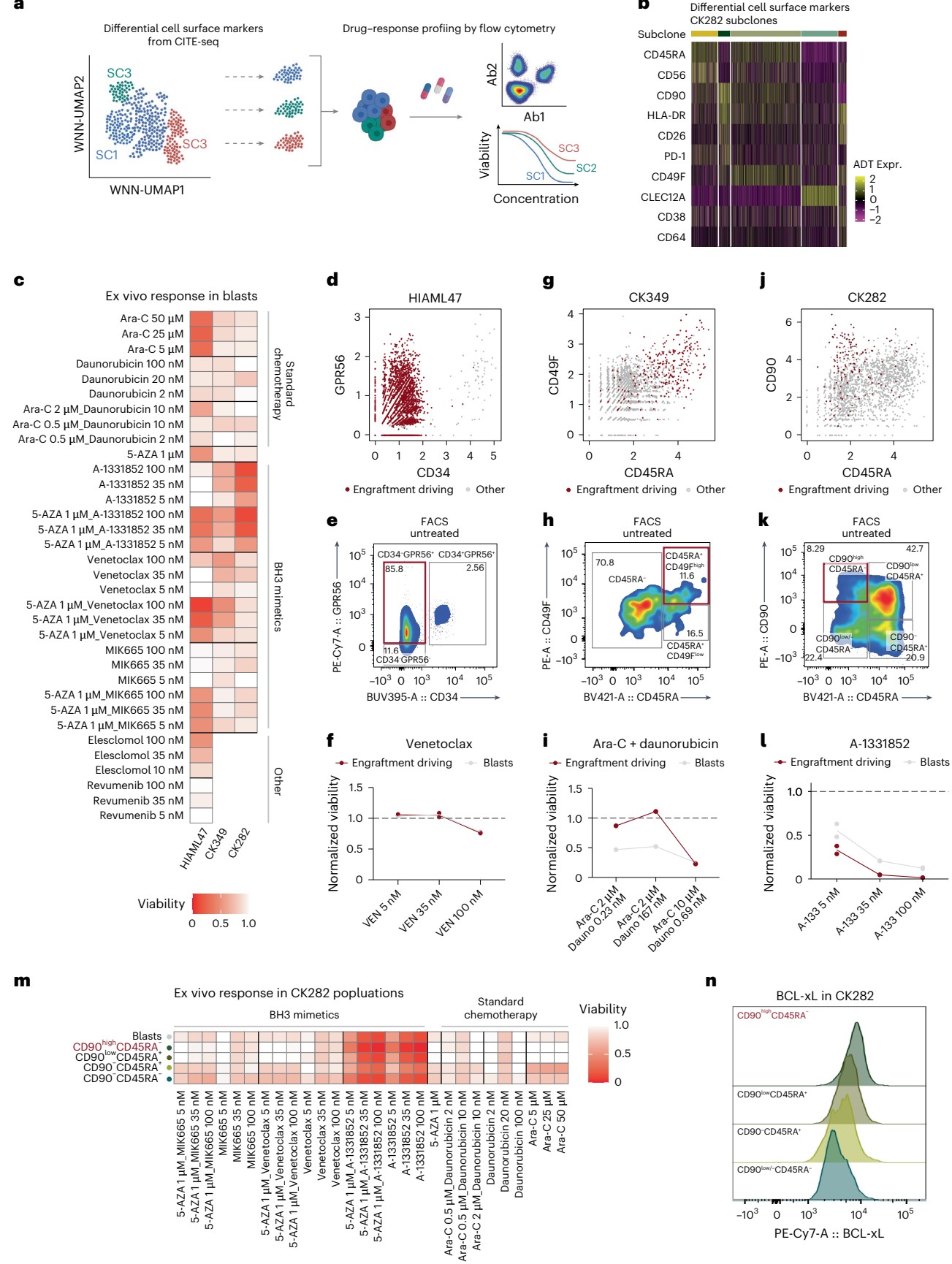

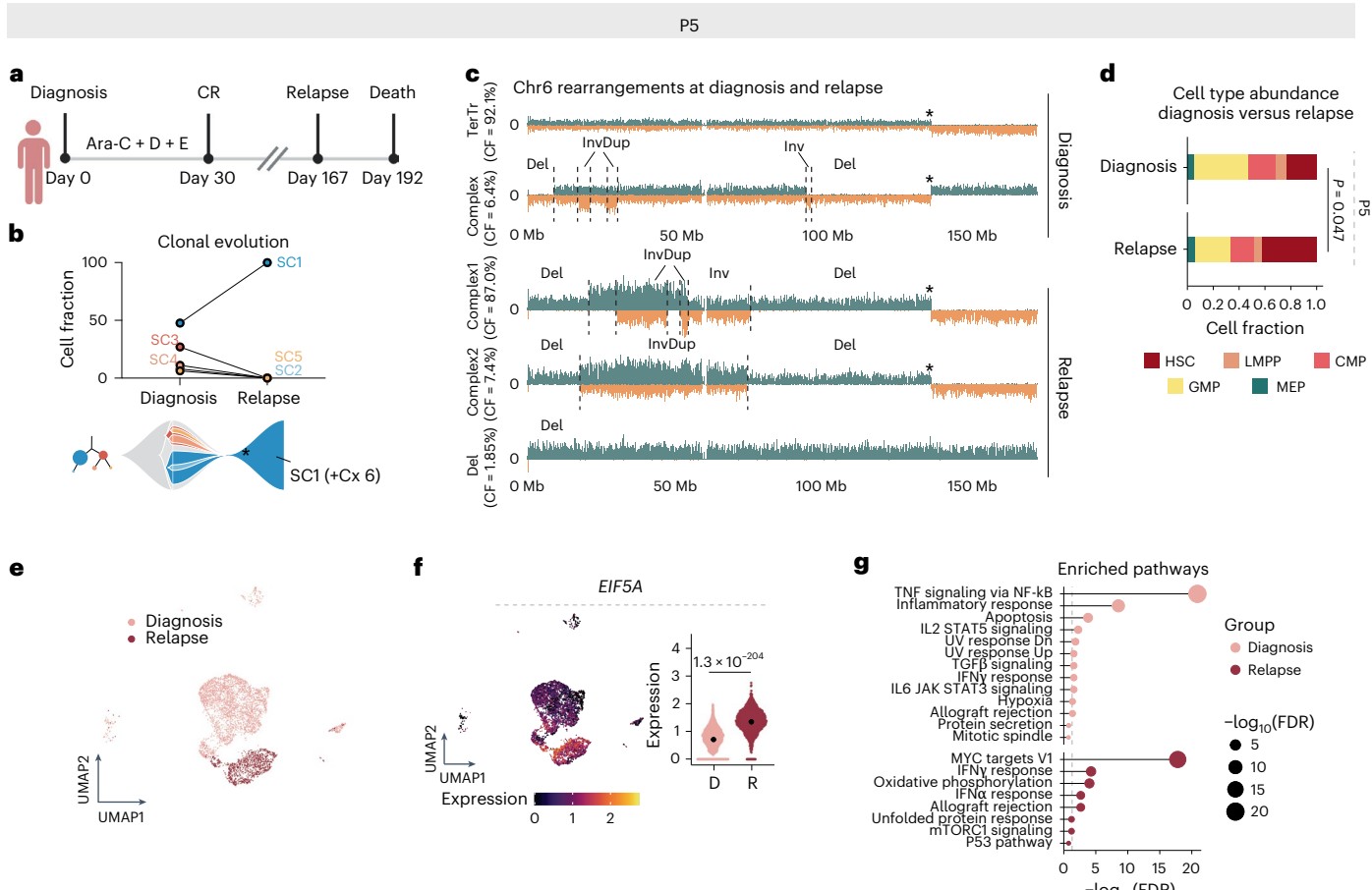

**Fig. 6 | Relapse is driven by a genetically evolving subclone in patient P5.**
**a**, Disease timeline for patient P5. Panel **a** created with BioRender.com. **b**, Cell fraction of patient P5 subclones at diagnosis (D1922) and at relapse (R0836) based on the scTRIP data. The lines connect different time points (diagnosis versus relapse) of the same subclone (top). Fish plot (bottom) shows the inferred clonal evolution pattern and the subclonal tree the hierarchy of structural variant subclones at diagnosis, with the size of the circle relative to the clonal population. **c**, Depiction of example cells at diagnosis and relapse with differing rearrangements at chromosome 6. Asterisk denotes translocation breakpoint. **d**, Stacked bar plots showing the fraction of indicated HSPC-like states out of all cells at diagnosis and relapse. Cell types were annotated using a micrococcal nuclease (MNase)-seq reference dataset from index-sorted healthy CD34+ bone marrow cells and cell typing was pursued using scNOVA. The P value indicates

the different abundance of HSC-like cells between the time points from two-sided Fisher's exact test ($n_{\text{Diagnosis-HSC}} = 15$ and $n_{\text{Relapse-HSC}} = 23$, $n_{\text{Diagnosis-other}} = 48$ and $n_{\text{Relapse-other}} = 31$). **e**, Weighted nearest neighbor-based UMAP plots of diagnosis and relapse leukemic cells from patient P5 CITE-seq data. Cells are colored based on disease stage. **f**, Expression of *EIF5A* in single cells at diagnosis and relapse ($n_{\text{Diagnosis}} = 3,444$ and $n_{\text{Relapse}} = 1,102$). Beeswarm plots show the 95% CI for the mean and the gene expression comparison shows the $P_{\text{adj}}$ value from two-sided, pairwise Welch's t-test. **g**, Enriched pathways at diagnosis and relapse. Genes with false discovery rate (FDR) < 0.05 and log(fold-change) > 0.25 were included in the analysis. CMP, common myeloid progenitor; CR, complete remission; Cx, complex; D, daunorubicin; E, etoposide; GMP, granulocyte–macrophage progenitor; TerTr, terminal translocation.

at the genetic level, caused by the seismic amplification in SC3. We observed subclone-specific increased expression of several genes involved in cellular stress and DNA-damage response (for example, *LDHA*, *SESN1*, *PRDX1*, *PRDX2*, *PRDX4*, *ATM*, *ALDH2* and *ALDH1A1*), many of which also showed reduced nucleosome occupancy (Fig. 3d, Extended Data Fig. 7e and Supplementary Tables 5 and 6), suggesting that these may be deregulated as a consequence of ongoing recombinatorial rearrangements of the respective circular DNA. It is interesting that these SC3 cells also upregulated classic cell proliferation-associated pathways, including the G2M checkpoint, MYC targets and mitotic spindle-associated gene signatures (Fig. 3e and Extended Data Fig. 7f), arguing that they may have a relatively higher proliferative activity compared with the other subclones in the same sample, which might contribute to the rapid mutation acquisition of this subclone.

In summary, our integrated framework enabled us to capture phenotypic intrapatient heterogeneity of genetically related yet distinct leukemic subclones. This revealed both shared and subclone-specific pathway dysregulation and cell-type biases (Supplementary Note 4,

Supplementary Fig. 8 and Supplementary Table 7), driving distinct molecular programs that are simultaneously present within the same patient.

**CK-AML clonal evolution patterns in mice**

We hypothesized that the observed phenotypic diversity may also result in differences in functional disease-propagating capacity. To explore this, we established PDX models for five patients (Supplementary Table 1) and analyzed the engrafting cells using scNOVA (Fig. 4a). This revealed two engraftment patterns in PDX: (1) engraftment of the dominant clone (HIAML85 or HIAML47) or (2) engraftment of a minor subclone (CK282, CK349 or CK397) (Fig. 4b,c). Detailed characterization of patient-specific clonal dynamics in the PDX can be found in Supplementary Note 5. Transcriptomically, the engraftment-driving cells shared programs involved in cell growth, proliferation and oxidative phosphorylation, whereas downregulated gene sets were associated with inflammation (Supplementary Note 5, Supplementary Fig. 9a,b and Supplementary Table 8). Overall, the engrafted CK-AMLs

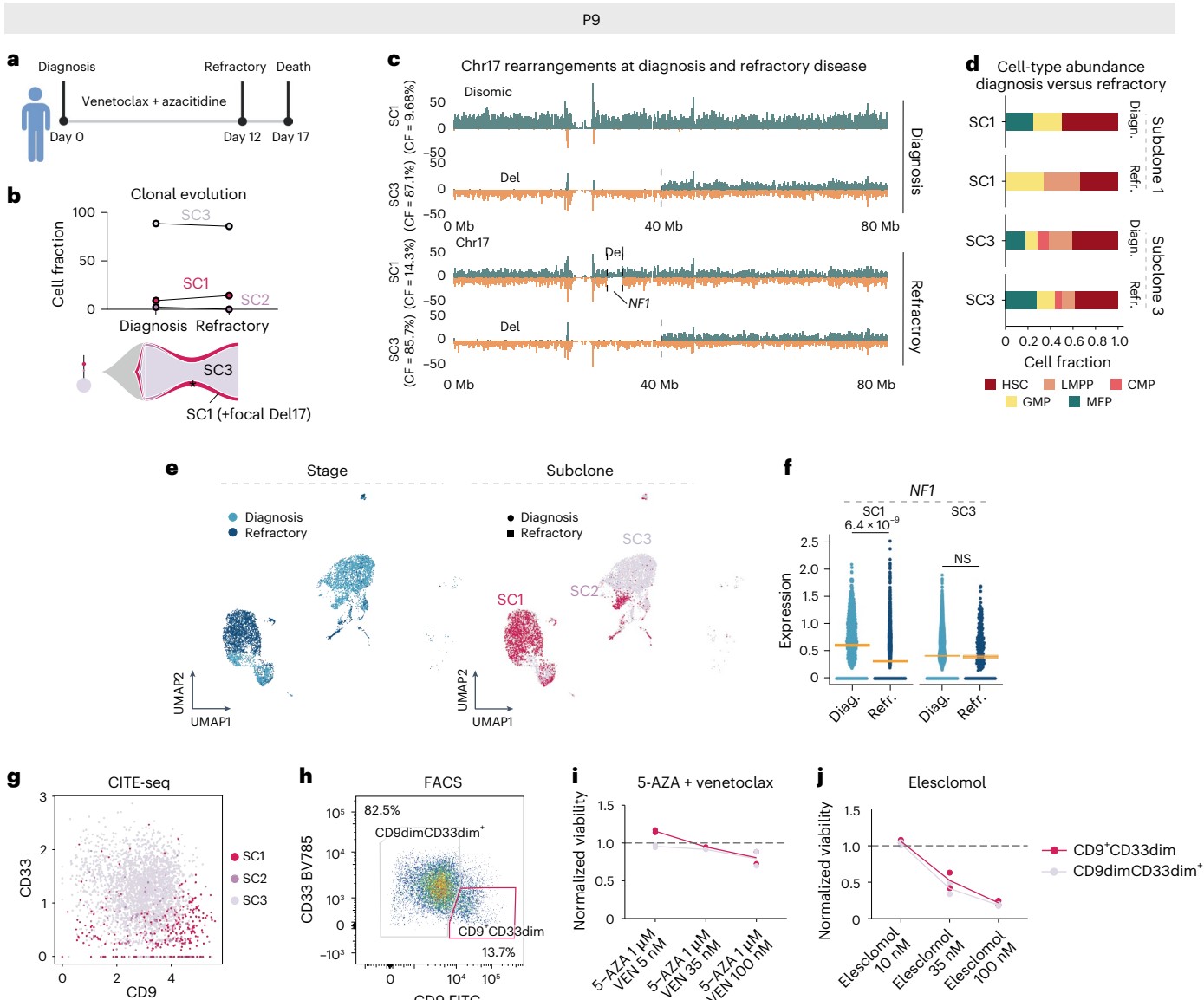

**Fig. 7 | Disease resistance is driven by subclone-specific mechanisms in patient P9. a**, Disease timeline for patient P9. Panel **a** created with BioRender. com. **b**, Cell fraction (CF) of patient P9 subclones at diagnosis (P9D) and refractory disease (P9R) based on the scTRIP data. The lines connect different time points (diagnosis (diagn.) versus refractory (refr.) disease) of the same subclone (top). Fish plot (bottom) shows the inferred clonal evolution pattern and the subclonal tree the hierarchy of somatic structural variant subclones at diagnosis, with the size of the circle relative to the clonal population. **c**, Depiction of example cells at diagnosis and refractory disease representing cells from SC1 and SC3 with differing rearrangements at chromosome 17. **d**, Stacked bar plots showing the fraction of indicated HSPC-like states out of all cells at diagnosis and refractory disease. Cell types were annotated using an MNase-seq reference dataset from index-sorted, healthy, CD34⁺ bone marrow cells and cell typing was pursued using scNOVA. **e**, Weighted nearest neighbor-based UMAP plots of diagnosis and refractory leukemic cells from P9 CITE-seq data. Cells are colored

based on disease stage (left) and subclones identified using scTRIP (right). **f**, Expression of *NF1* in single cells at diagnosis and refractory disease faceted based on subclone (SC1: $n_{Diagnosis} = 680$ and $n_{Refractory} = 1,418$; SC3: $n_{Diagnosis} = 3,130$ and $n_{Refractory} = 263$). $P_{adj}$ value from two-sided, pairwise Welch's $t$-tests between disease stages is shown and beeswarm plots show the 95% CI for the mean. **g**, Scatter plot of CD9 and CD33 expression from CITE-seq data at diagnosis (P9D) pre-gated to leukemic cells highlighted according to subclones. **h**, FACS plot displaying expression of CD9 and CD33 on pre-gated leukemic cells. The gates highlight two populations with different CD9 and CD33 expressions, representing SC1- and SC3-enriched populations. **i**, Viabilities of different populations after 24 h of ex vivo exposure with the indicated concentrations of venetoclax together with azacitidine. **j**, Viabilities of different populations after 24 h of ex vivo exposure with the indicated concentrations of elesclomol. In **i** and **j**, ex vivo viabilities were calculated as a fraction of viable cells compared with an untreated control. NS, not significant.

---

in the PDXs showed increased structural variant burden but reduced karyotype heterogeneity compared with the corresponding primary patient samples (Fig. 4d, Extended Data Fig. 8a and Supplementary Table 3), consistent with expansion of a single or a few engrafted LSCs that may continue to undergo genomic evolution. Indeed, we also found unstable chromosomes in two of five PDXs already present in the primary samples and singleton structural variants in individual cells in

four of five PDXs (Extended Data Fig. 8b–e). Thus, engraftment of LSCs in mice can be accompanied by spontaneous generation of de novo karyotype diversity.

To exemplify the clinical relevance of engraftment-driving LSCs, we analyzed karyograms from patient CK349 at relapse after chemotherapy treatment (Fig. 4e and Supplementary Fig. 10). At relapse, 88% (22 out of 25 cells) of chemotherapy-resistant cells lacked the

trisomy 8 present at diagnosis, but harbored a large marker chromosome instead (Fig. 4f). The remaining 12% (3 out of 25) had a normal female karyotype and thus originated from the allogeneic HSC transplantation donor (Fig. 4f). Similarly, engraftment in CK349 PDX was driven by cells lacking trisomy 8 but harboring the complex seismic amplification at chromosome 11 (SC3; Fig. 4c,g) with the relative size of the engraftment-driving subclone increasing from 5.5% (5 out of 91 cells) at diagnosis in the patient to 97.5% (39 out of 40 cells) in the PDX (Fig. 4c,g). M-FISH analysis of the PDX cells confirmed that the amplifications on chromosome 11 resulted in a large ring chromosome or linearized marker chromosome (Fig. 4g and Extended Data Fig. 8b), consistent with the karyotype of the relapse-driving clone. These data strongly indicate that LSCs from the most genetically unstable subclone (SC3) at the time of diagnosis not only engrafted the leukemia in the PDX, but also drove clonal relapse in patient CK349. In summary, we identified different clonal evolution fates and patterns during CK-AML reconstitution in mice. Our data further indicate that PDX engraftment-driving subclones may also drive relapse outgrowth in patients with CK-AML, as in the case of CK349 (refs. 27,28).

### Single-cell multiomics to dissect drug–response profiles

We next leveraged our single-cell multiomics data to study drug–response profiles of different genetic subclones ex vivo and examine the possible clinical relevance of functional LSCs. Based on the availability of primary material for follow-up studies, we included three patient samples that showed linear or branched polyclonal growth patterns at diagnosis (HIAML47, CK349 and CK282). We used our CITE-seq data to design antibody panels specific to the distinct subclones in each sample and assessed the drug–response profiles of each subclone by flow cytometry (Fig. 5a,b, Supplementary Fig. 11, Supplementary Table 9 and Supplementary Note 6).

In line with the known poor clinical therapy response of patients with CK-AML, all samples showed different levels of resistance to most of the tested drugs ex vivo (Fig. 5c). However, in HIAML47 and CK349, the LSC-enriched CD34⁻GPR56⁺ and CD45RA⁺CD49F⁺ cells, respectively, showed considerable response to the hypomethylating agent azacitidine (Extended Data Fig. 9a,b), supporting the favorable clinical trends for azacitidine in patients with AML and poor-risk cytogenetics[29]. It is interesting that HIAML47 cells exhibited no marked response to venetoclax monotherapy (Fig. 5d–f), even though the engraftment-driving LSCs demonstrated a notable response to high concentrations of venetoclax when combined with azacitidine (Extended Data Fig. 9a). Reflecting the ex vivo findings, patient HIAML47 exhibited an initial response to venetoclax and azacitidine treatment, but the leukemia re-emerged rapidly with an immunophenotype matching the engraftment-driving LSCs (Supplementary Figs. 10 and 11a and Supplementary Table 1). In CK349, we observed a distinct resistance exclusively in the engraftment-driving LSCs to cytarabine and daunorubicin, the same chemotherapy regimen that the patient received as first-line treatment (Fig. 5g–i, Extended Data Fig. 9b–d and Supplementary Fig. 10). Yet, the engrafted cells from CK349 showed considerable response to elesclomol (Extended Data Fig. 9e), a drug inducing apoptosis by oxidative stress[30].

CK-AML cells of CK282 showed a striking response to the BCL-xL inhibitor A-1331852. Although this was the case for all CK282 subpopulations, CD90^high^CD45RA⁻ LSC-enriched cells showed the strongest response in the primary sample (Fig. 5j–m and Extended Data Fig. 9f,g) and the PDX cells continued to be sensitive to this treatment (Supplementary Fig. 12a). In line with these results, BCL-xL protein expression levels were the highest in the engraftment-driving LSCs (Fig. 5n and Extended Data Fig. 9h). As the CD90^high^-expressing cells showed resistance to all other tested drugs, including standard chemotherapy (Fig. 5m and Supplementary Fig. 12b,c), BCL-xL inhibition may provide a valid alternative to standard chemotherapy regimens in a subset of CK-AML[31]. Beyond identifying alternative therapeutic options

to explore further, the observed drug responses of functional LSCs largely reflected the clinical responses of the patients, providing a proof-of-concept method for larger screening efforts.

### Longitudinal evolution of CK-AML in response to therapy stress

To further exemplify the biological and clinical relevance of single-cell clonal evolution analysis, we performed longitudinal scNOVA–CITE analysis on two patients (P9 and P5) where paired diagnosis or post-treatment samples were available (Supplementary Note 7). Patient P5 achieved complete remission after induction chemotherapy but relapsed 167 days later (Fig. 6a). At diagnosis the patient harbored five distinct subclones (SC1–SC5), whereas, at relapse, only SC1 cells were detected (Fig. 6b). Of the relapse cells, 98% (53 out of 54 cells) had additionally acquired a new complex rearrangement on chromosome 6, reminiscent of chromothripsis and manifesting as a marker chromosome (Fig. 6b,c, Extended Data Fig. 10a and Supplementary Table 1). Relapse cells also showed enrichment of immature HSC-like cells as evident by nucleosome occupancy-based cell typing ($P = 0.047$, Fisher's exact test; Fig. 6d), which was accompanied by increased stemness scores (Extended Data Fig. 10b). Compared with treatment-naive cells, genes involved in translation (for example, *EIF5A*, *EIF3F* and *EIF3L*) were upregulated in relapse cells, which was consistent with upregulation of MYC targets and oxidative phosphorylation gene signatures (Fig. 6e–g, Extended Data Fig. 10c,d and Supplementary Table 10). Collectively, the relapse in patient P5 was probably driven by a chromothripsis event on chromosome 6 in SC1. This generated CK-AML cells with increased stemness as well as a steady increase in cell growth and oxidative phosphorylation, driving clonal disease progression.

Unlike patient P5, patient P9 received first-line treatment with the BCL-2 inhibitor venetoclax in combination with azacitidine, but was clinically refractory (Fig. 7a and Supplementary Fig. 10). At diagnosis, P9 cells consisted of three subclones, with two persisting after 12 days of treatment (Fig. 7b). In the refractory sample, 14.3% of cells (3 out of 21 cells) resembled diagnosis subclone SC1 and 85.7% (18 out of 21 cells) resembled SC3 (Fig. 7b,c and Extended Data Fig. 10e). Post-treatment, SC3-derived cells showed an increase in megakaryocyte–erythroid progenitor (MEP)-like cells (17.9% versus 27.7%), but a decrease in lymphoid-primed multipotent progenitor (LMPP)-like cells (20.5% versus 11.1%) (Fig. 7d). Meanwhile, SC1-derived cells acquired a new 5-Mb focal deletion on chromosome 17q (Fig. 7c). This includes the *NF1* tumor-suppressor gene, which showed reduced expression specifically in the SC1-derived refractory cells (Fig. 7e,f and Supplementary Table 10). In addition, refractory cells upregulated inflammation-associated gene signatures, including tumor necrosis factor via nuclear factor κ-light-chain enhancer of activated B cells (NF-κB) signaling (Extended Data Fig. 10f,g). Finally, ex vivo drug–response profiling revealed that both SC1- and SC3-enriched populations were resistant to venetoclax monotherapy and azacitidine combination therapy already at diagnosis (Fig. 7g–i and Extended Data Fig. 10h), mimicking the clinical response. Strikingly, these venetoclax-resistant subclones showed sensitivity to elesclomol (Fig. 7j), a drug previously observed to induce cell death in venetoclax-resistant cells[32]. Collectively, patient P9 exhibited persistence of two distinct subclones post-treatment, with each having acquired subclone-specific mechanisms to further resistance: a shift toward MEP-like cells and *NF1* loss leading to increased RAS signaling. Notably, both subclones were susceptible to the oxidative stress inducer elesclomol, a finding deserving of further preclinical and clinical investigation in the future.

### Discussion

We dissected the intrapatient heterogeneity of ten samples from patients with CK-AML at unprecedented single-cell multiomics resolution, including structural variant mapping and functional assays. This approach provided intriguing insights into CK-AML heterogeneity

and revealed key resistance mechanisms. Single-cell structural variant mapping identified three modes of clonal growth in CK-AML: monoclonal, linear and branched polyclonal growth. Although previous studies using bulk whole-genome and single-cell DNA sequencing in AML have identified similar clonal evolution patterns based on single nucleotide variants[33,34], inferring evolutionary history of structural variants is highly challenging in CK-AML as a result of an extensive number of alterations (up to 63 structural variant-altered segments in individual cells) and spontaneous karyotype diversity[35,36]. Despite known limitations[14,16,37], our findings emphasize the need for single-cell resolution technologies (Supplementary Notes 8 and 9).

Strand-seq data, compared with single-cell RNA-sequencing (scRNA-seq) data, offer superior resolution for detecting structural variants and studying subclonal dynamics, often not fully captured by scRNA-seq data alone because of limited resolution[13,38]. Yet, our integrative framework coupling high-resolution genomic data based on Strand-seq and scNOVA with CITE-seq provided deeper insights into the transcriptomic states of subclones than Strand-seq alone. Using scNOVA, we identified cells with extreme chromosomal instability as well as rare pre-LSCs lacking structural variants, consistent with recent findings in secondary AML[39]. Using CITE-seq, we showed that pre-LSCs displayed reduced cell proliferation compared with the CK-AML cells in the same sample, whereas extreme chromosomal instability was reflected in the upregulation of cellular stress and DNA-damage response, together with increased proliferation. In the context of venetoclax resistance, our integrative analysis revealed subclone-specific mechanisms to further resistance such as de novo structural variant acquisition and lineage plasticity, insights that would have probably remained obscured by either single-cell method alone.

Although ex vivo drug testing provides a predictive assay for new treatments, sensitivity of the results is significantly influenced by the method used[40,41]. Bulk assays yield lower sensitivity compared with flow cytometry-based assays that enable blast and LSC-specific readouts[40,41]. In the present study, utilizing distinct cell-surface phenotypes of different subclones identified by our framework, we recapitulated clinical responses in three patients using ex vivo drug testing, effectively targeting leukemia-regenerating cells in one patient with adverse genetics using BCL-xL inhibition. Although we were not able to identify inhibitors with strong efficacy toward LSCs in all patients, our platform shows promise for discovering alternative treatments in CK-AML, which may be particularly relevant for personalized cancer therapy[42,43]. One such drug was elesclomol, which showed efficacy in both venetoclax resistance-driving subclones of patient P9. This underscores the need for expanded screening to identify patient-specific, LSC-targeting options through ex vivo drug testing with subclonal readouts.

## Online content

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

Aino-Maija Leppä [1,2,3,19], Karen Grimes [4,19], Hyobin Jeong [4,5,6,19], Frank Y. Huang [1,2,3], Alvaro Andrades[4], Alexander Waclawiczek[1,2], Tobias Boch[7], Anna Jauch[8], Simon Renders[1,2,9], Patrick Stelmach [1,2,9], Carsten Müller-Tidow [9], Darja Karpova[1,2], Markus Sohn[1,2], Florian Grünschläger[1,2,3], Patrick Hasenfeld[4], Eva Benito Garagorri[4], Vera Thiel[1,2,3], Anna Dolnik [10], Bernardo Rodriguez-Martin[4], Lars Bullinger [10], Krzysztof Mrózek [11,12], Ann-Kathrin Eisfeld [11,12], Alwin Krämer[13], Ashley D. Sanders [14,15,16,20], Jan O. Korbel [4,17,20] ✉ & Andreas Trumpp [1,2,18,20] ✉

¹Division of Stem Cells and Cancer, German Cancer Research Center (DKFZ) and DKFZ-ZMBH Alliance, Heidelberg, Germany. ²Heidelberg Institute for Stem Cell Technology and Experimental Medicine (HI-STEM gGmbH), Heidelberg, Germany. ³Faculty of Biosciences, Heidelberg University, Heidelberg, Germany. ⁴European Molecular Biology Laboratory, Genome Biology Unit, Heidelberg, Germany. ⁵Hanyang Institute of Bioscience and Biotechnology, Hanyang University, Seoul, Republic of Korea. ⁶Department of Systems Biology, College of Life Science and Biotechnology, Yonsei University, Seoul, Republic of Korea. ⁷University Hospital Mannheim, Heidelberg University, Mannheim, Germany. ⁸Institute of Human Genetics, University of Heidelberg, Heidelberg, Germany. ⁹Department of Internal Medicine V, Hematology, Oncology and Rheumatology, Heidelberg University Hospital, Heidelberg, Germany. ¹⁰Charité Medical Department, Division of Hematology, Oncology and Tumor Immunology, Berlin, Germany. ¹¹Ohio State University Comprehensive Cancer Center, Columbus, OH, USA. ¹²Clara D. Bloomfield Center for Leukemia Outcomes Research, Ohio State University Comprehensive Cancer Center, Columbus, OH, USA. ¹³Clinical Cooperation Unit Molecular Hematology/Oncology, German Cancer Research Center (DKFZ) and Department of Internal Medicine V, University of Heidelberg, Heidelberg, Germany. ¹⁴Max Delbrück Center for Molecular Medicine in the Helmholtz Association, Berlin, Germany. ¹⁵Berlin Institute of Health, Berlin, Germany. ¹⁶Charité-Universitätsmedizin, Berlin, Germany. ¹⁷Bridging Research Division on Mechanisms of Genomic Variation and Data Science, German Cancer Research Center, Heidelberg, Germany. ¹⁸German Cancer Consortium (DKTK), Heidelberg, Germany. ¹⁹These authors contributed equally: Aino-Maija Leppä, Karen Grimes, Hyobin Jeong. ²⁰These authors jointly supervised this work: Ashley D. Sanders, Jan O. Korbel and Andreas Trumpp. ✉e-mail: jan.korbel@embl.org; a.trumpp@dkfz-heidelberg.de

## Methods

### Samples from patients with primary AML
All samples were obtained from patients who provided written informed consent for the research use of their specimens in agreement with the Declaration of Helsinki. The project was approved by the Ethics Committee/Institutional Review Board of the Medical Faculty of Heidelberg and Cancer and Leukemia Group B (GALGB) (NCT-MASTER platforms S-206/2011 and S-169/2017, and GALGB studies CALGB 8461, CALGB 9665 and CALGB 20202). The protocols involved collection of bone marrow aspirates and peripheral blood samples. Part of the cohort was provided by the NCT (National Center for Tumor Diseases) Liquid and Cell Biobank, a member of the BioMaterialBank Heidelberg (BMBH). Bone marrow and peripheral blood mononuclear cells were isolated by density gradient centrifugation and stored in liquid nitrogen until further use. Patient characteristics are listed in Supplementary Table 1.

### Processing of primary AML cells for single-cell sequencing
Viably cryopreserved AML bone marrow and/or peripheral blood samples were thawed at 37 °C in Iscove's modified Dulbecco's medium (IMDM) containing 10% fetal bovine serum and treated with DNase I for 15 min (100 μg ml$^{-1}$).

**Strand-seq in leukemia cells.** For Strand-seq analysis, recovered cells were cultured using previously established protocols[44,45] with IMDM, 15% BIT (bovine serum albumin, insulin, transferrin; STEMCELL Technologies, 09500), 20 ng ml$^{-1}$ of granulocyte colony-stimulating factor (G-CSF; PeproTech, 300-23), 50 ng ml$^{-1}$ of FLT3-L (PeproTech, 300-19), 100 ng ml$^{-1}$ of stem cell factor (SCF; PeproTech, 300-07), 20 ng ml$^{-1}$ of interleukin-3 (IL-3) (PeproTech, 200-03), 100 μM β-mercaptoethanol (Thermo Fisher Scientific, 31350010), 500 nM SR1 (StemRegenin 1, STEMCELL Technologies, 72342), 500 nM UM729 (STEMCELL Technologies, 72332) and 1% penicillin–streptomycin (Sigma-Aldrich, P4458-100ML). Bromodeoxyuridine (BrdU; 40 μM) was incorporated for the duration of one cell division (52–62 h) to perform nontemplate strand labeling. Single nuclei from the appropriate time point were sorted into 96-well plates using a BD FACSMelody cell sorter, followed by Strand-seq library preparation, as described previously[14,46]. Libraries were sequenced on an Illumina NextSeq 500 sequencing platform (75-bp, paired-end sequencing protocol).

**CITE-seq in leukemia cells.** For combined scRNA-seq and antibody-derived tag sequencing (CITE-seq) analysis, recovered cells were stained with a total of 38 or 149 antibody-derived tags (ADTs) and in some cases also with a hashtag oligo (HTO; Supplementary Table 11), and sorted for live CD45$^+$ cells using a BD FACSAria II or III cell sorter. CITE-seq library preparation was performed as previously reported[15] using the Chromium Single Cell 3′ Library and Gel Bead Kit (10x Genomics, 1000128). Then, 5,000–10,000 cells were targeted for each sample and processed according to the manufacturer's instructions (10x Genomics) and 0.2 mM ADT additive oligonucleotides or 3′ feature complementary DNA Primers2 (10x Genomics) were spiked into the cDNA amplification PCR (13 cycles). After PCR, a large cDNA fraction was separated from ADTs or HTOs using 0.6× solid-phase reversible immobilization (SPRI). The cDNA fraction was processed using the 10x Genomics Single Cell 3′ v.3.1 protocol to generate the transcriptome libraries. To generate the ADT libraries, ADTs were indexed with Truseq Small RNA RPIx primers by PCR for ten cycles, followed by library purification and reamplification for five additional cycles with P5 or P7 generic primers. To generate the ADT or HTO libraries, ADTs/HTOs were indexed with Dual Index NT primers by PCR for 12 cycles, followed by library purification. ADTs or HTOs and scRNA-seq libraries were either pooled in a ratio of 25% ADT and 75% RNA or sequenced separately on an Illumina NovaSeq 6000 S1 (300 pM with 1% PhiX loading concentration, 28 + 94-bp read configuration).

### Strand-seq-based structural variant discovery
Paired-end sequencing reads were aligned to the human reference genome (GRCh38) using the Burrows–Wheeler alignment algorithm[47] and duplicated reads were marked using biobambam[48] as described previously for the Strand-seq data analysis[16]. Good quality (mapping quality MAPQ ≥ 10) and nonduplicated reads were used in the downstream analysis. Reads aligned to the Watson and Crick strands were counted separately in the 100-kb genomic bins. We used reads mapping to the Watson and Crick strands to resolve the Strand-seq data by chromosome-length haplotype[49]. Based on the read depth, strand orientation and haplotype information, structural variant calling was performed using the scTRIP method[16]. In brief, the scTRIP framework infers structural variants in the segmented data by employing a Bayesian model that estimates the genotype likelihoods for each segment and each cell. Using this Bayesian model, the most probable structural variant type was assigned to each segment, followed by manual inspection of each structural variant. Cells were assigned to subclones based on the presence of shared structural variants, whereby a subclone was defined by three or more cells sharing a set of structural variants. For cells presenting clear progeny of a larger subclone, also fewer than three cells were considered as subclones (see linear growth samples in Fig. 2c).

### Structural variant burden and intrapatient karyotype heterogeneity
Using the structural variant calls from scTRIP, individual structural variant-altered segments were annotated and counted for each cell. Structural variant burden was calculated as the sum of all identified structural variant-altered segments per cell. The s.d. of the structural variant burdens per patient was used as a measure of patient-level, intrapatient karyotype heterogeneity. For subclone-level, intrapatient karyotype heterogeneity, the s.d. of the structural variant burden per subclone was used.

### Nucleosome occupancy-based cell-type classification of CK-AML cells
Using single-cell Strand-seq libraries of CK-AML, scNOVA analysis was performed to obtain nucleosome occupancy at gene bodies for each single cell as previously described[13]. As genetic SCNA can confound the nucleosome occupancy measurement at gene bodies, copy-number normalization of nucleosome occupancy, based on the ploidy status inferred by PloidyassignR using 1-Mb bins and 500-kb sliding window (https://github.com/lysfyg/PloidyAssignR), was performed. The copy-number-normalized nucleosome occupancy matrix was used as input for the nucleosome occupancy-based cell-type classifier of HSPCs[38] to predict the most likely cell type for each single-cell Strand-seq library.

### Differentially occupied genes in subclones based on scNOVA
Using the copy-number-normalized nucleosome occupancy measurement at gene bodies, as described above, differential gene activity analysis of scNOVA[13] was performed for samples with linear or branched growth. To infer differentially active genes for each subclone, the single cells in a subclone were compared with all other single cells in the same sample using an alternative mode of scNOVA based on partial least squares-discriminant analysis. The inferred cell type was considered as a confounding factor in the differential analysis.

### Haplotype-specific nucleosome occupancy analysis
First, the chromosome-wide haplotype of nucleosome occupancy at gene bodies was resolved. The nucleosome occupancy of two haplotypes for each gene were compared using two-tailed Wilcoxon's test followed by a Benjamini–Hochberg multiple correction. Using 10% FDR cutoff, genes showing haplotype-specific nucleosome occupancy were identified.

### CITE-seq data pre-processing and integration
Cell Ranger v.6.0 (10x Genomics) was used to align the sequencing reads to the GRCh38 human reference genome build, distinguish cells

from the background and generate a unified feature-barcode matrix that contains gene expression counts, alongside cell-surface protein feature counts for each cell barcode.

**Quality control of CITE-seq data.** The R package Seurat v.4.0.4 was used to calculate the quality control metrics[50]. Cells were removed from the analysis if <200 or >8,000 distinct genes, <1,000 counts or >15% of reads mapping to mitochondrial genes were detected.

**Pre-processing and dimensional reduction of CITE-seq data.** Pre-processing and dimensional reduction of CITE-seq data were performed independently on both RNA and ADT assays. Gene counts were normalized by applying regularized negative binomial regression using the Seurat sctransform function[51], followed by principal component analysis (PCA) with highly variable genes as input. Cell-surface protein counts were centered log-ratio transformed across cells using the Seurat NormalizeData function with 'CLR' method, followed by scaling and PCA.

**Weighted nearest neighbor analysis of CITE-seq data.** For each cell, its closest neighbors in the dataset were calculated based on a weighted combination of RNA and protein similarities, using the Seurat FindMultiModalNeighbors function[52]. For the RNA modality, 30 dimensions were used and, for the protein modality, 18 dimensions. Downstream analysis including uniform manifold approximation and projection (UMAP) visualization and $t$-distributed stochastic neighbor embedding visualization of the data, as well as clustering, was performed based on a weighted combination of RNA and protein data. Clustering of the cells was done using the FindClusters function.

### scNOVA–CITE workflow

**Targeted SCNA recalling.** SCNA calling from the gene expression counts from CITE-seq data was done using the CONICSmat R package. In brief, to determine the copy-number status of each cell, CONICSmat fits a two-component Gaussian mixture model for each provided chromosomal region. The mixture model is fit to the average gene expression of genes within a region and cells with a deletion of the region will show an on-average lower expression from the region than cells without the deletion. The posterior probabilities for each cell belonging to one of the components can then be used to decipher the copy-number status of each cell[25].

In the present study, the structural variant discovery from scTRIP was used to construct a list of chromosomal regions containing SCNAs. These were used to infer the copy-number status of each cell for each chromosomal region using the $\log_2$(counts per million)/10 + 1) normalized gene counts from CITE-seq data. To be able to detect SCNAs affecting smaller regions, posterior probabilities were computed for regions with more than ten expressed genes (modified VisualizePosterior.R script; line 107 if(length(chr_genes)>10)). After obtaining the mixture model results, uninformative noisy regions were filtered based on the likelihood ratio test, adjusted $P$ ($P_{adj}$) < 0.01 and Bayes information criterion >200. A posterior probability cutoff of 0.8 was used for a confident SCNA assignment.

**Assignment of CITE-seq cells to genetic subclones.** SCNA regions from CONICSmat passing filtering were used as 'marker structural variants' matching subclone-specific structural variants identified using scTRIP. These marker structural variants were used to assign each cell to its corresponding genetic subclone. Cells not reaching confidence cutoff of 0.8 were termed 'unassigned' and excluded from downstream subclone-level analyses. For pre-LSCs in HIAML47, cells annotated as HSPCs and reaching confidence cutoff for the absence of marker structural variants were considered.

**'Reference-based' annotation of leukemic cells.** Single leukemic cells were assigned to their corresponding healthy counterparts using automatic cell-type annotation with SingleR[53] by determining similarity to reference bone marrow cells based on Spearman's correlation. A previously published CITE-seq dataset, which consists of 30,672 scRNA-seq profiles measured alongside a panel of 25 antibodies from bone marrow, was used as the reference bone marrow atlas[54].

**Finding differentially expressed features between subclones.** Marker genes that defined each structural variant group by differential expression were identified using the scran findMarkers function with two-sided Welch's $t$-test as the pairwise test. To account for the biases driven by different cell types in the structural variant groups, cell-type variable together with the structural variant group variable were used as predictors in the linear model via the design argument of findMarkers. Only upregulated marker genes were considered. Genes with an FDR-corrected $P \le 0.05$ and at least a 0.1-log(fold-change) in expression ($\log_1$(pFC) $\ge 0.1$) were considered as differentially expressed unless otherwise stated.

**Molecular phenotype analysis in gene sets.** AUCell[55] was used for signature score calculations between subclones with default parameters, using Hallmark modules from MSigDB[56]. LSC stemness scores were calculated for each cell as the mean expression of the normalized gene counts of the signature genes obtained by Ng and colleagues[57]. Gene-set over-representation analysis using enricher function from clusterProfiler was performed to model gene expression changes across the Hallmark modules from MSigDB[56]. For each gene set, the significance of overlap between the target gene set and genes exhibiting differential gene expression between subclones was computed using hypergeometric tests, followed by controlling the FDR at 0.05.

### Mouse experiments

NOD.*Prkdc*$^{scid}$.*Il2rg*$^{null}$ (NSG) mice were bred and housed under specific pathogen-free conditions in individually ventilated cages with controlled temperature (approximately 22 °C) and humidity (50%) under 12 h:12 h light:dark cycle at the central animal facility of the German Cancer Research Center (DKFZ). Animal experiments were conducted in compliance with all relevant ethical regulations. We obtained written, informed consent for all experiments and they were approved by the Regierungspräsidium Karlsruhe under Tierversuchsantrag nos. G42/18 and G-140-21.

**Xenotransplantations.** Female mice aged 8–12 weeks were sublethally irradiated (175 cGy) 24 h before xenotransplantation assays. AML samples were stained with human CD3 MicroBeads (Miltenyi Biotec, 130-050-101) for depletion of CD3$^+$ T cells. Magnetic-activated cell sorting (MACS) was performed according to manufacturer's instructions and unlabeled cells run through the MACS column were collected. Then, $1 \times 10^6$–$2 \times 10^6$ bulk, CD3-depleted AML cells were injected into the femoral bone marrow cavity of sublethally irradiated mice. Human leukemic engraftment in mouse bone marrow was evaluated by flow cytometry at 10 weeks, 16 weeks and endpoint (maximum 30 weeks unless endpoint criteria were reached earlier), using anti-human-CD45-AF700 (clone HI30; BD Biosciences, 560566), anti-human-CD34-BUV395 (clone 581; BD Biosciences, 563778), anti-human-CD38-BUV496 (clone HIT2; BD Biosciences, 612946), anti-human-GPR56-PE (clone CG4; BioLegend, 358204), anti-human-CD19-APC (clone HIB19; eBioscience, 17-0199-42), anti-human-CD33-APC (clone WM53; BioLegend, 740974) and anti-mouse-CD45-FITC (clone 30-F11; eBioscience, 11-0451-82). Mice were considered 'engrafted' if human cells represented >1% of the bone marrow cell population and 'leukemic/myeloid' if the human cells showed >80% CD33 positivity. At the endpoint, bone marrow cells were harvested from tibiae, femurs, iliac crests and spine by bone crushing. Spleen cells were harvested by mincing the spleen with a plunger. After red blood cell lysis, cells were resuspended in Cryostore (Sigma-Aldrich, C2874-100) and stored in liquid nitrogen until further use.

## Optical genome mapping

OGM was performed on primary HIAML85 sample and xenotransplantation samples from CK282 and CK397. Ultra-high molecular mass DNA was extracted from AML cells recovered from bone marrow or spleen following the manufacturer's protocols (Bionano Genomics). Briefly, the cells were digested followed by DNA precipitation and binding with a nanobind magnetic disk. Labeling of the ultra-high molecular mass DNA was performed following the manufacturer's instructions (Bionano Genomics), with 750 ng of DNA labeled using the standard direct labeling enzyme 1. The fluorescently labeled DNA molecules were imaged sequentially across nanochannels on a Saphyr instrument. A coverage of approximately 300× was achieved for all samples.

Somatic structural variants were analyzed using the Rare Variant Analyses software (Bionano Solve software) provided by Bionano Genomics. Molecules were aligned against the GRCh38 human reference genome build, without ploidy assumption. Consensus genome maps (*.cmaps) were assembled from clustered sets of molecules identifying the same variant, then realigned to GRCh38. Fractional SCNA analysis was performed from the alignment of molecules and labels against GRCh38 (alignmolvrefsv). A sample's raw label coverage was normalized against relative coverage from normal human controls, segmented and baseline copy-number state estimated by calculating the mode of coverage of all labels. Significant deviations from the baseline were used to assess the copy-number states, with high-variance regions masked.

## Multiplex FISH

M-FISH analysis was performed on xenotransplantation samples from CK282 and CK349. Cells were cultured the same as for Strand-seq analysis (see above) using previously established protocols[44,45]. M-FISH was performed as described previously[58]. In brief, seven pools of flow-sorted, whole-chromosome painting probes were amplified and combinatorially labeled by degenerative oligonucleotide-primed-PCR using DEAC-, FITC-, Cy3-, TexasRed- and Cy5-conjugated nucleotides and biotin-dUTP and digoxigenin-dUTP, respectively. Metaphase spreads were digested with pepsin (0.5 mg ml$^{-1}$; Sigma-Aldrich) in 0.2 N HCL (Roth), post-fixed in 1% formaldehyde, dehydrated with a degraded ethanol series and air dried, followed by denaturation of slides. Hybridization mixture was hybridized to the denatured metaphase preparations and incubated for 48 h at 37 °C. Three layers of antibodies were used to visualize biotinylated probes: streptavidin Alexa Fluor-750 conjugate (Invitrogen, S21384), biotinylated goat anti-avidin (Vector, BA-0300), followed by a second streptavidin Alexa Fluor-750 conjugate (Invitrogen, S21384). Two layers of antibodies were used to visualize digoxigenin-labeled probes: rabbit anti-digoxin (Sigma-Aldrich, D7782) followed by goat anti-rabbit immunoglobulin G Cy5.5 (Linaris, PAK0027). Slides were counterstained with DAPI and covered with antifade solution. A DMRXA epifluorescence microscope (Leica Microsystems) equipped with a Sensys CCD camera (Photometrics) was used to capture images of metaphase spreads for each fluorochrome using highly specific filter sets (Chroma Technology). Leica Q-FISH software was used to control the camera and microscope. Leica MCK software was used to process the images that were presented as multicolor karyograms (Leica Microsystems Imaging solutions).

## Fusion transcript detection from bulk RNA-seq

STAR-aligner-based Arriba fusion detection tool[59] was used to detect fusion transcripts from bulk RNA-seq data. First, reads were demultiplexed and STAR aligner 2.5.3a was used to align FASTQ files containing reads for individual samples by two-pass alignment[60]. Reads were aligned to a STAR index generated using the GRCh38 genome build. Detection of chimeric reads was enabled. Next, the Arriba fusion detection tool was used to extract the Chimeric.out.sam and Aligned.out.bam files and to create a list of fusion predictions passing Arriba's filters.

## Ex vivo drug screening

Ex vivo drug screening was performed on thawed cells from four diagnosis samples and human CD45$^+$ cells from two PDX samples (Supplementary Table 1). Cells were cultured the same as for Strand-seq analysis (see above) using previously established protocols[44,45]. Then, $0.5 \times 10^5$ AML cells per well were seeded in flat-bottomed, 96-well plates and cells were treated with up to 12 treatment conditions consisting of standard chemotherapy regimens as well as new compounds for 24 h, and for selected conditions for another 72 h (Supplementary Table 9). After 24 or 72 h, the cells were stained with cell-surface antibodies (Supplementary Table 12). The same amount of CountBright Absolute Counting Beads (Thermo Fisher Scientific, C36950) together with 7-aminoactinomycin D (BD Biosciences, 559925) was added to each sample before analysis with a BD LSRFortessa Cell Analyzer.

## Intracellular staining for BCL-2 family members

Intracellular staining was performed on thawed cells from four diagnosis samples as previously described[41] (Supplementary Table 12). Thawed cells were stained with Zombie NIR Fixable Viability stain in phosphate-buffered saline (BioLegend, 423105), followed by cell-surface antibody staining (Supplementary Table 12). Stained cells were fixed and permeabilized using the Fixation/Permeabilization Solution Kit (BD Biosciences, 554714) according to the manufacturer's instructions. To enhance intracellular staining, a secondary permeabilization step using Permeabilization Buffer Plus (BD Biosciences, 561651) was performed. Fixed and permeabilized cells were stained for anti-human-BCL-2-AF647 (clone 124; Cell Signaling, 82655), anti-human-MCL-1-AF488 (clone D2W9E; Cell Signaling, 58326) and anti-human-BCL-xL-PE-Cy7 (clone 54H6; Cell Signaling, 81965) (Supplementary Table 12). Samples were analyzed using a BD LSRFortessa Cell Analyzer.

## Quantification and statistical analysis

Methods used for statistical analyses are detailed in the figure legends. All statistical analyses were done using R 4.0.0. Flow cytometry data analysis was done using FlowJo v.10.5.3.

## Reporting summary

Further information on research design is available in the Nature Portfolio Reporting Summary linked to this article.

## Data availability

Sequencing data from this study can be retrieved from the European Genome-phenome Archive (EGA) and ArrayExpress. Data from primary CK-AML cells and PDXs are available under the following accessions: Strand-seq and CITE-seq (EGA, EGAS00001007436); bulk RNA-seq (ArrayExpress, E-MTAB-14420). Human patient data stored at the EGA are managed by the EGA Data Access Committee, following their most current standards for patient-derived omics data. This ensures that the data remain nonidentifiable while being accessible to researchers, typically within 2 weeks of submitting a reasonable request to the committee. We also used publicly available databases as follows: human GRCh38 reference database (Ensembl: http://ftp.ensembl.org) and Molecular Signature Database (MSigDB: https://www.gsea-msigdb.org/gsea/msigdb).

## Code availability

The computational software used in the present study include scNOVA (https://github.com/jeongdo801/scNOVA), Mosaicatcher (https://github.com/friendsofstrandseq/mosaicatcher-pipeline), Strand-PhaseR (https://github.com/daewoooo/StrandPhaseR), CON-ICSmat (https://github.com/diazlab/CONICS), Delly2 (https://github.com/dellytools/delly), NO_based_HSPC_classifier (https://github.com/jeongdo801/NO_based_HSPC_classifier), PloidyAssignR (https://github.com/lysfyg/PloidyAssignR), BWA[47] (v.0.7.15), STAR[60] (v.2.7.9a and v.2.5.3a), SAMtools[61] (v.1.3.1), biobambam2 (ref. [48]) (v.2.0.76), Sambamba[62] (v.0.6.5), R[63] (v.4.0.0), DESeq2 (ref. [64]), Cell Ranger[65] (v.6.0),

Seurat[66] (v.4.3.0.1), scran[67] (1.28.2), AUCell[55] (v.1.2.2.0), SingleR[53] (2.2.0), Arriba[59] (v.1.2.0), FlowJo (v.10.5.3), GraphPad Prism (v.9.3.1), Bionano Solve (v.3.7), Bionano Access (v.1.7.1) and BD FACSDiva. Analysis notebooks for the figures are available at https://github.com/amleppa/scNOVA-CITE_paper.

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

## Acknowledgements

We thank all technicians of A.T.'s laboratory for technical assistance and K. Stumpf, A. Narr and other lab members for constructive discussions. We are grateful to J.-P. Mallm and K. Bauer from the DKFZ Single-cell Open Lab, S. Schmitt, M. Eich, K. Hexel, T. Rubner and F. Blum from the DKFZ Flow Cytometry Core Facility for their assistance, and K. Reifenberg, P. Prückl, M. Durst, A. Rathgeb and all animal caretakers of the DKFZ Central Animal Laboratory for excellent animal welfare and husbandry. We also thank the DKFZ Genomics and Proteomics Core Facility for their assistance, as well as the DKFZ ODCF System Administration, and the European Molecular Biology Laboratory (EMBL) Flow Cytometry Core Facility for assistance in cell sorting and the EMBL Genomics Core Facility for assisting in Strand-seq single-cell automation. This work was partly supported by: the SPP2036, FOR2674 and SFB873 funded by the Deutsche Forschungsgemeinschaft (DFG); the DKTK joint funding project 'RiskY-AML'; the 'Integrate-TN' Consortium funded by the Deutsche Krebshilfe; the European Research Council (ERC) Advanced Grant SHATTER-AML (grant AdG-101055270); the ERC Consolidator Grant MOSAIC (grant CoG-773026); the Dietmar Hopp Foundation; and the National Institutes of Health (grants R01CA262496, R01CA284595-01 and R01CA283574-01). A.M.L. was supported by the Ida Montin Foundation. A.W. was supported by the European Molecular Biology Organization Postdoctoral Fellowship and the Marie Curie Individual Fellowship. B.R.M. was supported by a Bridging Excellence Fellowship provided by the Life Science Alliance. Graphic illustrations were created with BioRender.com.

## Author contributions

A.M.L., K.G., H.J., A.D.S., J.O.K. and A.T. conceptualized the study. A.M.L., K.G., F.Y.H., E.V.B. and P.H. performed Strand-seq experiments. K.G., A.M.L., H.J., F.Y.H., A.D.S. and J.O.K. performed structural variant analysis. K.G., A.M.L., F.Y.H. and J.O.K. performed subclonal reconstruction as well as measurement of intrapatient karyotype heterogeneity using Strand-seq data. H.J. and A.A. performed haplotype-specific nucleosome occupancy analysis and cell-type classification. A.M.L. and F.Y.H. performed CITE-seq experiments. A.M.L., F.Y.H. and F.G. performed alignment of CITE-seq data. A.M.L. carried out the analysis of CITE-seq data. A.M.L., A.W. and M.S. performed in vivo transplantation experiments. A.M.L., F.Y.H. and A.W. performed ex vivo drug-screening experiments. A.M.L. and A.J. performed M-FISH experiments. A.M.L. carried out OGM experiments. B.R.M. contributed to data analysis. A.W., T.B., D.K., V.T., A.D. and L.B. contributed to patient sample and PDX processing. A.K., S.R., P.S., C.M.T., A.K.E. and K.M. provided samples and clinical information. A.M.L., J.O.K. and A.T. wrote the manuscript with support from K.G. and A.D.S. and additional contributions from all authors.

## Funding

## Competing interests

A.D.S. and J.O.K. have previously disclosed a patent application (no. EP19169090) that is relevant to this manuscript. A.K.E. received an honorarium from AstraZeneca for serving on their diversity, equity and inclusion advisory board, and her spouse has ownership interest and is employed by Karyopharm Therapeutics. The remaining authors declare no competing interests.

## Additional information

**Extended data** is available for this paper at https://doi.org/10.1038/s41588-024-01999-x.

**Correspondence and requests for materials** should be addressed to Jan O. Korbel or Andreas Trumpp.

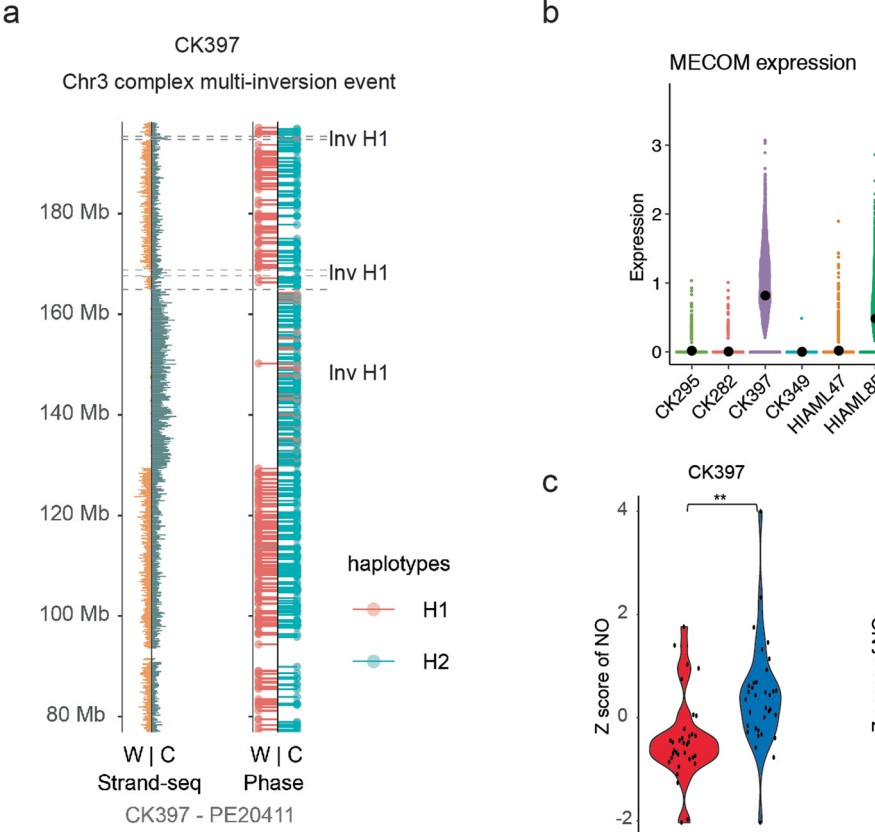

**Extended Data Fig. 1 | Chromosomal rearrangements at 3q and *MECOM* deregulation. a** Complex multi-inversion event in CK397 at chromosome 3. Shown is strand-specific read depth (left) separated into the phase data channel (right) of a representative CK397 cell. Reads denoting somatic structural variants, discovered using scTRIP, mapped to the Watson (W; orange) or Crick (C; green) strand. Reads overlapping single nucleotide polymorphisms were assigned to haplotype H1 (red lollipops) or H2 (blue lollipops). Grey: single cell IDs. **b** Expression of *MECOM* in single cells in primary CK-AML patient samples. Beeswarm plots show the 95% confidence interval for the mean. **c** Violin plot showing haplotype-specific nucleosome occupancy (NO) at the *MECOM* gene body (10% FDR) for HIAML85 and CK397. Nucleosome occupancy was assessed from all informative cells in which reads could be phased (WC or CW configuration) (*n* = 26 and 34 cells, respectively). H1 contains the inversion resulting in *RPN1-MECOM* rearrangement whereas H2 is normal at *MECOM* locus. Gene-body nucleosome occupancy measurements from both haplotypes were converted into log$_2$-scale and compared using two-tailed Wilcoxon test. Chr: Chromosome, Inv: Inversion.

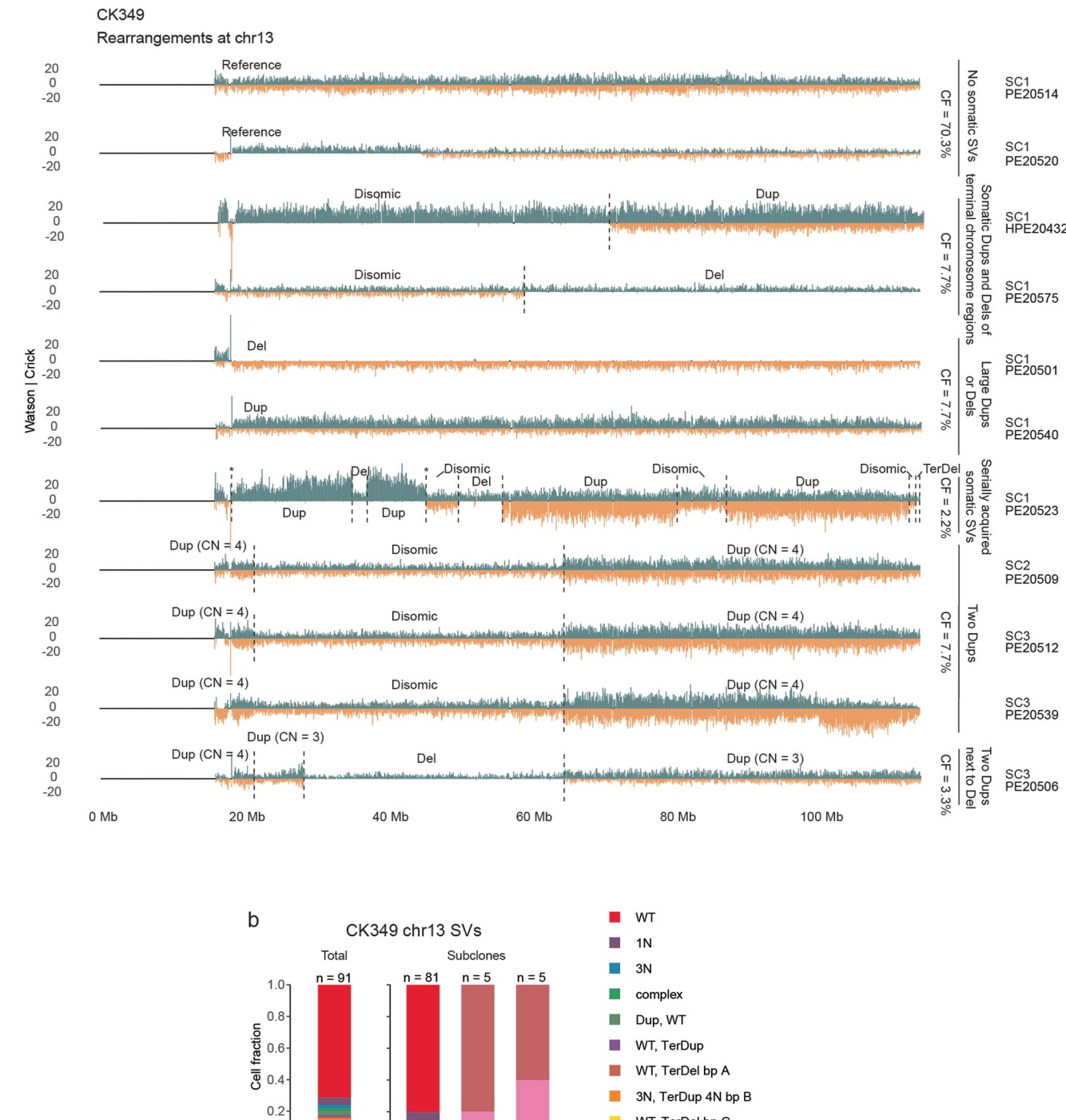

**Extended Data Fig. 2 | Genomic rearrangements at chromosome 13 in CK349.**
**a** Strand-specific read depth of representative single cells from CK349 showing different rearrangements detected at chromosome 13 in different subclones. Reads denoting somatic structural variants, discovered using scTRIP, mapped to the Watson (orange) or Crick (green) strand. **b** Stacked barplot showing the cell fraction of different rearrangements detected at chromosome 13 in CK349 at diagnosis. The number of cells is indicated on top of the bar and the distinct rearrangements are labelled below. CF: Cell fraction, SV: Structural variant, Chr: Chromosome, Dup: Duplication, Del: Deletion, Ter: Terminal, CN: Copy number, WT: Wild-type, bp: Break point.

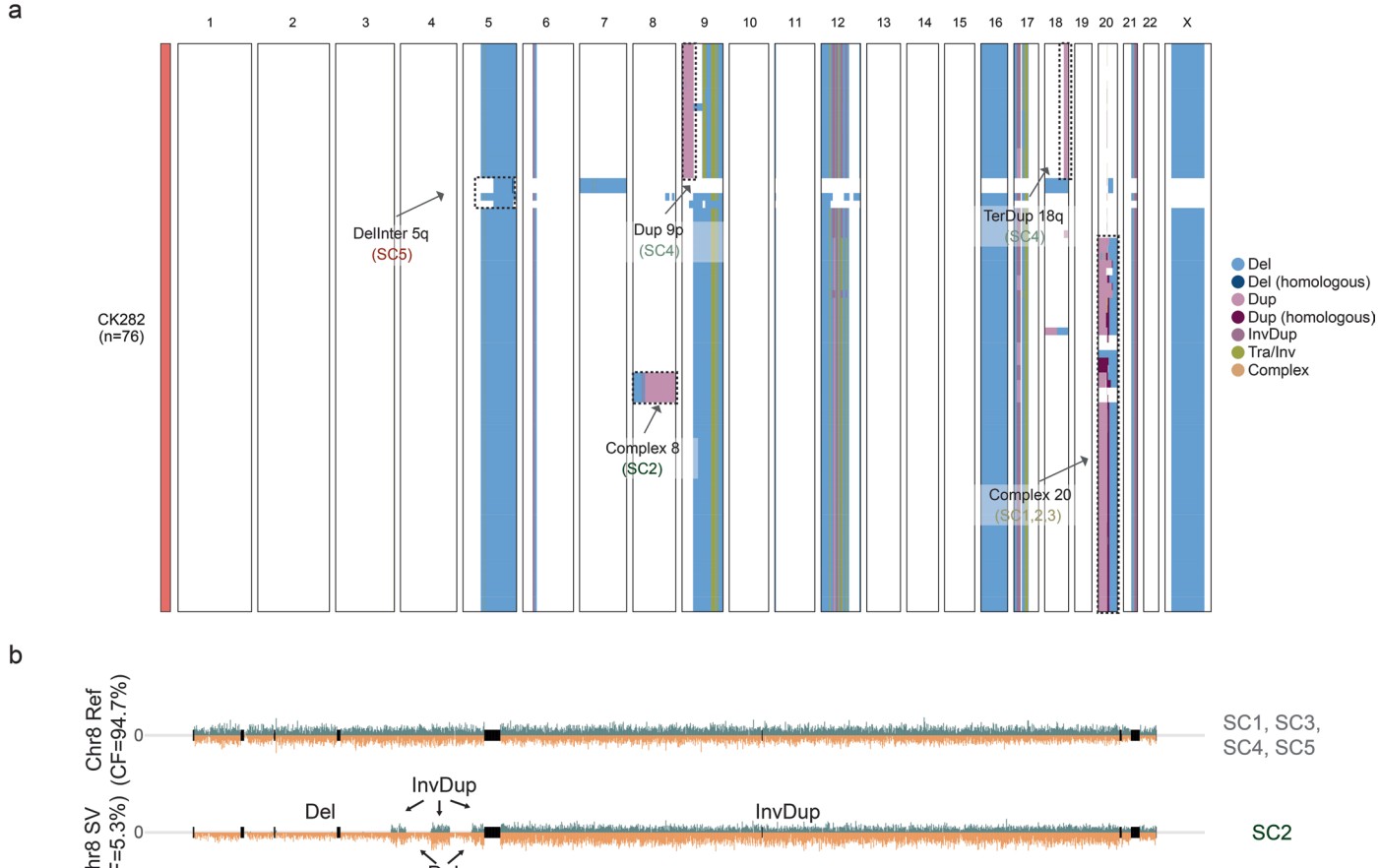

**Extended Data Fig. 3 | Subclonal heterogeneity in CK282. a** Karyotype heatmap of 76 single cells arranged using Ward's method for hierarchical clustering of structural variant genotypes in CK282. Examples of subclone-specific structural variants are labelled in the heatmap. **b** Strand-specific read depth of two representative single cells from CK282 showing a normal chromosome 8 (reference, top) and a complex genetic rearrangement comprising of two inverted duplications (InvDups), three deletions (Dels) and one larger InvDup, spanning the whole chromosome 8 (bottom). Reads denoting somatic structural variants, discovered using scTRIP, mapped to the Watson (orange) or Crick (green) strand. Del: Deletion, Dup: Duplication, Inv: Inversion, Tra: Translocation, Inter: Interstitial, Ter: Terminal, Chr: Chromosome, CF: Cell fraction, SV: Structural variant.

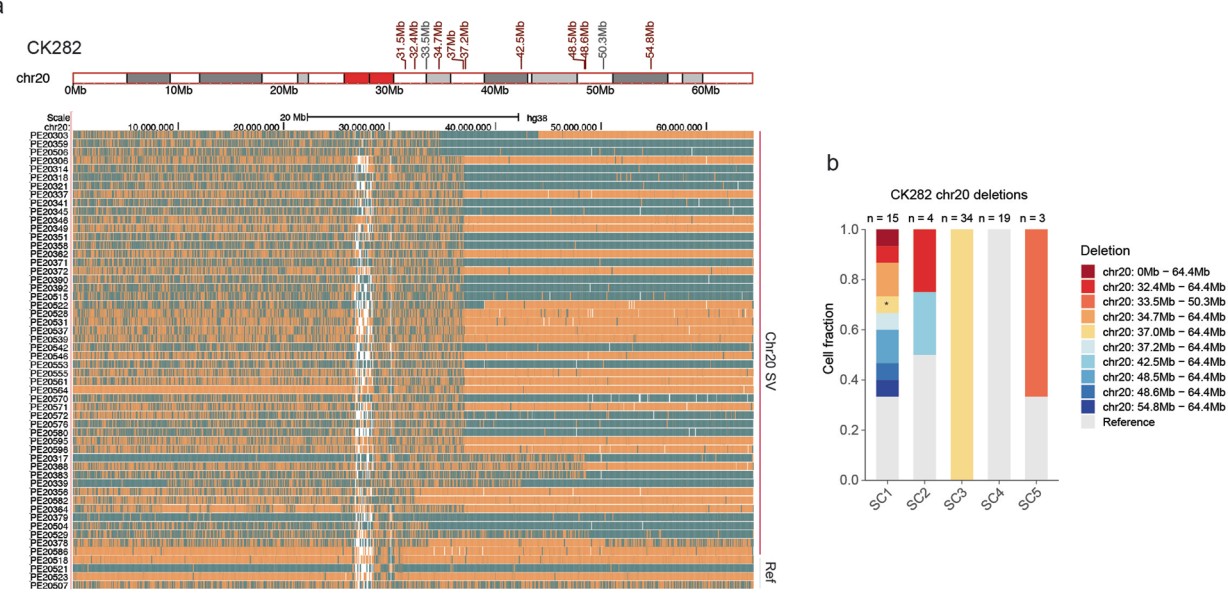

Extended Data Fig. 4 | See next page for caption.

**Extended Data Fig. 4 | Active mutational processes in CK282 and CK349. a**
Signs of active mutational processes at chromosome 20 in CK282 displayed by
varying breakpoints of the terminal deletion at 20q in representative cells. Reads
mapped to the Watson (orange) or Crick (green) strand. The terminal deletion
breakpoints are annotated above the ideogram in red and interstitial deletion
breakpoints in grey. **b** Stacked barplot showing the cell fraction of different
structural variants detected at chromosome 20 in the different subclones in
CK282 at diagnosis. The number of cells in each subclone is indicated on top of
the bar and the type of structural variant with the corresponding breakpoint(s)
labelled on the right. (*, additional complex rearrangement at 20p). **c** Strand-
specific read depth of representative single cells from CK349 showing signs of
active mutational processes at chromosome 17. **d** Stacked barplot showing the
cell fraction of different terminal deletions detected at chromosome 17 in the
different subclones in CK349 at diagnosis. The number of cells in each subclone
is indicated on top of the bar and the terminal deletion with the corresponding
breakpoint labelled on the right. Chr: Chromosome, SV: Structural variant, Del:
Deletion, Inv: Inversion.

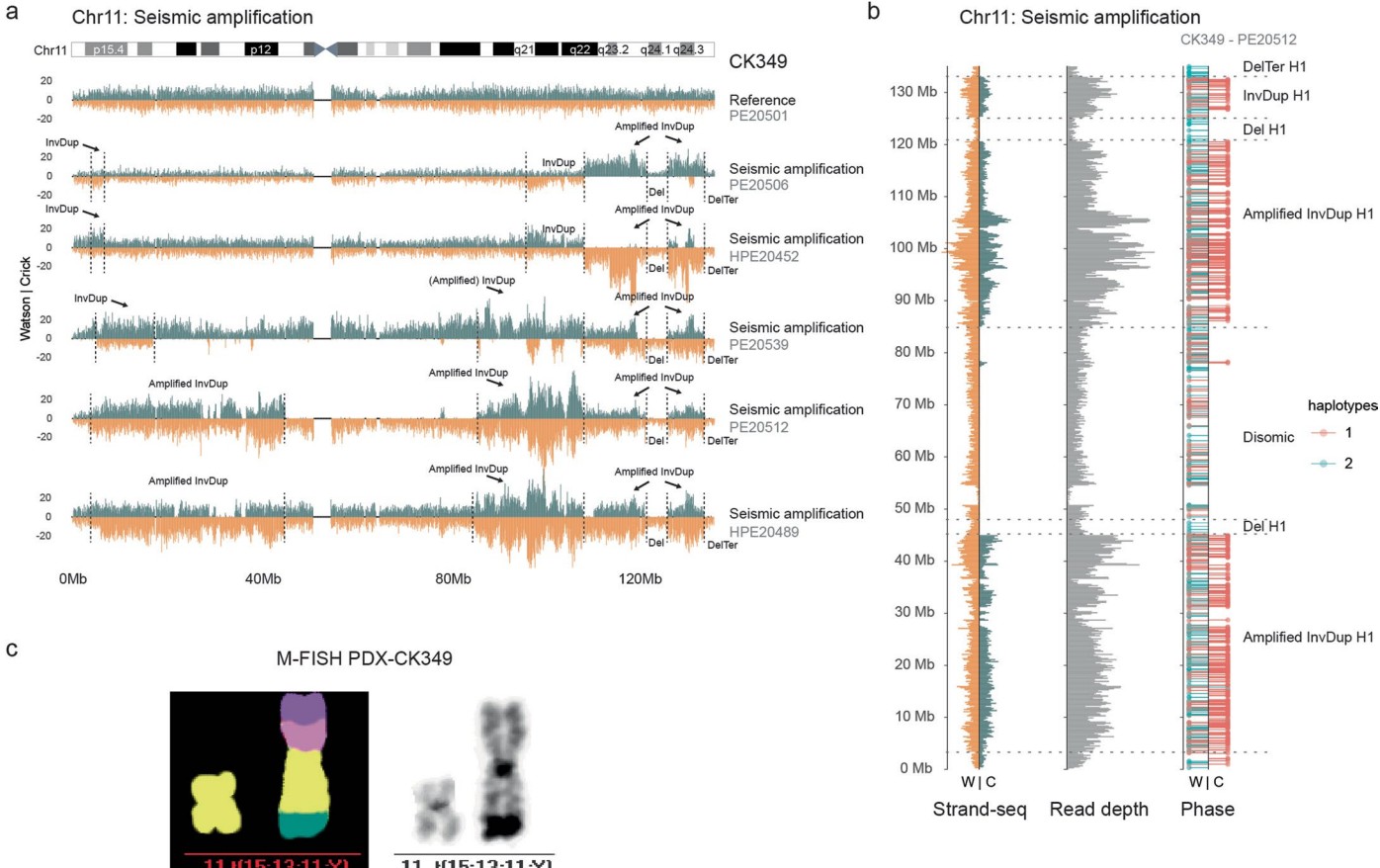

**Extended Data Fig. 5 | Seismic amplification at chromosome 11 in CK349.**
**a** Strand-specific read depth of all single cells from CK349 showing differing
amplification signals at chromosome 11 representing seismic amplifications,
and a representative cell with a normal chromosome 11 (top, major clone). Reads
denoting somatic structural variants, discovered using scTRIP, mapped to the
Watson (W; orange) or Crick (C; green) strand. Grey: single cell IDs. **b** Strand-
specific read depth of seismic amplification (left) separated into read depth and
phase (right) of a representative CK349 cell. Reads overlapping single nucleotide
polymorphisms were assigned to haplotypes H1 (red lollipops) or
H2 (blue lollipops). Grey: single cell ID. **c** Multiplex fluorescence *in situ*
hybridization (M-FISH) of a cell with normal chromosome 11 and a linearized
marker chromosome containing segments from chromosome 15, 13, 11 and
Y obtained from the secondary patient-derived xenograft (PDX) of CK349. Chr:
Chromosome, InvDup: Inverted Duplication, Del: Deletion, Ter: Terminal,
t: Translocation.

a

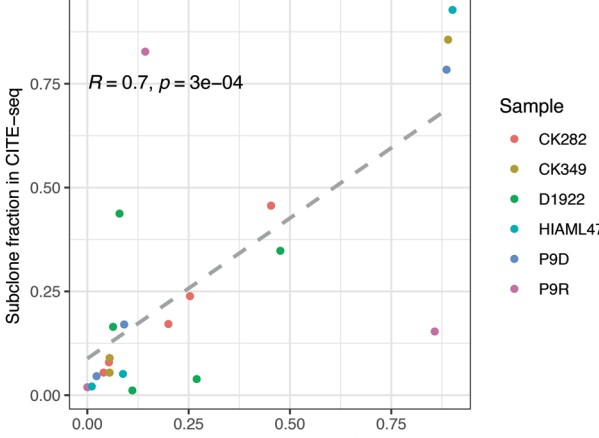

c

Strand–seq vs CITE–seq subclone fractions

Strand–seq and CITE–seq subclonal fraction show good accordance

b

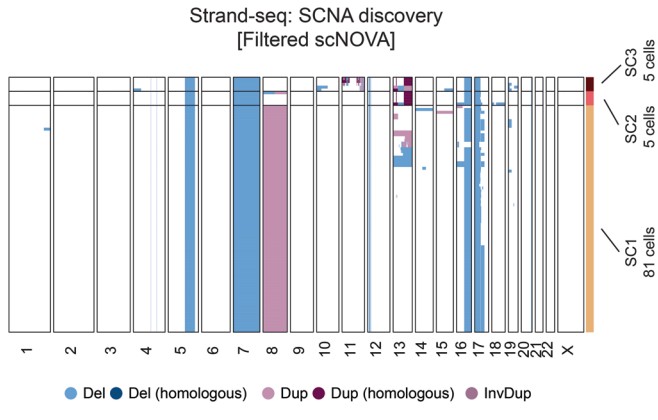

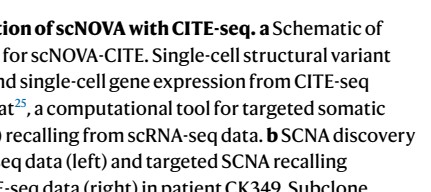

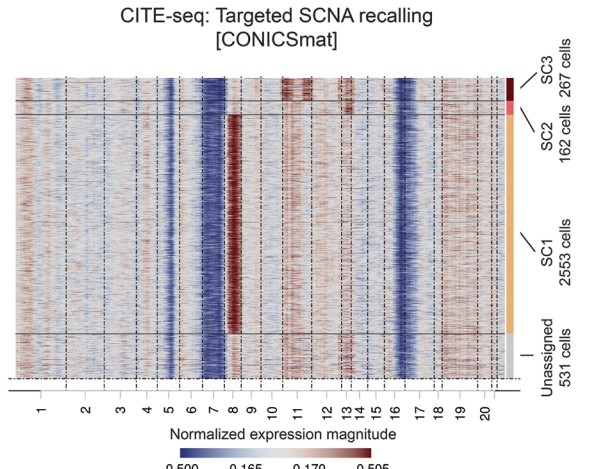

**Extended Data Fig. 6 | Integration of scNOVA with CITE-seq. a** Schematic of the data integration framework for scNOVA-CITE. Single-cell structural variant (SV) information from scTRIP and single-cell gene expression from CITE-seq was used as input for CONICSmat[25], a computational tool for targeted somatic copy-number alteration (SCNA) recalling from scRNA-seq data. **b** SCNA discovery based on scNOVA from Strand-seq data (left) and targeted SCNA recalling based on CONICSmat from CITE-seq data (right) in patient CK349. Subclone assignments and corresponding cell numbers are shown on the right of each heatmap. **c** Subclone fraction in Strand-seq data vs. subclone fraction in CITE-seq data. Each dot represents a subclone and the dashed line shows the linear fit. Correlation was calculated using two-tailed Spearman correlation. CNV: Copy number variation, Del: Deletion, Hom: Homologous, Dup: Duplication, InvDup: Inverted Duplication.

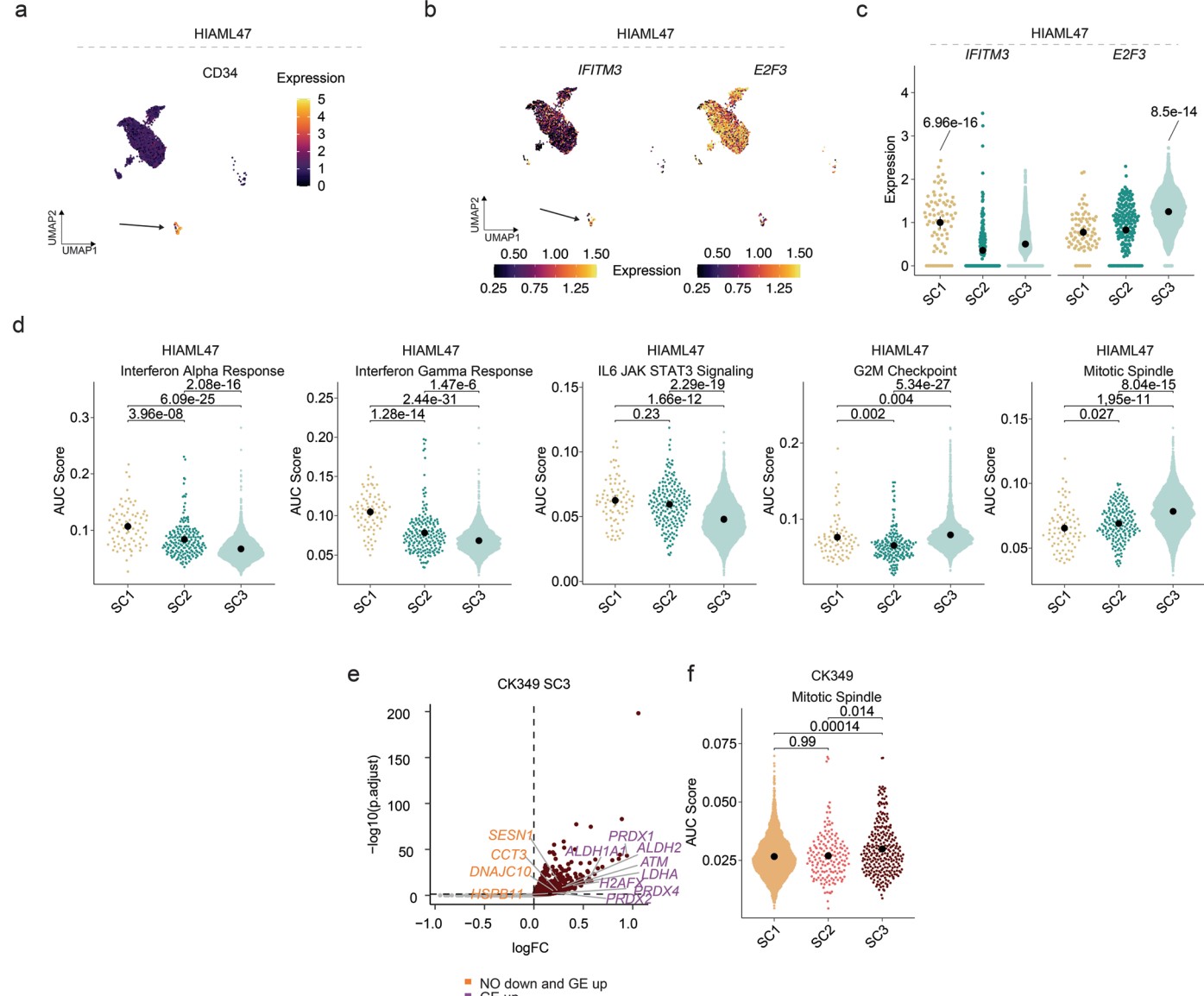

**Extended Data Fig. 7 | Molecular expression networks in HIAML47 and CK349. a** Cell surface expression of CD34 in single cells in HIAML47 plotted on the UMAP. Arrow indicates the pre-LSCs (SC1). **b** Expression of *IFITM3* and *E2F3* in single cells in HIAML47 plotted on the UMAP. Arrow indicates the pre-LSCs (SC1). **c** Expression of *IFITM3* and *E2F3* in the subclones in HIAML47 (*n* = 77 – 3,404 single cells). **d** Area Under the Curve (AUC) score for activity of indicated gene sets for each cell in the different subclones in HIAML47 (*n* = 77 – 3,404 single cells). **e** Upregulated genes in CK349-SC3. Orange labels highlight genes showing deregulation of cellular stress and DNA damage response based on nucleosome occupancy (NO) and gene expression (GE) and purple labels only based on gene expression. **f** AUC score for activity of Mitotic spindle gene set for each cell in the different subclones in CK349 (*n* = 162 – 2,553 single cells). In **c-d** and **f**, beeswarm plots show the 95% confidence interval for the mean, gene expression comparisons show the adjusted *P*-value from two-tailed pairwise Welch t-tests between subclones, and AUC scores were compared using two-tailed Wilcoxon test followed by Benjamini-Hochberg multiple correction testing. Expression levels of the individual genes in the score were calculated from normalized and variance stabilized counts.

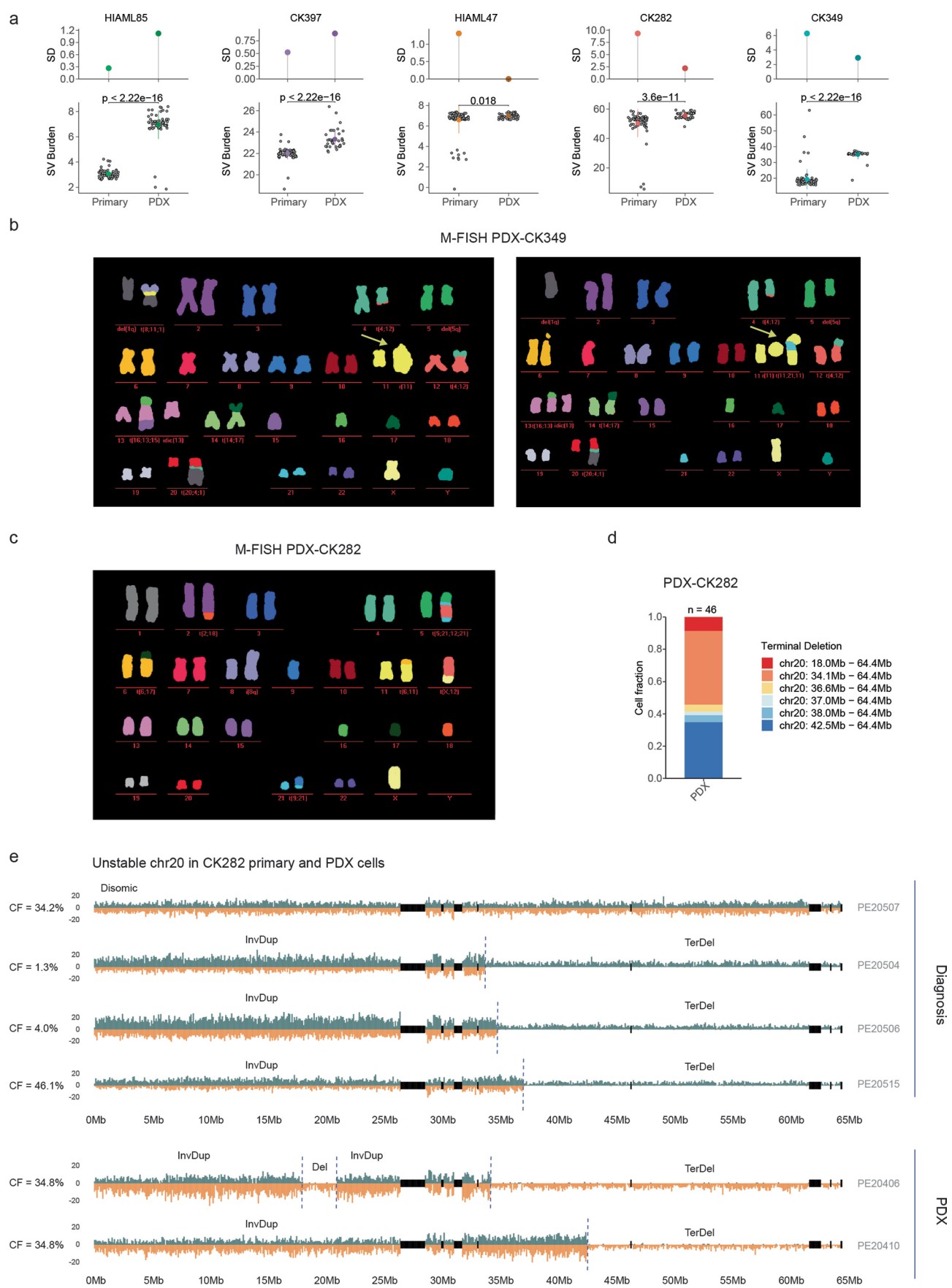

**Extended Data Fig. 8 | See next page for caption.**

**Extended Data Fig. 8 | Clonal evolution of CK-AML in patient-derived xenografts. a** Karyotype heterogeneity between primary and patient-derived xenograft (PDX) cells based on structural variant burden (bottom) and its standard deviation (top). Each grey dot represents a single cell. The structural variant burdens were compared using two-tailed Wilcoxon test (HIAML85: Primary ($n = 66$) and PDX ($n = 62$); HIAML47: Primary ($n = 91$), and PDX ($n = 54$); CK282: Primary ($n = 76$) and PDX ($n = 46$); CK349: Primary ($n = 91$) and PDX ($n = 40$); CK397: Primary ($n = 70$) and PDX ($n = 36$)); Point ranges was defined by minima = mean - 2X standard deviation, maxima = mean + 2X standard deviation, point = mean. **b** Multiplex fluorescence *in situ* hybridization (M-FISH) of two representative engrafted cells from the secondary PDX of CK349. Arrows indicate the ring and linearized marker chromosomes. **c** M-FISH of a representative engrafted cell from the PDX of CK282. **d** Stacked barplot showing the cell fraction of different terminal deletions detected at chromosome 20 in the PDX of CK282. The number of cells assessed is indicated on top of the bar and the genomic positions of the terminal deletions are shown on the right. **e** Strand-specific read depth of representative single cells from CK282 and PDX-CK282 showing different rearrangements detected at chromosome 20. Reads denoting somatic structural variants, discovered using scTRIP, mapped to the Watson (orange) or Crick (green) strand. SV: Structural variant, SD: Standard deviation, Chr: Chromosome, CF: Cell fraction, InvDup: Inverted Duplication, Del, Deletion, Ter: Terminal.

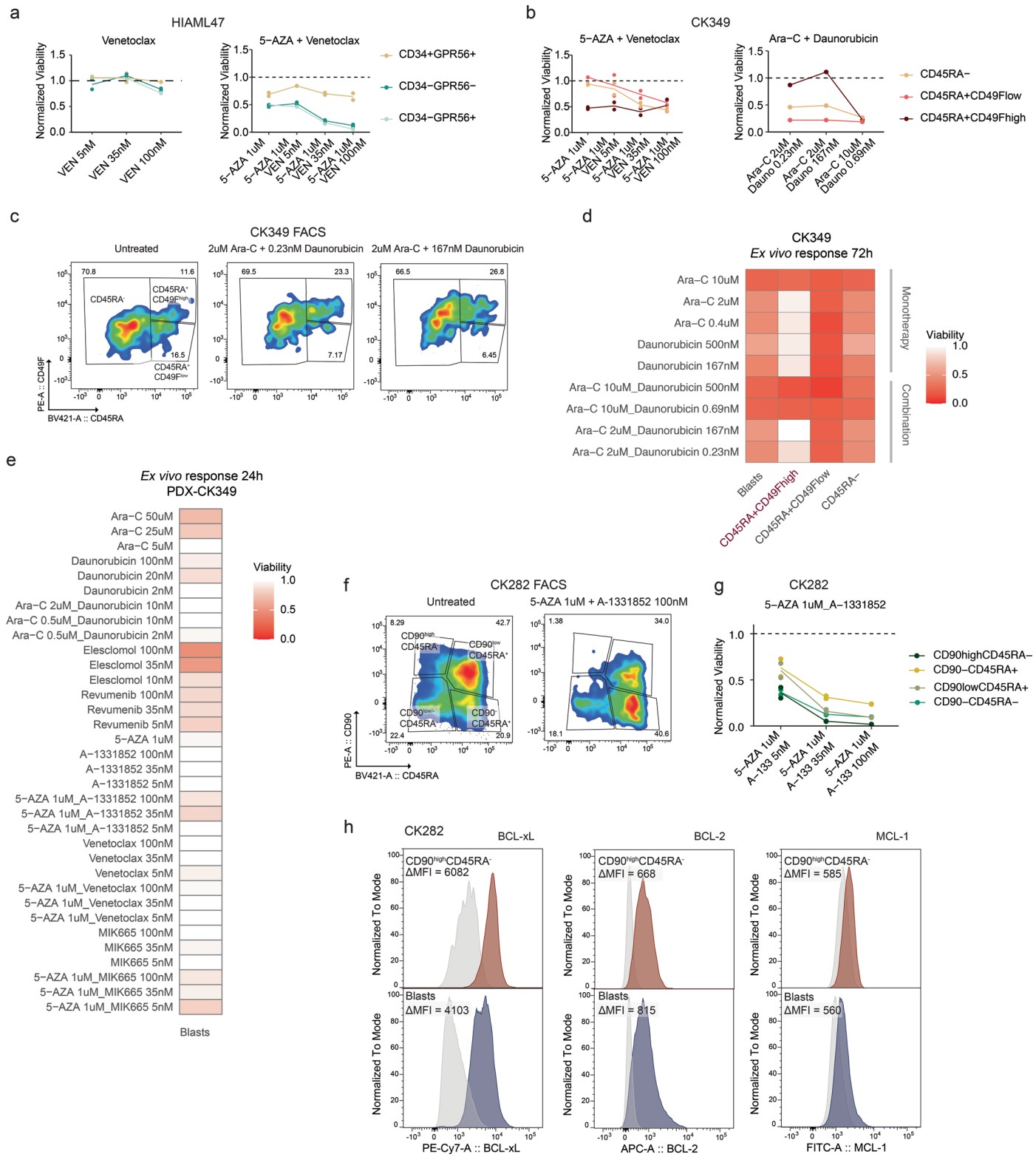

**Extended Data Fig. 9 | See next page for caption.**

**Extended Data Fig. 9 | *Ex vivo* drug screening in CK-AML. a** Viabilities of different populations in HIAML47 after 24 h *ex vivo* exposure with indicated concentrations of venetoclax (left) and venetoclax together with azacitidine (right). **b** Viabilities of different populations in CK349 after 24 h ex vivo exposure with indicated concentrations of azacitidine and venetoclax (left) and 72 h ex vivo exposure with indicated concentrations of cytarabine together with daunorubicin (right). **c** FACS plot displaying expression of CD45RA and CD49F on pre-gated leukemic cells in CK349. The gates highlight three populations with different CD45RA and CD49F expressions. Cells from untreated (left) and cytarabine (2 uM, middle and right) together with daunorubicin-treated (0.23 nM, middle, and 167 nM, right) conditions are shown after 72 h ex vivo exposure. **d** Viabilities of different populations after 72 h ex vivo exposure with indicated concentrations of cytarabine and daunorubicin in CK349. Engraftment-driving population is highlighted in red. **e** Viabilities of human blasts after 24 h ex vivo exposure with indicated concentrations of 12 treatment conditions in the patient-derived xenograft (PDX) of CK349. Shown are the mean viabilities of two replicates. **f** FACS plot displaying expression of CD45RA and CD90 on pre-gated leukemic cells in CK282. The gates highlight four populations with different CD45RA and CD90 expressions. Cells from untreated (left) and BCL-xL inhibitor-treated (A-1331852, 100 nM) together with hypomethylating agent (5-AZA, 1 uM, right) conditions are shown after 24 h ex vivo exposure. **g** Viabilities of different populations in CK282 after 24 h ex vivo exposure with indicated concentrations of venetoclax together with azacitidine. **h** Fluorescence intensity of BCL-xL (left), BCL-2 (middle) and MCL-1 (right) protein expression in CD90$^{high}$CD45RA$^-$ cells (red) compared to all blasts (blue) in CK282. Delta mean fluorescence intensity (MFI) shown at the top of the plots was calculated as the difference in MFI between the specific protein expression (colored histogram) and its IgG control (grey histogram) in the assessed population. Ex vivo viabilities were calculated as the fraction of viable cells compared to untreated control. VEN: Venetoclax, 5-AZA: Azacitidine, Ara-C: Cytarabine, Dauno: Daunorubicin.

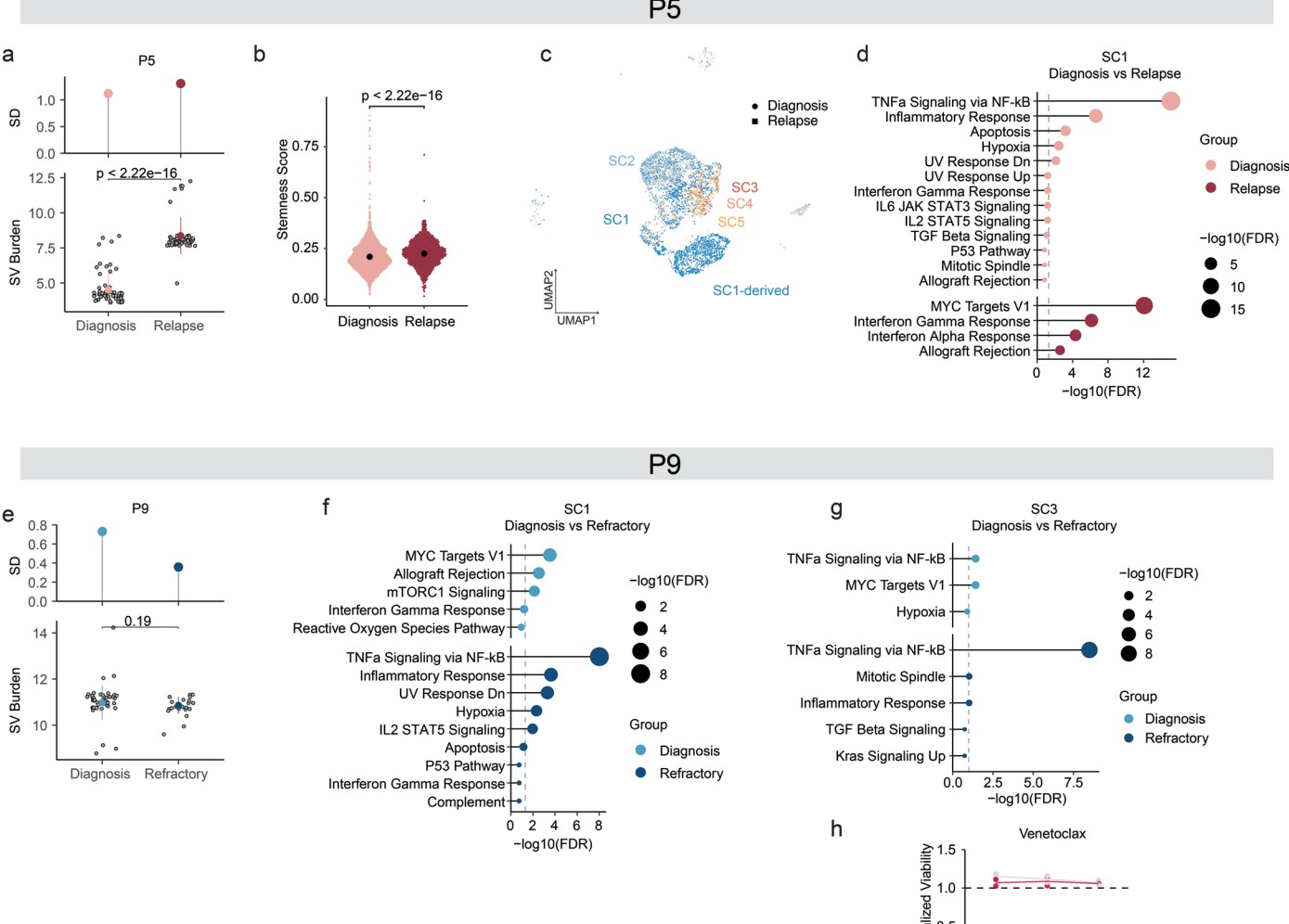

**Extended Data Fig. 10 | Longitudinal evolution of CK-AML under therapy stress. a** Karyotype heterogeneity between diagnosis and relapse cells in patient P5 based on structural variant (SV) burden (bottom) and its standard deviation (SD; top). Each grey dot represents a single cell. The structural variant burdens were compared using two-tailed Wilcoxon test (Diagnosis ($n$ = 63), Relapse ($n$ = 54)); Point ranges was defined by minima = mean - 2X standard deviation, maxima = mean + 2X standard deviation, point = mean. **b** Expression of the Ng et al. LSC Up transcriptomic stemness scores[27] in the single cells at diagnosis vs. relapse in patient P5 ($n_{Diagnosis}$ = 3,444 and $n_{Relapse}$ = 1,102). Stemness scores between disease stages were compared using two-tailed Wilcoxon test. Expression levels of the individual genes in the score were calculated from normalized and variance stabilized counts. Beeswarm plots show the 95% confidence interval for the mean. **c** Weighted nearest neighbor-based UMAP plots of diagnosis and relapse leukemic cells from patient P5 CITE-seq data. Cells are colored based on subclones identified using scTRIP and are shaped based on disease stage. **d** Enriched pathways at diagnosis and relapse in SC1-derived cells in patient P5. Genes with FDR < 0.05 and log-fold-change > 0.25 were included in the analysis. **e** Karyotype heterogeneity between diagnosis and refractory cells in patient P9 based on structural variant burden (bottom) and its standard deviation (top). Each grey dot represents a single cell. The structural variant burdens were compared using two-tailed Wilcoxon test (Diagnosis ($n$ = 44), Refractory ($n$ = 21)); Point ranges was defined by minima = mean - 2X standard deviation, maxima = mean + 2X standard deviation, point = mean. **f** Enriched pathways at diagnosis and refractory disease in SC1-derived cells in patient P9. Genes with FDR < 0.05 and log-fold-change > 0.25 were included in the analysis. **g** Enriched pathways at diagnosis and refractory disease in SC3-derived cells in patient P9. Genes with FDR < 0.05 and log-fold-change > 0.25 were included in the analysis. **h** Viabilities (fraction of viable cells compared to untreated control) of different populations after 24 h ex vivo exposure with indicated concentrations of venetoclax (VEN) in P9 diagnosis cells (P9D).

Jan Korbel

# Reporting Summary

## Statistics

For all statistical analyses, confirm that the following items are present in the figure legend, table legend, main text, or Methods section.

| n/a | Confirmed | |
|---|---|---|
| ☐ | ☒ | The exact sample size (*n*) for each experimental group/condition, given as a discrete number and unit of measurement |
| ☐ | ☒ | A statement on whether measurements were taken from distinct samples or whether the same sample was measured repeatedly |
| ☐ | ☒ | The statistical test(s) used AND whether they are one- or two-sided *Only common tests should be described solely by name; describe more complex techniques in the Methods section.* |
| ☒ | ☐ | A description of all covariates tested |
| ☐ | ☒ | A description of any assumptions or corrections, such as tests of normality and adjustment for multiple comparisons |
| ☐ | ☒ | A full description of the statistical parameters including central tendency (e.g. means) or other basic estimates (e.g. regression coefficient) AND variation (e.g. standard deviation) or associated estimates of uncertainty (e.g. confidence intervals) |
| ☐ | ☒ | For null hypothesis testing, the test statistic (e.g. *F*, *t*, *r*) with confidence intervals, effect sizes, degrees of freedom and *P* value noted *Give P values as exact values whenever suitable.* |
| ☒ | ☐ | For Bayesian analysis, information on the choice of priors and Markov chain Monte Carlo settings |
| ☒ | ☐ | For hierarchical and complex designs, identification of the appropriate level for tests and full reporting of outcomes |
| ☐ | ☒ | Estimates of effect sizes (e.g. Cohen's *d*, Pearson's *r*), indicating how they were calculated |

*Our web collection on statistics for biologists contains articles on many of the points above.*

## Software and code

Policy information about availability of computer code

Data collection | Our study uses data output by illumina sequencers and bionano saphyr optical genome mapping system and hence no special software was used for collecting it.

Data analysis | The computational software used in the study include scNOVA (https://github.com/jeongdo801/scNOVA), Mosaicatcher (https://github.com/friendsofstrandseq/mosaicatcher-pipeline), Strand-PhaseR (https://github.com/daewoooo/StrandPhaseR), CONICSmat (https://github.com/diazlab/CONICS), Delly2 (https://github.com/dellytools/delly), NO_based_HSPC_classifier (https://github.com/jeongdo801/NO_based_HSPC_classifier), PloidyAssignR (https://github.com/lysfyg/PloidyAssignR), BWA (v.0.7.15), STAR (v.2.7.9a and v.2.5.3a), SAMtools (v.1.3.1), biobambam2 (v.2.0.76), Sambamba (v.0.6.5), R (v.4.0.0), DESeq2, Cell Ranger (v.6.0), Seurat (v.4.3.0.1), scran (1.28.2), AUCell (v.1.2.2.0), SingleR (2.2.0), Arriba (v.1.2.0), FlowJo (v.10.5.3), Prism (v.9.3.1), Bionano Solve software (v3.7), Bionano Access Software (v1.7.1), and BD FACSDiva. Analysis notebooks for the figures are available at: https://github.com/amleppa/scNOVA-CITE_paper.

For manuscripts utilizing custom algorithms or software that are central to the research but not yet described in published literature, software must be made available to editors and reviewers. We strongly encourage code deposition in a community repository (e.g. GitHub). See the Nature Portfolio guidelines for submitting code & software for further information.

## Data

Policy information about availability of data

All manuscripts must include a data availability statement. This statement should provide the following information, where applicable:

- Accession codes, unique identifiers, or web links for publicly available datasets
- A description of any restrictions on data availability
- For clinical datasets or third party data, please ensure that the statement adheres to our policy

Sequencing data from this study can be retrieved from the European Genome-phenome Archive (EGA) and ArrayExpress. Data from primary CK-AML cells and PDXs are available under the following accession numbers: Strand-seq and CITE-seq (EGAS00001007436); bulk RNA-seq (E-MTAB-14420). Access to human patient data deposited at EGA is governed by the EGA Data Access Committee. We also used publicly available databases as follows: human GRCh38 reference database (ENSEMBL; http://ftp.ensembl.org/) and Molecular signature database (MSigDB; https://www.gsea-msigdb.org/gsea/msigdb/).

## Research involving human participants, their data, or biological material

Policy information about studies with human participants or human data. See also policy information about sex, gender (identity/presentation), and sexual orientation and race, ethnicity and racism.

| | |
|---|---|
| Reporting on sex and gender | Patient sex was reported in accordance with the sex identified on the national identification card. We analyzed 8 CK-AML samples, four with female sex and four with male sex. Sex was not considered in the study design since this study focuses on intra-sample comparison in a case-based manner rather than on performing statistical tests between groups of samples. |
| Reporting on race, ethnicity, or other socially relevant groupings | We analyzed samples from 8 CK-AML patients (samples obtained at diagnosis/salvage stage and relapse/refractory stage). Clinical covariates of CK-AML patients are given in Supplementary Table 1. |
| Population characteristics | We analyzed samples from 8 CK-AML patients (samples obtained at diagnosis/salvage stage and relapse/refractory stage). Clinical covariates of CK-AML patients are given in Supplementary Table 1. |
| Recruitment | All samples were obtained from patients that provided written informed consent for the research use of their specimens in agreement with the Declaration of Helsinki. The project was approved by the Ethics Committee/Institutional Review Board of the Medical Faculty of Heidelberg and Cancer and Leukemia Group B (GALGB) (NCT-MASTER Platform S-206/2011 and S-169/2017, and GALGB studies CALGB 8461, CALGB 9665 and CALGB 20202). The protocols involved collection of bone marrow (BM) aspirates and peripheral blood (PB) samples. Part of the cohort were provided by the NCT cell and liquid biobank, a member of the BMBH. Samples included in the study were chosen based on their clinical classification as complex karyotype and the availability of at least two viably-frozen sample vials. |
| Ethics oversight | The project was approved by the Ethics Committee/Institutional Review Board of the Medical Faculty of Heidelberg and Cancer and Leukemia Group B (GALGB) (NCT-MASTER Platform S-206/2011 and S-169/2017, and GALGB studies CALGB 8461, CALGB 9665 and CALGB 20202). |

Note that full information on the approval of the study protocol must also be provided in the manuscript.

# Field-specific reporting

Please select the one below that is the best fit for your research. If you are not sure, read the appropriate sections before making your selection.

☒ Life sciences     ☐ Behavioural & social sciences     ☐ Ecological, evolutionary & environmental sciences

For a reference copy of the document with all sections, see nature.com/documents/nr-reporting-summary-flat.pdf

# Life sciences study design

All studies must disclose on these points even when the disclosure is negative.

| | |
|---|---|
| Sample size | No sample-size calculation was performed since this study focuses on intra-sample comparison in a case-based manner rather than on performing statistical tests between groups of samples. The cohort size was determined by the number of CK-AML samples available. |
| Data exclusions | In Strand-seq data, we excluded low quality single-cell libraries that showed very low, uneven coverage, of an excess of 'background reads' yielding noisy single cell data prior to analysis. Cells with incomplete BrdU incorporation or cells undergoing more than one DNA synthesis phase under BrdU exposure are largely excluded during cell sorting and thus get only rarely sequenced during Strand-seq experiments. In CITE-seq data, we excluded cells from the analysis if fewer than 200 or more than 8,000 distinct genes, fewer than 1,000 counts or more than 15% of reads mapping to mitochondrial genes were detected. |
| Replication | We have repeated the analyses of our datasets a minimum of two times and can confirm consistent and reproducible results from the workflows used. To ensure reproducibility of our experimental findings, we generated replicates wherever possible confirming the reproducibility of the results. Use of replicates is indicated in the figure legends. All attempts of replication were successful. |

| Randomization | Does not apply (there are no experimental groups in our study). |
|---|---|
| Blinding | Does not apply (the study focuses on intra-sample comparison rather than performing statistical tests between groups of samples). |

# Reporting for specific materials, systems and methods

We require information from authors about some types of materials, experimental systems and methods used in many studies. Here, indicate whether each material, system or method listed is relevant to your study. If you are not sure if a list item applies to your research, read the appropriate section before selecting a response.

## Materials & experimental systems

| n/a | Involved in the study |
|---|---|
| ☐ | ☒ Antibodies |
| ☒ | ☐ Eukaryotic cell lines |
| ☒ | ☐ Palaeontology and archaeology |
| ☐ | ☒ Animals and other organisms |
| ☒ | ☐ Clinical data |
| ☒ | ☐ Dual use research of concern |
| ☒ | ☐ Plants |

## Methods

| n/a | Involved in the study |
|---|---|
| ☒ | ☐ ChIP-seq |
| ☐ | ☒ Flow cytometry |
| ☒ | ☐ MRI-based neuroimaging |

## Antibodies

| Antibodies used | FACS (clone; manufacturer; catalogue number; application; reactivity): |
|---|---|
| | FITC mouse anti-human mCD45 (clone 30-F11; eBioscience; # 11-0451-82; FC; Mouse) |
| | AF700 mouse anti-human mCD45 (clone 30-F11; BioLegend; # 103128; FC; Mouse) |
| | BV421 mouse anti-human CD45RA (clone HI100; BD Biosciences; # 562885; FC; Human) |
| | BV510 mouse anti-human CD3 (clone UCHT1; BioLegend; # 317332; FC; Human) |
| | BV510 mouse anti-human CD20 (clone 2H7; BioLegend; # 302340; FC; Human) |
| | BV510 mouse anti-human CD235a (clone GAR2; BD Biosciences; # 740174; FC; Human) |
| | AF700 mouse anti-human CD45 (clone HI30; BD Biosciences; # 560566; FC; Human) |
| | FITC mouse anti-human CD45 (clone HI30; BioLegend; # 304006; FC; Human) |
| | APC-Cy7 mouse anti-human CD45 (clone HI30; BioLegend; # 304014; FC; Human) |
| | PE mouse anti-human CD49F (clone GoH3; BioLegend; # 313611; FC; Human, Mouse, Cynomolgus, Rhesus) |
| | PE-Cy7 mouse anti-human GPR56 (clone CG4; BioLegend; # 358206; FC; Human, Mouse) |
| | BUV395 mouse anti-human CD34 (clone 581; BD Biosciences; # 563778; FC; Human) |
| | BUV496 mouse anti-human CD38 (clone HIT2; BD Biosciences; # 612946; FC; Human) |
| | FITC mouse anti-human CD99 (clone HCD99; BioLegend; # 318006; FC; Human) |
| | PE mouse anti-human CD90 (clone 5E10; BioLegend; # 328109; FC; Human) |
| | BV711 mouse anti-human CD13 (clone WM15; BioLegend; # 301721; FC; Human) |
| | BV785 mouse anti-human CD117 (clone 104D2; BioLegend; # 313238; FC; Human) |
| | PE mouse anti-human HLA-DR (clone L243; BioLegend; # 307605; FC; Human, Cynomolgus, Rhesus) |
| | PE-CF594 mouse anti-human CD123 (clone 7G3; BD Biosciences; # 562391; FC; Human) |
| | FITC mouse anti-human CD81 (clone 5A6; BioLegend; # 349504; FC; Human) |
| | APC mouse anti-human CD18 (clone TS1/18; BioLegend; # 302114; FC; Human) |
| | AF700 mouse anti-human CD54 (clone HA58; BioLegend; # 353126; FC; Human) |
| | BV711 mouse anti-human CD11b (clone ICRF44; BioLegend; # 301344; FC; Human, Cynomolgus, Rhesus) |
| | BV786 mouse anti-human CD33 (clone WM53; BD Biosciences; # 740974; FC; Human) |
| | FITC mouse anti-human CD9 (clone HI9a ; BioLegend; # 312104; FC; Human) |
| | APC mouse anti-human CD64 (clone 10.1 ; BioLegend; # 305014; FC; Human, Cynomolgus, Rhesus) |
| | BV711 mouse anti-human CD45 (clone HI30; BD Biosciences; # 564357; FC; Human) |
| | PE mouse anti-human CD11b (clone ICRF44; BioLegend; # 982606; FC; Human) |
| | AF488 mouse anti-human MCL-1 (clone D2W9E; Cell Signaling; # 58326; FC; Human, Mouse, Rat) |
| | AF647 mouse anti-human BCL-2 (clone 124; Cell Signaling; # 82655; FC; Human) |
| | PE-Cy7 mouse anti-human BCL-xL (clone 54H6; Cell Signaling; # 81965; FC; Human, Mouse, Rat, Monkey) |
| | Zombie NIR (clone -; BioLegend; # 423105; FC; All species) |
| | 7-AAD (clone -; BD Biosciences; # 559925; FC; All species) |
| | |
| | CITE-seq (clone; manufacturer; catalogue number; application; reactivity): |
| | Total-seq A mouse anti-human CD26 (clone BA5b; BioLegend; # 302720; PG; Human) |
| | Total-seq A mouse anti-human CD45 (clone HI30; BioLegend; # 304064; PG; Human) |
| | Total-seq A mouse anti-human TIM3 (clone F38-2E2; BioLegend; # 345047; PG; Human) |
| | Total-seq A mouse anti-human CD99 (clone 3B2/TA8; BioLegend; # 371317; PG; Human) |
| | Total-seq A mouse anti-human CD33 (clone P67.6; BioLegend; # 366629; PG; Human) |
| | Total-seq A mouse anti-human CD38 (clone HIT2; BioLegend; # 303541; PG; Human) |
| | Total-seq A mouse anti-human CD44 (clone IM7; BioLegend; # 103045; PG; Human, Mouse) |
| | Total-seq A mouse anti-human CD117 (clone 104D2; BioLegend; # 313241; PG; Human) |
| | Total-seq A mouse anti-human CD34 (clone 581; BioLegend; # 343537; PG; Human) |
| | Total-seq A mouse anti-human CD90 (clone 5E10; BioLegend; # 328135; PG; Human) |

Total-seq A mouse anti-human CD49F (clone GoH3; BioLegend; # 313633; PG; Human, Mouse, Cynomolgus, Rhesus)
Total-seq A mouse anti-human CD10 (clone HI10a; BioLegend; # 312231; PG; Human, Cynomolgus, Rhesus)
Total-seq A mouse anti-human CD135 (clone BV10AH2; BioLegend; # 313317; PG; Human)
Total-seq A mouse anti-human CD123 (clone 6H6; BioLegend; # 306037; PG; Human)
Total-seq A mouse anti-human CD371 (CLEC12A) (clone 50C1; BioLegend; # 353613; PG; Human)
Total-seq A mouse anti-human CD7 (clone CD7-6B7; BioLegend; # 343123; PG; Human)
Total-seq A mouse anti-human HLA-DR (clone L243; BioLegend; # 307659; PG; Human, Cynomolgus, Rhesus)
Total-seq A mouse anti-human GPR56 (clone CG4; BioLegend; # 358207; PG; Human, Mouse)
Total-seq A mouse anti-human CD45RA (clone HI100; BioLegend; # 304157; PG; Human)
Total-seq A mouse anti-human CD64 (clone 10.1; BioLegend; # 305037; PG; Human, Cynomolgus, Rhesus)
Total-seq A mouse anti-human CD11b (clone ICRF44; BioLegend; # 301353; PG; Human, Cynomolgus, Rhesus)
Total-seq A mouse anti-human CD3 (clone UCHT1; BioLegend; # 300475; PG; Human)
Total-seq A mouse anti-human CD4 (clone SK3; BioLegend; # 344649; PG; Human)
Total-seq A mouse anti-human CD8 (clone SK1; BioLegend; # 344751; PG; Human, Cynomolgus, Rhesus)
Total-seq A mouse anti-human CD25 (clone BC96; BioLegend; # 302643; PG; Human)
Total-seq A mouse anti-human CD19 (clone HIB19; BioLegend; # 302259; PG; Human)
Total-seq A mouse anti-human CD56 (clone 5.1H11; BioLegend; # 362557; PG; Human)
Total-seq A mouse anti-human CD16 (clone 3G8; BioLegend; # 302061; PG; Human, Cynomolgus, Rhesus)
Total-seq A mouse anti-human CD274 (PD-L1) (clone 29E.2A3; BioLegend; # 329743; PG; Human)
Total-seq A mouse anti-human CD223 (LAG-3) (clone 11C3C65; BioLegend; # 369333; PG; Human)
Total-seq A mouse anti-human CD152 (CTLA-4) (clone BNI3; BioLegend; # 369619; PG; Human)
Total-seq A mouse anti-human CD279 (PD-1) (clone EH12.2H7; BioLegend; # 329955; PG; Human)
Total-seq A mouse anti-human CD86 (clone IT2.2; BioLegend; # 305443; PG; Human, Cynomolgus, Rhesus)
Total-seq A mouse anti-human CD226 (DNAM-1) (clone 11A8; BioLegend; # 338335; PG; Human)
Total-seq A mouse anti-human CD314 (NKG2D) (clone 1D11; BioLegend; # 320835; PG; Human)
Total-seq A mouse anti-human CD119 (IFNGR1) (clone GIR-208; BioLegend; # 308607; PG; Human)
Total-seq A mouse anti-human CD155 (PVR) (clone SKII.4; BioLegend; # 337623; PG; Human)
Total-seq A mouse anti-human Streptavidin (clone -; BioLegend; # 405251; PG; All Species)
Total-seq A mouse anti-human pan-NK2GDL (clone -; R&D Systems; # 1299-NK-050; Binding Activity; Human)
Total-seq B mouse anti-human CD86 (clone IT2.2; BioLegend; # 399904; PG; Human)
Total-seq B mouse anti-human CD274 (B7-H1, PD-L1) (clone 29E.2A3; BioLegend; # 399904; PG; Human)
Total-seq B mouse anti-human CD270 (HVEM, TR2) (clone 122; BioLegend; # 399904; PG; Human)
Total-seq B mouse anti-human CD155 (PVR) (clone SKII.4; BioLegend; # 399904; PG; Human)
Total-seq B mouse anti-human CD112 (Nectin-2) (clone TX31; BioLegend; # 399904; PG; Human)
Total-seq B mouse anti-human CD47 (clone CC2C6; BioLegend; # 399904; PG; Human)
Total-seq B mouse anti-human CD48 (clone BJ40; BioLegend; # 399904; PG; Human)
Total-seq B mouse anti-human CD40 (clone 5C3; BioLegend; # 399904; PG; Human)
Total-seq B mouse anti-human CD154 (clone 24-31; BioLegend; # 399904; PG; Human)
Total-seq B mouse anti-human CD52 (clone HI186; BioLegend; # 399904; PG; Human)
Total-seq B mouse anti-human CD3 (clone UCHT1; BioLegend; # 399904; PG; Human)
Total-seq B mouse anti-human CD8 (clone SK1; BioLegend; # 399904; PG; Human)
Total-seq B mouse anti-human CD56 (NCAM) (clone 5.1H11; BioLegend; # 399904; PG; Human)
Total-seq B mouse anti-human CD19 (clone HIB19; BioLegend; # 399904; PG; Human)
Total-seq B mouse anti-human CD33 (clone P67.6; BioLegend; # 399904; PG; Human)
Total-seq B mouse anti-human CD11c (clone S-HCL-3; BioLegend; # 399904; PG; Human)
Total-seq B mouse anti-human HLA-A,B,C (clone W6/32; BioLegend; # 399904; PG; Human)
Total-seq B mouse anti-human CD45RA (clone HI100; BioLegend; # 399904; PG; Human)
Total-seq B mouse anti-human CD123 (clone 6H6; BioLegend; # 399904; PG; Human)
Total-seq B mouse anti-human CD7 (clone CD7-6B7; BioLegend; # 399904; PG; Human)
Total-seq B mouse anti-human CD105 (clone 43A3; BioLegend; # 399904; PG; Human)
Total-seq B mouse anti-human anti-human/mouse CD49f (clone GoH3; BioLegend; # 399904; PG; Human)
Total-seq B mouse anti-human CD194 (CCR4) (clone L291H4; BioLegend; # 399904; PG; Human)
Total-seq B mouse anti-human CD4 (clone RPA-T4; BioLegend; # 399904; PG; Human)
Total-seq B mouse anti-human anti-mouse/human CD44 (clone IM7; BioLegend; # 399904; PG; Human)
Total-seq B mouse anti-human CD14 (clone M5E2; BioLegend; # 399904; PG; Human)
Total-seq B mouse anti-human CD16 (clone 3G8; BioLegend; # 399904; PG; Human)
Total-seq B mouse anti-human CD25 (clone BC96; BioLegend; # 399904; PG; Human)
Total-seq B mouse anti-human CD45RO (clone UCHL1; BioLegend; # 399904; PG; Human)
Total-seq B mouse anti-human CD279 (PD-1) (clone EH12.2H7; BioLegend; # 399904; PG; Human)
Total-seq B mouse anti-human TIGIT (VSTM3) (clone A15153G; BioLegend; # 399904; PG; Human)
Total-seq B mouse anti-human Mouse IgG1, κ isotype  (clone MOPC-21; BioLegend; # 399904; PG; Human)
Total-seq B mouse anti-human Mouse IgG2a, κ isotype  (clone MOPC-173; BioLegend; # 399904; PG; Human)
Total-seq B mouse anti-human Mouse IgG2b, κ isotype  (clone MPC-11; BioLegend; # 399904; PG; Human)
Total-seq B mouse anti-human Rat IgG2b, κ isotype  (clone RTK4530; BioLegend; # 399904; PG; Human)
Total-seq B mouse anti-human CD20 (clone 2H7; BioLegend; # 399904; PG; Human)
Total-seq B mouse anti-human CD335 (NKp46) (clone 9E2; BioLegend; # 399904; PG; Human)
Total-seq B mouse anti-human CD31 (clone WM59; BioLegend; # 399904; PG; Human)
Total-seq B mouse anti-human Podoplanin (clone NC-08; BioLegend; # 399904; PG; Human)
Total-seq B mouse anti-human CD146 (clone P1H12; BioLegend; # 399904; PG; Human)
Total-seq B mouse anti-human IgM (clone MHM-88; BioLegend; # 399904; PG; Human)
Total-seq B mouse anti-human CD5 (clone UCHT2; BioLegend; # 399904; PG; Human)
Total-seq B mouse anti-human CD195 (CCR5) (clone J418F1; BioLegend; # 399904; PG; Human)
Total-seq B mouse anti-human CD32 (clone FUN-2; BioLegend; # 399904; PG; Human)
Total-seq B mouse anti-human CD196 (CCR6) (clone G034E3; BioLegend; # 399904; PG; Human)
Total-seq B mouse anti-human CD185 (CXCR5) (clone J252D4; BioLegend; # 399904; PG; Human)
Total-seq B mouse anti-human CD103 (Integrin αE) (clone Ber-ACT8; BioLegend; # 399904; PG; Human)

Total-seq B mouse anti-human CD69 (clone FN50; BioLegend; # 399904; PG; Human)
Total-seq B mouse anti-human CD62L (clone DREG-56; BioLegend; # 399904; PG; Human)
Total-seq B mouse anti-human CD161 (clone HP-3G10; BioLegend; # 399904; PG; Human)
Total-seq B mouse anti-human CD152 (CTLA-4) (clone BNI3; BioLegend; # 399904; PG; Human)
Total-seq B mouse anti-human CD223 (LAG-3) (clone 11C3C65; BioLegend; # 399904; PG; Human)
Total-seq B mouse anti-human KLRG1 (MAFA) (clone SA231A2; BioLegend; # 399904; PG; Human)
Total-seq B mouse anti-human CD27 (clone O323; BioLegend; # 399904; PG; Human)
Total-seq B mouse anti-human CD107a (LAMP-1) (clone H4A3; BioLegend; # 399904; PG; Human)
Total-seq B mouse anti-human CD95 (Fas) (clone DX2; BioLegend; # 399904; PG; Human)
Total-seq B mouse anti-human CD134 (OX40) (clone Ber-ACT35 (ACT35); BioLegend; # 399904; PG; Human)
Total-seq B mouse anti-human HLA-DR (clone L243; BioLegend; # 399904; PG; Human)
Total-seq B mouse anti-human CD1c (clone L161; BioLegend; # 399904; PG; Human)
Total-seq B mouse anti-human CD11b (clone ICRF44; BioLegend; # 399904; PG; Human)
Total-seq B mouse anti-human CD64 (clone 10.1; BioLegend; # 399904; PG; Human)
Total-seq B mouse anti-human CD141 (Thrombomodulin) (clone M80; BioLegend; # 399904; PG; Human)
Total-seq B mouse anti-human CD1d (clone 51.1; BioLegend; # 399904; PG; Human)
Total-seq B mouse anti-human CD314 (NKG2D) (clone 1D11; BioLegend; # 399904; PG; Human)
Total-seq B mouse anti-human CD35 (clone E11; BioLegend; # 399904; PG; Human)
Total-seq B mouse anti-human CD57 Recombinant (clone QA17A04; BioLegend; # 399904; PG; Human)
Total-seq B mouse anti-human CD272 (BTLA) (clone MIH26; BioLegend; # 399904; PG; Human)
Total-seq B mouse anti-human anti-human/mouse/rat CD278 (ICOS) (clone C398.4A; BioLegend; # 399904; PG; Human)
Total-seq B mouse anti-human CD58 (LFA-3) (clone TS2/9; BioLegend; # 399904; PG; Human)
Total-seq B mouse anti-human CD39 (clone A1; BioLegend; # 399904; PG; Human)
Total-seq B mouse anti-human CX3CR1 (clone K0124E1; BioLegend; # 399904; PG; Human)
Total-seq B mouse anti-human CD24 (clone ML5; BioLegend; # 399904; PG; Human)
Total-seq B mouse anti-human CD21 (clone Bu32; BioLegend; # 399904; PG; Human)
Total-seq B mouse anti-human CD11a (clone TS2/4; BioLegend; # 399904; PG; Human)
Total-seq B mouse anti-human CD79b (Igβ) (clone CB3-1; BioLegend; # 399904; PG; Human)
Total-seq B mouse anti-human CD244 (2B4) (clone C1.7; BioLegend; # 399904; PG; Human)
Total-seq B mouse anti-human CD169  (clone 7-239; BioLegend; # 399904; PG; Human)
Total-seq B mouse anti-human anti-human/mouse integrin β7 (clone FIB504; BioLegend; # 399904; PG; Human)
Total-seq B mouse anti-human CD268 (BAFF-R) (clone 11C1; BioLegend; # 399904; PG; Human)
Total-seq B mouse anti-human CD42b (clone HIP1; BioLegend; # 399904; PG; Human)
Total-seq B mouse anti-human CD54 (clone HA58; BioLegend; # 399904; PG; Human)
Total-seq B mouse anti-human CD62P (P-Selectin) (clone AK4; BioLegend; # 399904; PG; Human)
Total-seq B mouse anti-human CD119 (IFN-γ R α chain) (clone GIR-208; BioLegend; # 399904; PG; Human)
Total-seq B mouse anti-human TCR α/β (clone IP26; BioLegend; # 399904; PG; Human)
Total-seq B mouse anti-human Rat IgG1, κ isotype  (clone RTK2071; BioLegend; # 399904; PG; Human)
Total-seq B mouse anti-human Rat IgG2a, κ Isotype  (clone RTK2758; BioLegend; # 399904; PG; Human)
Total-seq B mouse anti-human CD192 (CCR2) (clone K036C2; BioLegend; # 399904; PG; Human)
Total-seq B mouse anti-human CD122 (IL-2Rβ) (clone TU27; BioLegend; # 399904; PG; Human)
Total-seq B mouse anti-human FcεRIα (clone AER-37 (CRA-1); BioLegend; # 399904; PG; Human)
Total-seq B mouse anti-human CD41 (clone HIP8; BioLegend; # 399904; PG; Human)
Total-seq B mouse anti-human CD137 (4-1BB) (clone 4B4-1; BioLegend; # 399904; PG; Human)
Total-seq B mouse anti-human CD163 (clone GHI/61; BioLegend; # 399904; PG; Human)
Total-seq B mouse anti-human CD83 (clone HB15e; BioLegend; # 399904; PG; Human)
Total-seq B mouse anti-human CD124 (IL-4Rα) (clone G077F6; BioLegend; # 399904; PG; Human)
Total-seq B mouse anti-human CD13 (clone WM15; BioLegend; # 399904; PG; Human)
Total-seq B mouse anti-human CD2 (clone TS1/8; BioLegend; # 399904; PG; Human)
Total-seq B mouse anti-human CD226 (DNAM-1) (clone 11A8; BioLegend; # 399904; PG; Human)
Total-seq B mouse anti-human CD29 (clone TS2/16; BioLegend; # 399904; PG; Human)
Total-seq B mouse anti-human CD303 (BDCA-2) (clone 201A; BioLegend; # 399904; PG; Human)
Total-seq B mouse anti-human CD49b (clone P1E6-C5; BioLegend; # 399904; PG; Human)
Total-seq B mouse anti-human CD81 (TAPA-1) (clone 5A6; BioLegend; # 399904; PG; Human)
Total-seq B mouse anti-human IgD (clone IA6-2; BioLegend; # 399904; PG; Human)
Total-seq B mouse anti-human CD18 (clone TS1/18; BioLegend; # 399904; PG; Human)
Total-seq B mouse anti-human CD28 (clone CD28.2; BioLegend; # 399904; PG; Human)
Total-seq B mouse anti-human CD38 (clone HIT2; BioLegend; # 399904; PG; Human)
Total-seq B mouse anti-human CD127 (IL-7Rα) (clone A019D5; BioLegend; # 399904; PG; Human)
Total-seq B mouse anti-human CD45 (clone HI30; BioLegend; # 399904; PG; Human)
Total-seq B mouse anti-human CD22 (clone S-HCL-1; BioLegend; # 399904; PG; Human)
Total-seq B mouse anti-human CD71 (clone CY1G4; BioLegend; # 399904; PG; Human)
Total-seq B mouse anti-human CD26 (clone BA5b; BioLegend; # 399904; PG; Human)
Total-seq B mouse anti-human CD115 (CSF-1R) (clone 9-4D2-1E4; BioLegend; # 399904; PG; Human)
Total-seq B mouse anti-human CD63 (clone H5C6; BioLegend; # 399904; PG; Human)
Total-seq B mouse anti-human CD304 (Neuropilin-1) (clone 12C2; BioLegend; # 399904; PG; Human)
Total-seq B mouse anti-human CD36 (clone 5-271; BioLegend; # 399904; PG; Human)
Total-seq B mouse anti-human CD172a (SIRPα) (clone 15-414; BioLegend; # 399904; PG; Human)
Total-seq B mouse anti-human CD72 (clone 3F3; BioLegend; # 399904; PG; Human)
Total-seq B mouse anti-human CD158 (KIR2DL1/S1/S3/S5) (clone HP-MA4; BioLegend; # 399904; PG; Human)
Total-seq B mouse anti-human CD93 (clone VIMD2; BioLegend; # 399904; PG; Human)
Total-seq B mouse anti-human CD49a (clone TS2/7; BioLegend; # 399904; PG; Human)
Total-seq B mouse anti-human CD49d (clone 9F10; BioLegend; # 399904; PG; Human)
Total-seq B mouse anti-human CD73 (Ecto-5'-nucleotidase) (clone AD2; BioLegend; # 399904; PG; Human)
Total-seq B mouse anti-human CD9 (clone HI9a; BioLegend; # 399904; PG; Human)
Total-seq B mouse anti-human TCR Vα7.2 (clone 3C10; BioLegend; # 399904; PG; Human)

Total-seq B mouse anti-human TCR Vδ2 (clone B6; BioLegend; # 399904; PG; Human)
Total-seq B mouse anti-human LOX-1 (clone 15C4; BioLegend; # 399904; PG; Human)
Total-seq B mouse anti-human CD158b (KIR2DL2/L3, NKAT2) (clone DX27; BioLegend; # 399904; PG; Human)
Total-seq B mouse anti-human CD158e1 (KIR3DL1, NKB1) (clone DX9; BioLegend; # 399904; PG; Human)
Total-seq B mouse anti-human CD142 (clone NY2; BioLegend; # 399904; PG; Human)
Total-seq B mouse anti-human CD319 (CRACC) (clone 162.1; BioLegend; # 399904; PG; Human)
Total-seq B mouse anti-human CD352 (NTB-A) (clone NT-7; BioLegend; # 399904; PG; Human)
Total-seq B mouse anti-human CD94 (clone DX22; BioLegend; # 399904; PG; Human)
Total-seq B mouse anti-human CD162 (clone KPL-1; BioLegend; # 399904; PG; Human)
Total-seq B mouse anti-human CD85j (ILT2) (clone GHI/75; BioLegend; # 399904; PG; Human)
Total-seq B mouse anti-human CD23 (clone EBVCS-5; BioLegend; # 399904; PG; Human)
Total-seq B mouse anti-human CD328 (Siglec-7) (clone 6-434; BioLegend; # 399904; PG; Human)
Total-seq B mouse anti-human HLA-E (clone 3D12; BioLegend; # 399904; PG; Human)
Total-seq B mouse anti-human CD82 (clone ASL-24; BioLegend; # 399904; PG; Human)
Total-seq B mouse anti-human CD101 (BB27) (clone BB27; BioLegend; # 399904; PG; Human)
Total-seq B mouse anti-human CD88 (C5aR) (clone S5/1; BioLegend; # 399904; PG; Human)
Total-seq B mouse anti-human CD224 (clone KF29; BioLegend; # 399904; PG; Human)
Total-seq B mouse anti-human TIM3 (CD366) (clone F38-2E2; BioLegend; # 345053; PG; Human)
Total-seq B mouse anti-human CD99 (clone 3B2/TA8; BioLegend; # 371323; PG; Human)
Total-seq B mouse anti-human CD117 (clone 104D2; BioLegend; # 313247; PG; Human)
Total-seq B mouse anti-human CD34 (clone 581; BioLegend; # 343539; PG; Human)
Total-seq B mouse anti-human CD90 (clone 5E10; BioLegend; # 328147; PG; Human)
Total-seq B mouse anti-human CD10 (clone HI10a; BioLegend; # 312235; PG; Human, Cynomolgus, Rhesus)
Total-seq B mouse anti-human CD135 (clone BV10AH2; BioLegend; # 313321; PG; Human)
Total-seq B mouse anti-human CD371 (CLEC12A)  (clone 50C1; BioLegend; # 353619; PG; Human)
Total-seq B mouse anti-human GPR56 (clone CG4; BioLegend; # 358211; PG; Human, Mouse)

M-FISH (clone; manufacturer; catalogue number; application; reactivity):
Goat anti-Avidin (clone -; Vector; # BA-0300; IF, ISH; All species)
Rabbit anti-Digoxin (polyclonal; Sigma Aldrich; # D7782; FISH, ChIP, AC)
Goat anti-rabbit Cy5.5 (polyclonal; Linaris, # PAK0027; IF, EB; Rabbit)
Streptavidin AF750 conjugate (clone -; Invitrogen; # S21384; IF; All species)

| | |
|---|---|
| Validation | For all antibodies we relied on manufacturers' validation for species reactivity and applications. Validation data is available on the manufacturer's website with respective statements from manufacturer's websites also  given above. |

## Animals and other research organisms

Policy information about studies involving animals; ARRIVE guidelines recommended for reporting animal research, and Sex and Gender in Research

| | |
|---|---|
| Laboratory animals | Female NOD.Prkdcscid.Il2rgnull (NSG) mice 8-12 weeks of age were used in the study. Mice were housed in individually ventilated cages with controlled temperature (approx. 22 ºC) and humidity (50%) under 12-12 h light-dark cycle. |
| Wild animals | The study did not involve wild animals. |
| Reporting on sex | Findings apply only to one sex. |
| Field-collected samples | The study did not include samples collected from the field. |
| Ethics oversight | Animal experiments were conducted in compliance with all relevant ethical regulations. We obtained written, informed consent for all experiments and they were approved by the Regierungspräsidium Karlsruhe under Tierversuchsantrag numbers G42/18 and G-140-21. |

Note that full information on the approval of the study protocol must also be provided in the manuscript.

## Flow Cytometry

### Plots

Confirm that:

☒ The axis labels state the marker and fluorochrome used (e.g. CD4-FITC).

☒ The axis scales are clearly visible. Include numbers along axes only for bottom left plot of group (a 'group' is an analysis of identical markers).

☒ All plots are contour plots with outliers or pseudocolor plots.

☒ A numerical value for number of cells or percentage (with statistics) is provided.

### Methodology

| | |
|---|---|
| Sample preparation | Primary human AML cells at diagnosis were recovered from cryopreserved bone marrow and/or peripheral blood samples. Patient-derived xenografts were generated by intrafemoral injection of 1-2 Million viable primary AML cells in NSG mice. |

PDX-derived cells were frozen until processing. All samples were thawed at 37 °C in Iscove's modified Dulbecco's medium (IMDM) containing 10% FBS, and treated with DNase I for 15min (100 μg/ml).

For CITE-seq analysis, recovered cells were stained with a total of 38 or 149 antibody-derived tags (ADTs) as well as CD45, CD3 and 7-AAD (see Supplementary Table 11). Cells were sorted for live CD45+ cells.

For ex vivo drug profiling, recovered cells were cultured using previously established protocol using IMDM, 15% BIT (bovine serum albumin, insulin, transferrin; Stem Cell Technologies, cat # 09500), 100 ng/ml SCF (PeproTech, cat # 300-07), 50 ng/ml FLT3-L (PeproTech, cat # 300-19), 20 ng/ml IL-3 (PeproTech, cat # 200-03), 20 ng/ml G-CSF (PeproTech, cat # 300-23), 100 μM β-mercaptoethanol (ThermoFisher, cat # 31350010), 500 nM SR1 (StemRegenin 1, STEMCELL Technologies, cat # 72342), 500 nM UM729 (STEMCELL Technologies, cat # 72332), and 1% penicillin-streptomycin (Sigma, cat # P4458-100ML). 0.5x10^5 AML cells/well were seeded in flat-bottom 96-well plates, and cells were treated with up to 12 treatment conditions consisting of standard chemotherapy regimens as well as novel compounds for 24h and for selected conditions also for 72h (Supplementary Table 9). After 24h/72h, the cells were stained with cell surface antibodies (see Supplementary Table 12). Same amount of CountBright Absolute Counting Beads (Thermo Fisher Scientific, cat # C36950) together with 7-AAD (BD Biosciences, cat # 559925) were added to each sample prior to analysis with BD LSRFortessa Cell Analyzer.

| Instrument | BD FACSAria™ Fusion I or II Cell Sorter, BD FACSAria™ III Cell Sorter, BD LSRFortessa™ Cell Analyzer |
|---|---|
| Software | FlowJo, BD FACSDiva |

**Cell population abundance**

Due to limited sample material, post-sorting purities were not re-assessed using flow cytometry. Instead, this was done by gating and quantificatoin of populations using FlowJo. On average 80.75% of the total events were included after gating out debris in the FSC-A vs SSC-A plot; on average 98.7% of these events were within the single cells gate (based on FSC-A vs FSC-H); on average 89.8% of these single cells were gated as viable cells (based on 7-AAD vs SSC-A); and the final sorting population of CD45+ cells represented on average 99.7% of these viable cells (based on CD45 vs SSC-A).

**Gating strategy**

For CITE-seq sorting: FSC-A vs SSC-A was the starting gate wherein debris was excluded. Next, single cells were gated based on the exclusion of outliers in FSC-A vs FSC-H. Viable cells were then gated within this population based on low 7-AAD staining (SSC-A vs 7-AAD). Finally, the ultimate sorting population of CD45+ leukocytes were gated based on dim to high expression of CD45 (SSC-A vs CD45).

For ex vivo drug profiling: The gating strategy of live leukemic cells is depicted in Supplementary Figure 12. FSC-A vs SSC-A was the starting gate wherein debris was excluded. Next, single cells were gated based on the exclusion of outliers in FSC-A vs FSC-H. Viable cells were then gated within this population based on low 7-AAD staining (SSC-A vs 7-AAD). Next lineage-positive cells were excluded and leukemic cells were gated based on low expression of the lineage markers (CD3/CD20/CD235a vs CD45). Finally, leukemic cells were discriminated from the remaining immune cells via dim expression of CD45 and low to mid/high SSC-A (CD45 vs SSC-A).

☒ Tick this box to confirm that a figure exemplifying the gating strategy is provided in the Supplementary Information.

