## [Peer Review File · Nature Genetics]

Single-cell multi-omics analysis reveals dynamic clonal evolution and targetable phenotypes in acute myeloid leukemia with complex karyotype

Corresponding Author: Professor Andreas Trumpp

A version of this paper was originally rejected for publication by Nature Genetics, however that decision was reconsidered after appeal by the authors.

Version 0:

Decision Letter:

22nd Aug 2023

Dear Professor Trumpp,

How are you? I hope you're well.

First, please accept my apologies for the delay in returning this decision to you. Thank you for bearing with me.

Your Article entitled "Identification of targetable relapse-driving leukemic stem cells in complex karyotype AML" has now been seen by 3 referees, whose comments are attached. While they find your work of potential interest, they have raised serious concerns which in our view are sufficiently important that they preclude publication of the work in Nature Genetics, at least in its present form.

While the referees find your work of some interest, they raise concerns about the strength of the novel conclusions that can be drawn at this stage. A key concern is the limited number of samples used. We fully appreciate that these are technically challenging and expensive experiments, but the reviewers concur that the sample size undercuts the generalisability of the research. If you are interested in appealing our decision, these issues will need to be treated as a priority although we would encourage you to address all concerns in full either with new analyses (where possible) or through textual changes (where appropriate).

Should further experimental data allow you to fully address these criticisms we would be willing to consider an appeal of our decision (unless, of course, something similar has by then been accepted at Nature Genetics or appeared elsewhere). This includes submission or publication of a portion of this work someplace else.

The required new experiments and data include, but are not limited to those detailed here. We hope you understand that until we have read the revised manuscript in its entirety we cannot promise that it will be sent back for peer review.

If you are interested in attempting to revise this manuscript for submission to Nature Genetics in the future, please contact me to discuss a potential appeal. Otherwise, we hope that you find our referees' comments helpful when preparing your manuscript for resubmission elsewhere.

Although we cannot publish your paper, it may be appropriate for another journal in the Nature Portfolio. If you wish to explore the journals and transfer your manuscript please use our manuscript transfer portal. You will not have to re-supply manuscript metadata and files, but please note that this link can only be used once and remains active until used. For more information, please see our [manuscript transfer FAQ](http://www.nature.com/authors/author_resources/transfer_manuscripts.html?WT.mc_id=EMI_NPG_1511_AUTHORTRANSF&WT.ec_id=AUTHOR) page.

Sincerely,

Safia Danovi
Editor
Nature Genetics

Referee expertise:

Referee #1: AML, leukemia evolution, single-cell analysis

Referee #2: haematopoietic malignancies malignancies, in vivo

Referee #3: single-cell methods, AML

Reviewers' Comments:

Reviewer #1:

Remarks to the Author:

In the article from Trumpp and colleagues, the authors use single cell methods to determine the intra-tumoral heterogeneity in complex karyotype AML and link this to leukemia stem cells and therapeutic sensitivity. The authors use various genotyping methods to determine the structural variants present at the single cell level. They deeply analyze four cases of CK-AML demonstrating a complex collection of clonally related subclones with evidence of multiple genetic mechanisms for the complex karyotype including chromothripsis, extrachromosomal circular DNA elements, inversions, deletions, and others. Using single cell methods of scNOVA and CITE-Seq, they demonstrate differences in epigenetic programs and some minor, but primarily overlapping, surface immunophenotypes. Notably, xenograft studies show that in 2 of 2 cases, only a single clone engrafts in multiple mice, and that clone had some increased measures of stem cell programs, although not markedly different from non-engrafting clones. They show that the xenograft engrafting clone in CK349 is responsible for relapse in the patient treated with several agents including allogeneic transplant. Finally, they explore potential drug sensitivity of two cases, implicating chemo resistance preferentially in SC3 in CK349, the clone responsible for the clinical relapse. In case CK282, they showed BCL-xL dependence and inhibitor sensitivity in the stem-like and engrafting SC2 subclone.

The study and characterization of CK-AML is of great interest to the field. The authors have conducted the most detailed genomic analysis of such cases to date, as well as the first single cell level analysis to dissect subclones and epigenetic programs. Many of the findings are novel and important for understanding this particularly poor-risk subset of AML. The paper is detailed and well-written. However, the engraftment characterization and link to relapse is conducted with only 2 samples and 1 sample, respectively, making it difficult to generate conclusions with broader implications for CK-AML. Moreover, the pharmacologic studies are very limited in significance and conclusions.

1. There are only 278 total cells analyzed for the genotyping which is very small ranging from 49-91 per sample and with only 1.2x genome coverage. This leaves a lot of potential for missing clones and significant inability to call SVs.
2. The authors do a good job of performing orthogonal experiments to validate observations made from their single cell SV analyses. Can the authors comment on how much additional information is lost or gained with scTRIP compared to bulk analyses? Although the SV resolution at the single cell level is enhanced, is the sensitivity of detecting SVs at a global level similar to bulk analysis methods? I would imagine that fewer SVs are detected using the single cell method compared to bulk techniques. Can the authors perform this comparison? Further, the authors used data derived from the Strand-seq data as input for their CNA recalling in the CITE-seq data. What is the value, if any, of using this single cell approach rather than break points observed from bulk tissue assays as input to CONICSmatrix? Would you still be able to confidently call subclones in the CITE-seq data without scTRIP?
3. In Fig 4 (line 362-364, 378-381), the authors demonstrate that the different subclones deconvolve differently into normal HSPC states with some subclones associated with more immature and stem cell-like states. They suggest that the genomic aberrations may be causal in the increased measures of stemness, but this correlation does not necessarily imply causation. It is also possible that the genomic aberrations of these clones occur within a more immature/stem compartment that is then retained in the eventual leukemic clone.
4. In the PDX engrafted cells, were new SVs or SNVs identified that suggested ongoing clonal evolution as the cells engrafted and proliferated in the mice? This might be expected as the engrafting clone demonstrated high heterogeneity and instability.
5. In Fig 6, the authors suggest that the CK349 SC3 clone responsible for engraftment and clinical relapse is more sensitive to venetoclax. However, CK-AML and TP53-mutant AML generally speaking is known to be poorly responsive to venetoclax and HMA-containing regimens. What is the venetoclax sensitivity of CK-AML from the BeatAML cohort? Is this case an outlier?
6. In Fig 6, the pharmacologic analysis of CK282 is questionable. Review of the data (panel 6H, and S14c) suggests only minor differences in BCL-xL expression between the SC2 clone and the remainder of the leukemic blasts. Are the authors proposing that this difference drives preferential susceptibility of SC2 to BCL-xL inhibition? Is there specificity for SC2 clone, as the heat map in panel 6g suggest all clones are similarly susceptible. This data detracts from the main genetic studies in the manuscript and could be removed.
7. The data on CK349 in Figure 5 and 6 could be combined into one figure as an example of how analysis of individual

subclones yields clinically relevant findings.

8. The Discussion is inadequate as many conclusions are stated as applicable to the broad class of CK-AML, but are derived from data from only 2, or even 1 sample, as is the case for the LSC, engraftment, and drug testing. Such studies with primary patient samples are difficult and expensive, but the conclusions should be tempered based on these considerations. Similarly, the depth of genomic profiling and the number of cells analyzed is also very small, and care should be taken to acknowledge these limitations as well.

Reviewer #2:

Remarks to the Author:

This manuscript presents a method for single multi-omic assessment of structural variants along with nucleosome occupancy and gene expression (merging existing scNOVA-seq from PMID 36424487 with CITE-seq) and applies it to a set of four AML patient samples with complex karyotype AML. The manuscript presents detailed single cell resolution of structural variants in both AML patient samples and in PDX's derived from AML patient samples. While the genomic studies provide novel observations about clonality of structural variants and chromosomal instability in AML, the data in Figures 5-6 which attempt to draw conclusions about relapse and therapy are underdeveloped. These issues are as follows:

-Figure 5 is based on a single patient sample and the relapse from this patient was not subject to any type of genomic analysis (other conventional karyotyping). It is therefore unclear that the conclusions about clones which drive relapse are solidly supported by the data. The analyses in Figure 5 would require more patient samples evaluated via sc-Nova at diagnosis and relapse at a minimum, and with PDX models if the authors want to draw conclusions about PDX-engrafting LSCs driving "resistance and relapse" as claimed.

-The therapeutic studies in the manuscript are quite modest in their findings. It is established from pivotal clinical trials of venetoclax and hypomethylating agents that patients with complex karyotype AML are less responsive to the regimen of azacitidine and venetoclax than patients with more favorable/intermediate risk cytogenetics (see PMID 32786187 amongst other publications). The lack of favorable outcomes of patients with CK AML to this regimen is the basis for much ongoing clinical trials and preclinical studies in AML. Thus, the single patient ex vivo observation in Figure 6 is clearly not representative of responses to this regimen in actual patients. This questions the reliability of the claims made and/or the assay used in Figure 6.

-Related to the above, Bcl-xL (the other therapeutic highlighted in Figure 6) is a well-known therapeutic target in AML and the same Bcl-xL inhibitor as shown here has been recently suggested to overcome p53 mutant AML which lacks response to azacitidine and venetoclax (PMID 36508699).

Reviewer #3:

Remarks to the Author:

In their manuscript "Identification of targetable relapse-driving leukemic stem cells in complex karyotype AML", Leppä et al. set out to characterize intra-tumor heterogeneity in four cases of complex karyotype acute myeloid leukemia (CK-AML). Specifically, the authors generate Strand-seq data for a total of 278 cells from the primary patient samples, which is analyzed using the scNOVA computational approach to reveal somatic structural variation (SV) >200kb and nucleosome occupancy profiles with single-cell resolution. The authors reveal the existence of multiple subclones with distinct karyotypes in two of the four CK-AML cases, which they suggest are shaped by (linear and circular) breakage fusion bridge cycles. In addition, both chromothripsis-(like) patterns and ongoing karyotype remodeling of some cells is observed. On the most genetically heterogeneous cases, CK282 and CK349, CITE-seq was performed, providing additional high-throughput transcriptome and targeted cell surface proteome data for ~14k cells. The authors propose the scNOVA-CITE workflow for integration of these datasets, enabling them to couple (dynamic) genomic, transcriptomic and functional information at the single-cell level. Based on analysis of the integrated data, the authors propose that genomically unstable subclones carry a leukemic stem cell (LSC)-associated stemness signature. Strand-seq & scNOVA analysis of patient-derived xenograft (PDX) models indicates superior engraftment of the identified LSC-like subclone in these two cases, which is supported by karyotyping in a CK349 relapse sample. Finally, the authors perform drug sensitivity profiling using combinations of standard chemotherapy regimens and BH3 mimetics to target these relapse-driving genomic instability and stemness-associated phenotypes, highlighting a potential therapeutic options to eradicate relapse-driving LSC populations.

The manuscript is clear and well-structured but could be more succinct (e.g., description of the subclones, SVs and potential mutational mechanisms in CK282 and CK349, lines 229–278). The Results contain several sections which are more hypothetical and would better benefit a Discussion, which is currently limited.

Overall, the main novelty presented lies in the combined scNOVA-CITE workflow and its integration of the different modalities generated (structural genomic, open chromatin, transcriptome and targeted surface proteome) on patient material and clinically relevant models. Specifically, the application to primary and PDX CK-AML samples provides a unique and intriguing insight into the evolution and intratumor genetic and phenotypic heterogeneity of the disease. The work may hence provide a blueprint for future (larger) studies into the interplay between genome and phenotype during tumor evolution.

That said, the number of samples is limited (4 patients initially, of which only 2 are actually explored using scNOVA-CITE) and the core of the integration pipeline (linking subclonal populations identified using Strand-seq and scNOVA to cell populations detected in CITE-seq), which underpins all downstream analyses, lacks assessment or benchmarking. The limited number of cells for engrafting subclones, small effect size for the LSC signatures, and potential limitations of the SV

burden and karyotype heterogeneity metric further limit the generalizability of claimed links between genomic instability, stemness, engraftment, and therapy resistance/relapse.

Specifically, I have the following main concerns:

1. Integration of the Strand- and CITE-seq data relies on the scNOVA-CITE workflow and accurate genotyping of subclone-specific copy number changes in the CITE-seq gene expression data. Copy number inference from scRNA-seq is noisy however, even when breakpoints are used as a prior, and especially when going to a lower threshold of 10 expressed genes as is the case here. In addition, methods for copy number calling from Strand-seq data have not been published (a manuscript by Christiansen et al. is said to be in preparation). While broad classes such as “gain” or “loss” may be easily picked up, specific copy number states will be harder to designate. Furthermore, while CITE-seq is performed directly on flow sorted cells from the thawed cryopreserved samples, the Strand-seq data derives from cultured patient material. As such, correspondence between the two modalities may not be guaranteed, as the passaging may induce biases. Considering the above and in the absence of an in-depth assessment or benchmark (for instance, using a ground truth single-cell genome + transcriptome dataset), it is hard for this reviewer to assess the accuracy of the integration workflow and its downstream results.

2. The authors use SV numbers and their variability across cells to assess karyotype heterogeneity. There are several caveats to this: (1) SVs <200kb and potentially also larger simple copy number gains are missed by scNOVA. The latter is most evident within amplified regions as in Figures 2a or 5c, where distinct coverage peaks support the existence of multiple missed segments within the larger regions. (2) SV detection and so these metrics are likely affected by sequencing depth per cell. This confounder is not explored. (3) By modifying expression of potentially hundreds or thousands of genes in-cis, SVs that alter copy number are likely to have more drastic effects on cell phenotypes and hence functionally relevant karyotype heterogeneity than balanced rearrangements such as inversions or translocations which may affect just a handful of genes near their breakpoints. Nevertheless, these classes are counted equally, potentially limiting the utility and comparability of the metrics. Did the authors consider other metrics for functionally relevant karyotype heterogeneity?

3. While this reviewer appreciates the amount of work performed, sample sizes (n=4 for Strand-seq, but only 2 for scNOVA-CITE) and cell counts do not always support the general claims made in the manuscript. For instance, based on perceived differences in cell type fractions assigned using correlation in nucleosome occupancy patterns for subclones represented by only 5 and 4 cells (Fig 4a) and small differences in expression signatures (Fig 4b), authors conclude on lines 362–364: “These data suggest an association between the genetic profile of a cell and its ability to generate more differentiated leukemic cells, arguing that complex SVs may drive stem-like features.” The authors also assume direction of causality here (as with their example of ALDH1A1), while it may be equally likely that stemness allows cells to survive accumulation of more complex SVs. These claims are then repeated as proven further in the text: e.g., lines 379–380: “... high intra-patient karyotypic heterogeneity mediated through seismic amplification and the resulting high stemness ...”. The discussion contains similarly general statements, e.g., line 521: “In CK-AML, minor subclones with complex SVs and continued karyotypic instability generate such disease-driving and therapy resistant LSCs.”

Minor comments:

4. Authors could perform CITE-seq or explore further cells with Strand-seq from the remaining two cases as they do appear to show some genetic heterogeneity, which may have become more evident with increased cell numbers. This might also help support or dismiss some of the claims highlighted above.

5. A more fine-grained and algorithmic phylogenetic reconstruction for the CK-AML cases would improve the impact and robustness of the findings and proposed methodology. The authors could explore using their copy number estimates/karyotypes and recently published tools such as SCARLET, Medalt, MEDICC2, among others.

6. It is unclear whether any non-leukemic diploid cells were captured in the Strand-seq and CITE-seq analyses, if/how they were identified and excluded from the analyses (e.g., to avoid biasing karyotype heterogeneity measures or assignments to clones).

7. Line 38: suggest modifying “characterized by highest level” to “characterized by the highest level”

8. Quality metrics are only provided for the Strand-seq data on the original 278 cells derived from the primary patient samples. I could not find those for the PDX-derived Strand-seq data, nor for the CITE-Seq experiments. (Partially) deidentified data supporting the analyses in the form of gene or binned-read counts were not included or referred to within the manuscript and EGA identifiers are not included.

9. A number of Extended Data Figure panels are of low quality and hard to read. Specifically, Extended Data Figures 2a-c would be improved if vectorized, 7 has the scale bars mirrored and pasted in, and in 10b the right M-FISH panel is of low quality.

10. Extended Data Figure 2d – it is unclear what the brown and blue sequences indicate.

11. Figure panels 2e and 2h are unnecessarily repeated in Fig 4a. An indication of the number of cells in each subclone (n = ...) would be more helpful to the reader to assess effects.

12. Lines 136–139: I would not describe the H1-specific reduced NO (and potentially H1-specific expression) as being “driven by epigenetic dysregulation”, as this is likely caused by the rearrangement and enhancer hijacking. Suggest rephrasing.
13. It is unclear whether whole-chromosome gains are counted towards the “SV burden”.
14. The authors refer to the identified complex amplification in CK349 SC3 as a seismic amplification and refer to Rosswog et al., Nat Genet, 2021. Is this not the same mechanism proposed earlier in Garsed et al, Cancer Cell, 2014? Especially since the authors do indeed observe large ring chromosomes and linearized versions in their PDX M-FISH analysis.
15. Lines 195–199: This part is largely hypothetical and may be shortened or moved to the discussion.
16. Lines 202–206: Do the circular and linearized chr11 derivatives occasionally co-occur or are they always present in distinct cells?
17. Line 234: What is the evidence supporting the statement that these SVs are “serially acquired”?
18. Can the authors expand their discussion of the remarkable ongoing instability observed for specific chromosomes in several subclones? E.g., the frequency of distinct chr13 and chr17 rearrangements in CK349 and of chr20 in CK282? What might be the driving forces?
19. Figure 1b and Extended Data Figure 6 are clustered by their SV genotype likelihoods. It is however unclear how these likelihoods are derived. A separate heatmap akin to these figures but showing the actual underlying likelihoods would also be helpful to assess confidence in the genotype assignments.
20. Lines 274–278: “... CK-AML is not only characterized by complex genomic rearrangements leading to extensive SV density ...” and “... this makes almost every leukemia cell unique ...”. Given the limited number of samples and cells explored, authors may wish to rephrase these general statements.
21. How robust are the differential NO findings for small subclones, which were captured with just 4 or 5 cells (e.g., CK282, SC2/5)? For instance, for SC2, a disproportionate amount of differential NO signal localizes to (clusters on) chr8, which is subclonally aberrated in these cells. A similar pattern is visible for chr7 in SC5. While some of this may indeed be true signal driven by rearrangements (nice to show if the case), it is unclear to this reviewer how much missed rearrangements, (small) CN changes not normalized away, read binning, or breakend position approximation bleed into these results?
22. For the scNOVA-CITE integration, authors state on lines 311–313 that “integration of the data showed clustering of the cells mostly along genetic subclones (Fig. 3d-e)”. This reasoning is somewhat circular, since the RNA data itself was directly used to assign these subclones.
23. To support the correspondence of culture-derived Strand-seq and direct CITE-seq, validity of the integration, and derived results, authors could explore the genome-wide (or for the identified genes) correlation between NO and gene expression changes. For instance, by plotting delta-NO vs log-fold change in gene expression for all genes per subclone.
24. Lines 328–330: is the observation of PI3K-Akt-mTOR activity and other pathways in multiple subclones not in disagreement with the NO-based results, where this was specific to SC2?
25. Line 334: Cells of different subclones occasionally seem to occupy similar cell states, as the authors indicate. Could the authors shed light onto what these cell states might be, e.g., via marker gene expression or pathway analysis or considering Gavish et al., Nature, 2023?
26. Lines 266–271 and 391–392. It is unclear to this reviewer how chr20 in CK282 may show a classical and shared BFB pattern (with the DelTer and adjacent InvDup) where the initiating DelTers are of different sizes. It would also be helpful to include a figure akin to Extended Data Figure 5 showing the different chr20 patterns in CK282 and its PDX model.
27. Lines 410–411: “... only some CK-AML subclones display high engraftment potential ...”. The observation that CK282 SC2 and CK349 SC3 are the major subclones in the PDXs, does not mean the other subclones do not have high engraftment potential. They might engraft efficiently on their own. “...which is associated with their karyotypic instability and stemness activity ...”. As indicated above, the evidence supporting this here remains weak at best.
28. Lines 427–429: “The majority of these genes reside in the genomic segments affected by the DNA rearrangements at chromosome 11, and were likely upregulated due to higher CN levels in the seismic amplification.” It would be helpful if the location of these genes and the CN profile of this chromosome were visualized together.
29. Figures 6b and 6e are overplotted, making it hard to observe the distribution for each subclone.
30. Lines 506–507: “This suggested a high dependency of these cells for BCL-xL, which is in agreement with their high sensitivity to the BCL-xL-specific inhibitor and their elimination by this treatment in culture.” Both the expression and drug

sensitivity differences between the CK349 populations appear relatively minor.

31. Lines 518–524: “Our data highlights the clinical relevance of intra-patient karyotype heterogeneity with ongoing chromosomal instability and punctuated evolution-like rearrangements. Such clonal diversification at the individual cell level may allow the selection of cancer stem cell-driven subclones with increased fitness. In CK-AML, minor subclones with complex SVs and continued karyotypic instability generate such disease-driving and therapy resistant LSCs. Functionally, these cells show characteristics associated with aggressive disease, namely ability to regenerate leukemia in vivo”

a. As highlighted above, this reviewer remains unconvinced that the current sample and cell numbers as well as effect sizes support these general statements.

32. Line 593: “SV segments”, suggest rephrasing.

33. Lines 611–612: Suggest rephrasing to: “To infer differentially active genes for each subclone, the single-cells in certain subclones were compared to all other single-cells in the same tumor using an alternative mode of scNOVA based on partial least squares-discriminant analysis.”

34. In the Weighted Nearest Neighbor analysis of CITE-seq data, 18 dimensions were used for the protein modality. Given that there are just under 40 proteins measured, this number seems rather high compared to the RNA where 30 dimensions are used for the thousands of genes measured. How was this parameter determined?

35. For the “Reference-based” mapping of leukemic cells onto a multimodal BM atlas, the authors use Anchors (Seurat’s FindTransferAnchors function). This implies that both the CK-AML and the reference bone marrow data contain at least some cells of the same type, i.e., normal bone marrow cells. However, it is not clear what fraction of normal cells is still present in the CK-AML CITE-seq data, and so whether the anchor-based mapping is reliable.

36. Could the authors use the high-coverage (300x) OGM data to refine breakpoint positions in the Strand-seq data, potentially even in the subclones? This may be especially helpful in SV-dense amplified regions which are currently undersegmented in the Strand-seq.

37. Lines 969–972/Figure 2: Parts of the abbreviation list are duplicated.

38. Line 1012: add “testing” to “followed by Benjamini-Hochberg multiple correction.”

39. Extended Data Figure 9c-d: it would be helpful if the order of samples on the horizontal axes was consistent between panels.

40. Extended Data Figure 10b: How representative are the karyotypes of these two cells? The right M-FISH panel appears to indicate a homozygous loss of both chr16 and chr5q, and is unlikely to be viable.

41. Extended Data Figure 14a: Is the indicated CD45dim/SSClow/SSCmid gating strategy for singlets correct (i.e., only side and no forward scatter was used)?

Version 1:

Decision Letter:

IMPORTANT: Please note the reference number: NG-A62852R-Z Trumpp. This number must be quoted whenever you communicate with us regarding this paper.

29th May 2024

Dear Dr Trumpp,

Thank you for asking us to reconsider our decision on your manuscript “Single-cell analysis reveals dynamic clonal evolution and targetable phenotypes in complex karyotype AML”. I have now discussed your appeal with my colleagues, and we think that you have some valid points. We therefore invite you to revise your manuscript along the lines that you propose.

When preparing a revision, please ensure that it fully complies with our editorial requirements for format and style; details can be found in the Guide to Authors on our website (<http://www.nature.com/ng/>).

Please be sure that your manuscript is accompanied by a separate letter detailing the changes you have made and your response to the points raised. At this stage we will need you to upload:

1) a copy of the manuscript in MS Word .docx format.

2) The Editorial Policy Checklist:

<https://www.nature.com/documents/nr-editorial-policy-checklist.pdf>

3) The Reporting Summary:

(Here you can read about the role of the Reporting Summary in reproducible science:

<https://www.nature.com/news/announcement-towards-greater-reproducibility-for-life-sciences-research-in-nature-1.22062>)

Please use the link below to be taken directly to the site and view and revise your manuscript:

Link Redacted

With kind wishes,

Safia Danovi, PhD
Senior Editor, Nature Genetics
ORCID: 0009-0007-7822-5479

Version 2:

Decision Letter:

27th Jun 2024

Dear Professor Trumpp,

Your Article, "Single-cell analysis reveals dynamic clonal evolution and targetable phenotypes in complex karyotype AML" has now been seen by 3 referees. You will see from their comments below that while they find your work of interest, some important points are raised. We are interested in the possibility of publishing your study in Nature Genetics, but would like to consider your response to these concerns in the form of a revised manuscript before we make a final decision on publication.

We therefore invite you to revise your manuscript taking into account all reviewer and editor comments. Please highlight all changes in the manuscript text file. At this stage we will need you to upload a copy of the manuscript in MS Word .docx or similar editable format.

*2) If you have not done so already please begin to revise your manuscript so that it conforms to our Article format instructions, available

[here](http://www.nature.com/ng/authors/article_types/index.html).

*3) Include a revised version of any required Reporting Summary: <https://www.nature.com/documents/nr-reporting-summary.pdf>

Please be aware of our [guidelines](https://www.nature.com/nature-research/editorial-policies/image-integrity) on digital image standards.

Link Redacted

We hope to receive your revised manuscript within four to eight weeks. If you cannot send it within this time, please let us know.

Sincerely,

Safia Danovi, PhD
Senior Editor, Nature Genetics
ORCID: 0009-0007-7822-5479

Reviewers' Comments:

Reviewer #1:

Remarks to the Author:

I thank the authors for their thoughtful and comprehensive response to my concerns, which they have addressed to my satisfaction. I have a few minor comments:

- Should "278 single-cell genomes" be updated to 855 genomes as they report increasing their sample size?
- Please include response duration and overall survival data in Table 1A.
- If space permits, please incorporate the discussion provided in the rebuttal in the final manuscript, perhaps as supplemental information. Specifically, Rebuttal Figure R1.5, and comparison between single cell and bulk approaches (Comment 2).

Reviewer #2:

Remarks to the Author:

The authors have substantially revised the manuscript and notably increased the number of patient samples analyzed throughout. The manuscript is greatly improved.

Reviewer #3:

Remarks to the Author:

In this revised manuscript, the authors have significantly improved the quality and impact of their work. This reviewer greatly appreciates the extra work done. In general, the comments raised by the reviewers have been addressed by the authors.

I do however still struggle to assess the robustness of the Strand- and CITE-seq integration workflow for assigning CITE-Seq cells to clones identified in the Strand-seq. In contrast to what the authors suggest, in Jeong et al., Nature Biotech 2022, this reviewer could only find benchmarking of the NO to gene expression models and comparison of RNA-based copy number inference tools, both of which do not address the issue of accurate clone mapping between the two methods. In fact, the relatively poor correlation in subclonal fractions between Strand- and CITE-Seq (new Extended Data Figure 8b, driven mostly by some very big and some very small subclones) and the inability of assigning cells in one of the polyclonal patients (CK295, despite large duplications on 8q and chr16 – I do appreciate the conservative approach here), demonstrates that this mapping is more problematic than apparent from the manuscript or even the literature (Jeong et al.). While performing extensive validation might be outside the scope of the current manuscript, more insights into the copy number calls or state probabilities (on both the Strand- and CITE-seq side) may already be helpful to demonstrate that this integration is in fact robust. A joint plot, combining Rebuttal Figure R3.1 (targeted CONICsmat CNA calling results) and Rebuttal Figure R.3.7 (PloidyAssignR CNA calls) where clones are aligned and indicated in both might already help, at least visually.

Version 3:

Decision Letter:

25th Jul 2024

Dear Professor Trumpp,

Your Article, "Single-cell analysis reveals dynamic clonal evolution and targetable phenotypes in complex karyotype AML" has now been seen by Reviewer #3 who has requested additional information.

We therefore invite you to revise your manuscript, addressing their remaining points in full. Please highlight all changes in the manuscript text file. At this stage we will need you to upload a copy of the manuscript in MS Word .docx or similar editable format. Our intention will be to assess the revised manuscript in-house but we may have to return to the reviewer if absolutely necessary.

*2) If you have not done so already please begin to revise your manuscript so that it conforms to our Article format instructions, available

[here](http://www.nature.com/ng/authors/article_types/index.html).

*3) Include a revised version of any required Reporting Summary: <https://www.nature.com/documents/nr-reporting-summary.pdf>

Link Redacted

We hope to receive your revised manuscript within four to eight weeks. If you cannot send it within this time, please let us know.

Nature Genetics is committed to improving transparency in authorship. As part of our efforts in this direction, we are now requesting that all authors identified as 'corresponding author' on published papers create and link their Open Researcher and Contributor Identifier (ORCID) with their account on the Manuscript Tracking System (MTS), prior to acceptance. ORCID helps the scientific community achieve unambiguous attribution of all scholarly contributions. You can create and link your ORCID from the home page of the MTS by clicking on 'Modify my Springer Nature account'. For more information please visit please visit www.springernature.com/orcid.

Sincerely,

Safia Danovi, PhD
Senior Editor, Nature Genetics
ORCID: 0009-0007-7822-5479

Reviewers' Comments:

Reviewer #3:

Remarks to the Author:

The Strand- and CITE-seq integration method used in the current paper relies on copy number inference from each modality separately, followed by clone inference and assignment (CNA-based approach). In contrast, the integration method in Grimes et al. relies on matching nucleosome occupancy patterns at marker genes (Strand-Seq) to their expression in scRNA (NO-based approach). As such, the benchmarks in Grimes et al. and Jeong et al. do not provide insights into the performance of the approach used here. Fig 5i-k from Grimes et al. only shows the results of CNA-based integration without a measure of reliability beyond mere similarity of clonal frequencies between Strand-Seq and scRNA. In fact, 17q-Del containing cells remain unresolved in the CNA-based approach.

That being said, this reviewer appreciates the conservative approach of not making assignments, e.g. in the 17-Del case above and CK295 in the current manuscript, where CNA-based matching seems to fail entirely. In addition, the co-occurrence analysis in Grimes et al., Supplementary Figure 26, does provide some degree of performance measure, even if only for a single case.

The new Extended Data Fig. 8c further also helps assuage this reviewer. It would have been nice had the authors included these side-by-side plots for all patients, as opposed to a single cherry-picked example. This would help readers gauge the robustness of assignments across cases profiled with Strand-seq plus CITE-/scRNA-seq (importantly also for the CK295 case where assignments fails) and add confidence the technique is generally applicable.

Finally, this reviewer fully agrees with the authors that integrating Strand-Seq with CITE-/scRNA-seq can be highly informative and is of general interest, as evident from the findings presented here and in previous work. Supporting its importance is also precisely why an assessment of integration performance, robustness, and pitfalls would have been helpful.

Version 4:

Decision Letter:

Our ref: NG-A62852R3

2nd Aug 2024

Dear Dr Trumpp,

Thank you for submitting your revised manuscript "Single-cell analysis reveals dynamic clonal evolution and targetable phenotypes in complex karyotype AML" (NG-A62852R3). It has now been seen by the original referees and their comments are below. The reviewers find that the paper has improved in revision, and therefore we'll be happy in principle to publish it in Nature Genetics, pending minor revisions to satisfy the referees' final requests and to comply with our editorial and formatting guidelines.

Sincerely,

Safia Danovi, PhD
Senior Editor, Nature Genetics
ORCID: 0009-0007-7822-5479

Reviewer #3 (Remarks to the Author):

Many thanks to the authors for including the full set of side-by-side plots as Supplementary Figure 2 and elaborating on the assignments. My apologies for being a pain about this. I do think the additions show how tricky some assignments can be in all fairness (e.g. D1922 SC2) and hope this provides confidence to people looking to apply this powerful approach to their own samples.

Version 5:

Decision Letter:

In reply please quote: NG-A62852R4 Trumpp

15th Oct 2024

Dear Dr Trumpp,

I am delighted to say that your manuscript "Single-cell multi-omics analysis reveals dynamic clonal evolution and targetable phenotypes in acute myeloid leukemia with complex karyotype" has been accepted for publication in an upcoming issue of Nature Genetics.

Your paper will be published online after we receive your corrections and will appear in print in the next available issue. You can find out your date of online publication by contacting the Nature Press Office (press@nature.com) after sending your e-proof corrections.

Please note that *Nature Genetics* is a Transformative Journal (TJ). Authors may publish their research with us through the traditional subscription access route or make their paper immediately open access through payment of an article-processing charge (APC). Authors will not be required to make a final decision about access to their article until it has been accepted. [Find out more about Transformative Journals](https://www.springernature.com/gp/open-research/transformative-journals)

Authors may need to take specific actions to achieve [compliance](https://www.springernature.com/gp/open-research/funding/policy-compliance-faqs) with funder and institutional open access mandates. If your research is supported by a funder that requires immediate open access (e.g. according to [Plan S principles](https://www.springernature.com/gp/open-research/plan-s-compliance)) then you should select the gold OA route, and we will direct you to the compliant route where possible. For authors selecting the subscription publication route, the journal's standard licensing terms will need to be accepted, including [those licensing terms](https://www.nature.com/nature-portfolio/editorial-policies/self-archiving-and-license-to-publish) will supersede any other terms that the author or any third party may assert apply to any version of the manuscript.

If you have not already done so, we strongly recommend that you upload the step-by-step protocols used in this manuscript to protocols.io. protocols.io is an open online resource that allows researchers to share their detailed experimental know-how. All uploaded protocols are made freely available and are assigned DOIs for ease of citation. Protocols can be linked to any publications in which they are used and will be linked to from your article. You can also establish a dedicated workspace to collect all your lab Protocols. By uploading your Protocols to protocols.io, you are enabling researchers to more readily reproduce or adapt the methodology you use, as well as increasing the visibility of your protocols and papers. Upload your Protocols at <https://protocols.io>. Further information can be found at <https://www.protocols.io/help/publish-articles>.

Sincerely,

Safia Danovi, PhD
Senior Editor, Nature Genetics
ORCID: 0009-0007-7822-5479

Click here if you would like to recommend Nature Genetics to your librarian
<http://www.nature.com/subscriptions/recommend.html#forms>

** Visit the Springer Nature Editorial and Publishing website at http://editorial-jobs.springernature.com?utm_source=ejp_NGen_email&utm_medium=ejp_NGen_email&utm_campaign=ejp_NGen for more information about our career opportunities. If you have any questions please click [here](mailto:editorial.publishing.jobs@springernature.com). **

Point-by-point response to the comments made by the three reviewers to our manuscript by Leppä*, Grimes*, Jeong* et al. Korbels and Trumpp§ (NG-A62852):

First of all, we would like to thank all three reviewers for their thoughtful comments, helpful suggestions and also constructive concerns regarding the limitations of our study. We were pleased to obtain positive comments such as: (1) a detailed and well-written study representing the most comprehensive genomic analysis conducted on CK-AML to date, (2) findings are novel and important for understanding this particularly poor-risk subset of AML, (3) novel observations about clonality of structural variants and chromosomal instability in AML, (4) unique and intriguing insights into the evolution and intratumor genetic and phenotypic heterogeneity of CK-AML, and (5) this study provides a blueprint for future (larger) studies into the interplay between genome and phenotype during tumor evolution.

However, we also took the overall criticism very seriously, particularly regarding the limited number of analyzed patient samples by scNOVA-CITE and aspects of the drug screening data using primary CK-AML cells. Guided by the reviewer comments and discussions with the editor, we undertook a major effort to more than double the analyzed CK-AML samples, increasing the total number of individual leukemia cells to over 42,000 (details below). Facilitated by the acquisition of additional novel and unique datasets, we extensively revised and restructured the manuscript. The extended and, in our view, much improved manuscript provides a series of unique insights into the dynamic genomic, transcriptomic, and functional complexity of individual CK-AML cells directly derived from patients, significantly expanding and strengthening our original conclusions.

Please find below the key improvements and additional findings we implemented in response to the two most critical aspects: the expansion of patient sample analysis and the refinement of the drug response studies:

- (1) One of the major concerns raised by more than one reviewer was the limited number of primary CK-AML samples analyzed by scNOVA-CITE and the number of PDX models. As outlined in new Figure 1a (see also below), we have now significantly increased our cohort size by adding 4 additional patients, effectively doubling the number of CK-AMLs at initial sampling ($n=8$). In addition, we established three new CK-AML PDX models, to increase the genomic, molecular and functional characterization to a total of 5 primary patient sample-PDX pairs, identifying two clonal evolution patterns during CK-AML reconstitution in mice (new Figure 4). Finally, we also embarked on a novel longitudinal scNOVA-CITE analysis for two patients, utilizing pre- and post-treatment patient sample pairs (new Figure 6). With the addition of these new samples, we have analyzed 2.3x more single-cell genomes than in the initial manuscript (364 vs 855 Strand-seq cells) and 2.8x more single-cell transcriptomes (14,941 vs 41,618 CITE-seq cells), making this to our knowledge the largest combined single cell genome and transcriptome dataset generated in the entire leukemia field to date.

Rebuttal Figure. Schematic study layout of single-cell multi-omics profiling with scNOVA and CITE-seq, applied to eight primary CK-AML patient samples at initial sampling, five matching patient-derived xenografts and two matching refractory/relapse samples.

- (2) The second overarching concern centered around the extent of our drug screening efforts in the original manuscript. A significant goal of our study was to showcase how single-cell multi-omics data could be utilized to investigate the drug-response profiles of various genetic subclones *ex vivo* – a capability we have now successfully demonstrated (see revised Figure 5). From a technical standpoint, it's important to highlight to the reviewers that the availability of primary viably frozen CK-AML patient material severely limits the feasibility of conducting large drug screens, which typically require >7 million viable leukemia cells per screen. Furthermore, drug screens are typically conducted using established homogeneous leukemic cell lines or bulk primary samples, which fail to reflect the heterogeneity observed in primary CK-AML cells, thus potentially overlooking clinically relevant responses. Importantly, to our knowledge, the differential response between different AML subclones within the same sample has never been explored or reported.

To further strengthen the drug profiling part, we performed additional *ex vivo* drug screenings in two new diagnostic samples (HIAML47 and P9D) and one PDX from the initial cohort (PDX-CK349). With the addition of the new samples, we were able to better compare the patient's clinical response with the *ex vivo* response of the corresponding sample. In the original manuscript we already reported a distinct resistance exclusively in the engraftment-driving LSCs of patient CK349 to cytarabine and daunorubicin – the same chemotherapy regime the patient received as first-line treatment. In the revised manuscript, we additionally found that the newly added patients HIAML47 and P9D were resistant to venetoclax treatment *ex vivo*, mimicking their clinical behavior (new Figures 5 and 6).

We also included two additional inhibitors to our drug screen, revumenib (Menin-inhibitor) and elesclomol. While the two major subclones from the newly included patient P9D were highly resistant to venetoclax *ex vivo* (new Figure 6l-n, new Extended Data Figure 14h), we found that both of these subclones showed good sensitivity to elesclomol (new Figure 6p-q). Moreover, cells from PDX-CK349 that were resistant to all other tested drugs also showed considerable efficacy to elesclomol (see new Extended Data Figure 12k). We found elesclomol to be relatively toxic for healthy HSPCs, which may restrict its clinical utility. Yet, its effectiveness against otherwise pan-resistant CK-AMLs may justify further preclinical and clinical investigation. Collectively, through the addition of new samples and two new inhibitors, we have significantly broadened and reinforced our drug screening data. Altogether the revised study showcases how single-cell multi-omics can be employed to identify and target different subclones within the same sample.

- (3) The inclusion of longitudinal scNOVA-CITE analysis for two patients, utilizing pre- and post-treatment patient sample pairs revealed novel and intriguing results (new Figure 6). In one patient, the post-treatment leukemia was driven by a chromothripsis event at chromosome 6 specific to one subclone. This appears to have generated CK-AML cells with increased stemness and a steady increase in cell growth and oxidative phosphorylation driving clonal disease progression. In the second CK-AML patient, the refractory cells of one subclone had gained a novel 5Mb focal deletion on chromosome 17q, encompassing the tumor suppressor gene *NFI*, known to cause RAS pathway activation, which has been linked to venetoclax resistance. Moreover, the other resistant subclone showed a subclone-specific shift towards the Megakaryocyte-Erythroid Progenitor (MEP) lineage, also associated with venetoclax resistance. In summary, these new data show that genetic evolution in conjunction with cell type plasticity were associated with disease progression in both of the newly added patients (new Figure 6).
- (4) With the expanded cohort size, we reaffirm that our high-resolution data combining single-cell genomics with single-cell transcriptomics provide new insights into the subclonal patterns and uncover an even higher intra-tumor complexity than previously reported for

CK-AML. Furthermore, our results provide, to our knowledge, the first single-cell level insights into the clonal evolution fates and patterns during CK-AML reconstitution in mice and during disease progression in patients. Our data strongly suggest that each cell within CK-AML has distinct characteristics and may be unique at the genetic/epigenetic level. This leads us to hypothesize that there may also be an extremely high degree of heterogeneity between CK-AML patients. This hypothesis can only be rigorously tested by in-depth analysis of large cohorts with hundreds of samples, for which we present now a conceptual framework.

- (5) Beyond the examples outlined above, we have addressed all comments and concerns by all three reviewers in full, many with new additional data. Please find below our point-by-point response to each of the raised comments.

Point-by-Point response to the three Referees' comments:

Response to the comments made by Reviewer 1:

Reviewer #1 (expertise: AML, leukemia evolution, single-cell analysis):

Remarks to the Author:

In the article from Trumpp and colleagues, the authors use single cell methods to determine the intra-tumoral heterogeneity in complex karyotype AML and link this to leukemia stem cells and therapeutic sensitivity. The authors use various genotyping methods to determine the structural variants present at the single cell level. They deeply analyze four cases of CK-AML demonstrating a complex collection of clonally related subclones with evidence of multiple genetic mechanisms for the complex karyotype including chromothripsis, extrachromosomal circular DNA elements, inversions, deletions, and others. Using single cell methods of scNOVA and CITE-Seq, they demonstrate differences in epigenetic programs and some minor, but primarily overlapping, surface immunophenotypes. Notably, xenograft studies show that in 2 of 2 cases, only a single clone engrafts in multiple mice, and that clone had some increased measures of stem cell programs, although not markedly different from non-engrafting clones. They show that the xenograft engrafting clone in CK349 is responsible for relapse in the patient treated with several agents including allogeneic transplant. Finally, they explore potential drug sensitivity of two cases, implicating chemo resistance preferentially in SC3 in CK349, the clone responsible for the clinical relapse. In case CK282, they showed BCL-xL dependence and inhibitor sensitivity in the stem-like and engrafting SC2 subclone.

The study and characterization of CK-AML is of great interest to the field. The authors have conducted the most detailed genomic analysis of such cases to date, as well as the first single cell level analysis to dissect subclones and epigenetic programs. Many of the findings are novel and important for understanding this particularly poor-risk subset of AML. The paper is detailed and well-written.

Our response: We thank this reviewer for providing such positive feedback, especially for recognizing that our study is well-written representing the most comprehensive genomic analysis conducted on CK-AML to date. The acknowledgments of its novelty and the broad relevance of our research is truly appreciated.

However, the engraftment characterization and link to relapse is conducted with only 2 samples and 1 sample, respectively, making it difficult to generate conclusions with broader implications for CK-AML.

Our Response: We thank the reviewer for these constructive comments, which motivated us to substantially increase the number of analyzed CK-AMLs. As discussed above and shown in the new Figure 1a, we have significantly expanded our cohort size by adding 4 additional patients, doubling the number of CK-AML cases collected at diagnosis/salvage stage (n=8) (revised Figure 1 and new Figure 2). In addition, we established three new CK-AML PDX models, to increase the genomic, molecular and functional characterization to a total of 5 primary patient sample-PDX pairs (new Figure 4). Using this primary patient sample-PDX pair cohort, we now identified two clonal evolution patterns during CK-AML reconstitution in mice. In 2 of 5 patients, the engrafted cells showed similar SVs to the dominant clone in the patients with only few additional SVs in individual cells (new Figure 4d). In 3 of 5 patients, minor subclones showed superior engraftment capabilities and demonstrated ongoing generation of karyotype diversity (new Figure 4e). Collectively, with the 5 PDXs we now demonstrate that superior engraftment in CK-AML can originate from LSCs within dominant subclones as well as from LSCs within minor subclones.

At the molecular level, we now show that the engraftment-driving LSCs shared transcriptomic programs involved in cell growth, proliferation and oxidative phosphorylation, whereas non-engrafting cells were enriched in gene sets associated with inflammation (new Figure 4k). Finally, as most engraftment studies have focused on assessing subclonal architecture upon transplantation based on single nucleotide variants (SNVs) (Klco et al., 2014; Morita et al., 2020; Shlush et al., 2017), our study adds intriguing insight into the subclonal repertoire of LSCs based on SVs.

Regarding the link between engraftment and relapse, we unfortunately lacked the material to perform this analysis in more samples. Others have shown that engrafted cells are skewed towards relapsed AML clones in PDX models generated from diagnosis AML cells (Kawashima et al., 2022; Shlush et al., 2017). To avoid broad generalization based on one sample, we have toned down the text, and indicated the potential of engrafting cells to also drive disease progression, as already shown by others. Lines 451-454: “[...] we identified different clonal evolution fates and patterns during CK-AML reconstitution in mice [...]. Our data further indicate that the PDX engraftment-driving subclone can also drive relapse outgrowth in CK-AML patients, as in the case of CK349.”

Despite our inability to further investigate the connection between engraftment and relapse, we embarked on a completely novel longitudinal scNOVA-CITE analysis for two newly added patients (P5 and P9), utilizing pre- and post-treatment sample pairs, as described above (new Figure 6). This allowed us to exemplify the biological and clinical relevance of single-cell clonal evolution analysis beyond PDX models. In patient P5, post-treatment relapse was driven by a chromothripsis event at chromosome 6 in subclone 1. This generated CK-AML cells with increased stemness and a steady increase in cell growth and oxidative phosphorylation driving clonal disease progression. In the other CK-AML patient, two treatment resistant subclones persisted. One resistant subclone had gained a novel 5Mb focal deletion on chromosome 17q, encompassing the tumor suppressor gene *NFI* (new Figure 6h), known to cause RAS pathway activation. The other subclone showed a shift towards Megakaryocyte-Erythroid Progenitor (MEP) lineage. Both RAS pathway activation and lineage plasticity have been linked to venetoclax resistance (Bandyopadhyay et al., 2024; Zhang et al., 2022). In summary, with these new data we could show that genetic evolution as well as cell type plasticity were associated with disease progression in both of our longitudinally assessed patients (new Figure 6).

Moreover, the pharmacologic studies are very limited in significance and conclusions.

Our response: One important aim in our study was to show that we can leverage the single-cell multi-omics data to study the drug-response profiles of different genetic subclones *ex vivo* - which we were able to successfully do (revised Figure 5). As discussed above, we performed additional *ex vivo* drug screenings in two new diagnostic samples (HIAML47 and P9D) and one PDX from the initial cohort (PDX-CK349) to further strengthen the drug profiling part. With the addition of the new samples, we were able to better compare the patient’s clinical response with the *ex vivo* response of the corresponding sample. In the original manuscript we already reported that in patient CK349, we observed a distinct resistance exclusively in the engraftment-driving cells to cytarabine and daunorubicin – the same chemotherapy regime the patient received as first-line treatment. In the revised manuscript, we additionally found that the newly added patients HIAML47 and P9D were resistant to venetoclax treatment *ex vivo* as well as in the clinic (revised Figure 5 and new Figure 6).

We also included two additional inhibitors to our drug screen, revumenib and elesclomol. Revumenib is a Menin inhibitor that we hypothesized could show efficacy as CK-AMLs have a high frequency of SVs affecting 11q and overlapping *KMT2A* (Rücker et al., 2012). Elesclomol induces cellular death presumably through an increase in cellular oxidation, and has shown increased efficacy *in vitro* in TP53-rearranged AMLs (Nechiporuk et al., 2019). Although revumenib exhibited no effect in the tested samples, elesclomol demonstrated notable efficacy in P9D. Cells from the two major subclones (SC1- and SC3-enriched cells) from P9D were highly resistant to venetoclax monotherapy as well as combination therapy with azacytidine *ex vivo* (new Figure 6l-n, new Extended Data Figure 14h), mimicking the clinical response of this patient. However, we found that both of these venetoclax resistant subclones showed good sensitivity to elesclomol (see Figure below and new Figure 6p-q). Moreover, cells from PDX-CK349 that were resistant to all other tested drugs also showed considerable efficacy to elesclomol (new Extended Data Figure 12k). However, as elesclomol was relatively toxic also in healthy HSPCs, this may restrict its clinical utility (see Figure below). Yet, its effectiveness against otherwise pan-resistant CK-AMLs may justify further preclinical and clinical investigation.

Regrettably, due to constraints posed by the availability of primary viably frozen patient material, conducting drug screens involving more than the 12 treatment conditions (7.5 million cells per screen) was deemed unfeasible. However, in general, we agree with the reviewer that larger screening efforts

using primary samples are needed to identify treatment options for CK-AMLs with better efficacy. We have now included this statement in the manuscript. Lines 502-503: “[...] providing a proof-of-concept method for a larger screening to discover novel treatment options, such as BCL-xL inhibition, for this subgroup of AML with dismal outcome.” Collectively, through the analysis of a larger number of primary and paired samples in the revised manuscript, we have significantly broadened and reinforced our dataset. Altogether the revised study showcases the biological and clinical relevance of single-cell multi-omics analysis in CK-AML.

Rebuttal Figure R1.1. a) Viabilities (fraction of viable cells compared to untreated control) of different populations in healthy CD34+ hematopoietic stem progenitor cells (HSPCs), one AML PDX, and two primary AML samples after 24h *ex vivo* exposure with indicated concentrations of elesclomol. b) Viabilities (fraction of viable cells compared to untreated control) of CK349 PDX cells after 24h *ex vivo* exposure indicated treatments.

1. There are only 278 total cells analyzed for the genotyping which is very small ranging from 49-91 per sample and with only 1.2x genome coverage. This leaves a lot of potential for missing clones and significant inability to call SVs.

Our response: The reviewer raises a fair point. For the revised manuscript, we have substantially increased our cohort size and have now a total of 855 cells analyzed for the genotyping. While Strand-seq allows single cell genome analysis at a large scale compared to other single cell technologies in the field, this comes with the limitation that rearrangement events are detected with 200kb resolution (Sanders et al., 2017; Sanders et al., 2020). Therefore, we focused our study on SVs that are >200kb in size. Aside from that, the ability of scTRIP to identify SVs from Strand-seq data has been evaluated by several means by us in Sanders et al. (2020, Nature Biotechnology), including specificity, sensitivity, and its performance at varying subclone frequencies. In this benchmark paper for scTRIP we also demonstrated that scTRIP enables the study of larger SVs with much less coverage than whole genome amplification-based methods. Since then, we have further shown in two additional studies that scTRIP can reliably identify SVs using hundreds of cells (1. Jeong et al. 2022, Nature Biotechnology – work pursued in cell lines as well as T-cell acute lymphoblastic leukemia; 2. Grimes et al. 2024, Nature Genetics (accepted), in which we examined somatic mosaicism in CD34+ cells from normal individuals).

Regarding missing clones, we agree that plate-based methods have an inherent limitation when it comes to cell numbers. Thus, we have included the following in the discussion:

Lines 581-584 “[...] plate-based single-cell methods have an inherent limitation when it comes to cell numbers (Griffiths et al., 2018) [...] Thus, the extent of genetic heterogeneity reported in this study could be underestimate as small subclones may have been missed [...]” For the Reviewer, we have also computed the theoretical probability of observing a clone with our cell numbers using a binomial distribution model. As shown in the figure below, we have a 50% chance of detecting a clone making 1.4% of the original sample when 50 cells are analyzed. When 90 cells are analyzed the probability to detect clones of this size increases to 72%. This shows that with our cell numbers, we may be missing small subclones that make <1.5 % of the original sample, while minor subclones present with higher clonal frequencies have a high chance to be captured through our approach.

That being said, we also used our CITE-seq data to identify potentially missing subclones as it contains many more cells (total of 41,524 cells analyzed in the revised study). For this, we performed somatic copy-number alteration (SCNA) analysis using InferCNV. Compared to CONICSmatrix, which we used for the SCNA re-calling from the CITE-seq data in the manuscript, InferCNV analysis performs SCNA calling by exploring the expression intensity of genes across positions of the entire tumor genome without the user having to specify SV breakpoint coordinates (<https://github.com/broadinstitute/inferCNV>). This enabled us to search for subclones with a set of SVs not detected in the Strand-seq data. As shown below for two samples, while the relative expression intensities across each chromosome analyzed by CITE-seq are more noisy than the Strand-seq data, we were able to re-discover the subclone-defining SVs detected in Strand-seq data when applying inferCNV. These results, furthermore, showed high concordance with SCNA re-calling in CITE-seq data when using the CONICSmatrix tool. Finally, we found no evidence of additional subclones when analyzing the CITE-seq based expression intensities in the single-cell dataset. This response should hopefully clarify the reviewer's concerns regarding the omission of potential clones and the performance of SV calling in our single cell data.

Rebuttal Figure R1.2. Theoretical probability of observing a clone with different cell numbers using a binomial distribution model. a) With 50 cells the probability to detect a clone making 1.4% of the original sample is 50%. b) With 90 cells the probability to detect a clone making 1.4% of the original sample is 72%.

Rebuttal Figure R1.3. Inference of somatic copy number alterations (SCNAs) using CITE-seq data from CK349 and P9D. a) InferCNV analysis of 3,514 high quality CK349 AML cells, and 267 control cells (cells sequenced by CITE-seq not originating from the myeloid lineage), profiled by CITE-seq. No additional SVs were detected on top of the SVs discovered by Strand-seq. b) CONICSmat-based targeted SCNA recalling of the subclone-defining SVs in CK349 using the high-resolution breakpoints derived from Strand-seq. Use of these SV breakpoints allowed CONICSmat to confidently call SC1 in 2556, SC2 in 163 and SC3 in 270 single-cells from the CITE-seq data. c) InferCNV analysis of 4,575 high quality P9D AML cells, and 673 control cells (cells sequenced by CITE-seq not originating from the myeloid lineage), profiled by CITE-seq. No additional SVs were detected on top of the SVs discovered by Strand-seq. d) CONICSmat-based targeted SCNA recalling of the subclone-defining SVs in P9D using the high-resolution breakpoints derived from Strand-seq. Use of these SV breakpoints allowed CONICSmat to confidently call SC1 in 681, SC2 in 183 and SC3 in 3133 single-cells from the CITE-seq data. Unassigned cells in CK349 and P9D fell below the posterior probability cutoff of 0.8 used for a confident SCNA assignment. (Note that the high variability observed on the 6p-arm in both InferCNV analyses, likely arose from the presence of MHC genes in this locus, as discussed at the InferCNV github vignette).

2. The authors do a good job of performing orthogonal experiments to validate observations made from their single cell SV analyses. Can the authors **comment** on how much additional information is lost or gained with scTRIP compared to bulk analyses? Although the SV resolution at the single cell level is enhanced, is the sensitivity of detecting SVs at a global level similar to bulk analysis methods? I would

imagine that fewer SVs are detected using the single cell method compared to bulk techniques. Can the authors perform this comparison? Further, the authors used data derived from the Strand-seq data as input for their CNA recalling in the CITE-seq data. What is the value, if any, of using this single cell approach rather than break points observed from bulk tissue assays as input to CONICSmatrix? Would you still be able to confidently call subclones in the CITE-seq data without scTRIP?

Our response: We are pleased to comment on these important issues. As discussed above, the ability of scTRIP to identify SVs from Strand-seq data has previously been carefully benchmarked and evaluated by several orthogonal means by us in Sanders et al. (2020, Nature Biotechnology), including with respect to specificity and sensitivity. scTRIP is limited to the detection of SVs larger than 200kb in size, so compared to bulk analyses, small SVs, SNPs and indels are missed. This we have now acknowledged in the discussion:

Lines 582-583: “[...] Strand-seq only captures SVs of at least 200 kb in size. Thus, the extent of genetic heterogeneity reported in this study could still be an underestimate [...]”. We note that we previously showed based on analyzing the Pan-Cancer Analysis of Whole Genomes resource that 80% of all SV drivers in cancer genomes are at least 200 kb in size, meaning we are likely to capture the majority of functionally relevant SVs using scTRIP (Jeong et al., 2023).

Furthermore, when compared to single cell sequencing, standard 30x–50x WGS sequencing is not able to capture both low-frequency variants and clonal complexity, as demonstrated in several prior works (such as by Griffith et al. (2015, Cell Systems). When focusing on SV calling, cell fractions >30% are generally thought to be amenable to bulk WGS-based SV calling (Layer et al., 2014), whereas scTRIP can detect SVs at cell fraction levels significantly below that – even cell fractions of around 1.5% (Sanders et al., 2020). As discussed above, our approach may not detect extremely small subclones with a clonal fraction (CF) less than 1.5%. However, SV subclones with a CF ranging from 2% to 10% are effectively identifiable using our technique, whereas their detection is practically unattainable with bulk tumor sequencing.

As evident in the new Figure 2, all samples with polyclonal growth (n = 6 samples) harbored SV subclones that made <30% of the cells, clonal fractions thought to be challenging to detect from bulk sequencing (Layer et al., 2014). These polyclonal samples made 75% of our cohort (6/8 samples), highlighting that the ability to detect subclones is important for the majority of CK-AML samples – and therefore arguing for the use of Strand-seq rather than bulk WGS. Moreover, different CK-AML subclones within the same sample can have different sets of SVs, evident in particular in the samples with branched polyclonal growth (new Figure 2e). For example, in patient CK349, SC3 (CF = 5.5%) that harbored the seismic amplification at chromosome 11 did not carry the chromosome 8 trisomy present in the majority of the cells (SC1; CF = 89%) (new Figure 2e). Had we relied on bulk WGS for detecting structural variants, the spectrum of chromosome 11 rearrangements would have likely remained obscured. Additionally, it would have been virtually impossible to determine the pattern of co-occurrence of chromosome 11 rearrangements and chromosome 8 trisomy within the same cell populations. The driving force behind the clonal relapse in patient CK349 was SC3 (new Figure 2e), underscoring the necessity of differentiating among subclones for biological and clinical insights, which in this case would have not been achievable through bulk WGS analysis.

Regarding the sensitivity of detecting SVs at a global level, Sanders et al. (2020, Nature Biotechnology) compared scTRIP SV calling to WGS SV calling. scTRIP SV calling was able to confirm 100% and 89% of the tested SVs present by WGS. Jeong et al. (2022, Nature Biotechnology) also verified all somatic SVs detected in a lymphoblastoid cell line using scTRIP with deep WGS data.

As for the value added by scTRIP to call subclones from CITE-seq data compared to bulk assays, we want to raise two important points. 1) scTRIP enabled us to identify SVs present in small subclones that are difficult to call from bulk WGS-based data. 2) scTRIP allowed us to reliably assess which SVs co-occurred in one subclone and were absent in other subclones. This was especially relevant in the samples with branched polyclonal growth where bulk WGS-based assays can only statistically infer subclonal structures. Both of these points were important for the SCNA re-calling from CITE-seq data as we used the information of several co-occurring SVs to re-call small and large subclones. To further illustrate

the value added by scTRIP to call subclones from CITE-seq data, we performed CNA re-calling from CITE-seq data for patient CK349 using SV breakpoints from all SVs (excluding singleton events; same as in the manuscript) and compared this to CNA re-calling using only SV break points from SVs present in >10% of cells – representing resolution amenable (or even higher) to bulk WGS-based SV calling. As evident in the figure below, using SV breakpoints from SVs present in >10% of cells we would not have been able to detect subclones SC2 and SC3 in patient CK349. We hope that these explanations clarify the advance that scTRIP analysis provides over bulk WGS in yielding much more reliable detection of SV subclones.

Rebuttal Figure R1.4. CONICSmatrix-based targeted SCNA re-calling of the subclone-defining SVs in CK349 using the high-resolution breakpoints derived from Strand-seq. a) Re-calling using SVs present in all cells enabled the annotation of SC2 and SC3 cells. b) Re-calling using SVs present in >10% of cells only enabled confident annotation of SC1 cells.

3. In Fig 4 (line 362-364, 378-381), the authors demonstrate that the different subclones deconvolve differently into normal HSPC states with some subclones associated with more immature and stem cell-like states. They suggest that the genomic aberrations may be causal in the increased measures of stemness, but this correlation does not necessarily imply causation. It is also possible that the genomic aberrations of these clones occur within a more immature/stem compartment that is then retained in the eventual leukemic clone.

Our response:

We thank the reviewer for this notion, and agree that our data does not exclude the possibility that immature cell state-enriched subclones may have simply evolved from a more primitive cell. We have substantially revised Figure 4, and with the addition of the new samples, we observe a marked correlation (Spearman's $R = 0.77$, $p = 0.00018$) with stemness and SV burden (new Extended Data Fig. 10c). But as we cannot clearly establish cause, we have rephrased the text and removed notions about causation and focused on the relationship between these variables.

Lines 399-400: “[...] high leukemic stem cell (LSC) scores generally showed association with primitive cell types and high SV burden [...]”

4. In the PDX engrafted cells, were new SVs or SNVs identified that suggested ongoing clonal evolution as the cells engrafted and proliferated in the mice? This might be expected as the engrafting closed demonstrated high heterogeneity and instability.

Our response:

We thank the reviewer for this interesting question. Ongoing evolution was observed in the PDXs with individual cells harboring additional SVs. This was also observed already in the primary samples with singleton cells having up to two rearrangements on top of the subclone-specific SVs. The ongoing generation of karyotype diversity has been well described in advanced cancers using large patient cohorts (Priestley et al., 2019; Watkins et al., 2020). Bollen et al. (2021, Nature Genetics) showed that in colorectal cancer patient-derived organoids, karyotype alterations are ongoing and prevalent. However, most *de novo* karyotypes are subject to negative selection and only some show increased fitness (Bollen et al., 2021). This broadly aligns with our data, where in 4 out of 5 PDXs we did not see an outgrowth of a new subclone (new Figure 4d-e). Only in one PDX (PDX-CK397), we detected two distinct subclones with the dominant clone in the PDX stemming from a subclone where only one cell at diagnosis resembled the cells forming this major clone in the PDX (new Figure 4e), reflecting previously undetected clonal evolution. While most PDXs did not reveal new subclones, overall PDXs showed increased SV burden compared to the primary patient samples (new Figure 4f), largely due to the subclones with higher SV burdens driving engraftment. PDXs also harbored singleton SVs present in individual cells and two samples were accompanied by unstable chromosomes that did not stabilize in the PDX (new Extended Data Fig. 11a-d), demonstrating that engraftment can be accompanied by spontaneous generation of *de novo* karyotype diversity. We have included this observation in the text.

The chromosome 11 rearrangements in patient CK349 is a particularly remarkable example of ongoing clonal evolution. At diagnosis in patient CK349, we detected step-wise acquisition of amplifications with cell-specific CN statuses on chromosome 11 with two cells showing intermediate levels of amplification mostly localized to the q arm as well as three cells with excessive and more widespread amplifications spanning both p and q arms (new Figure 2f, new Extended Data Fig. 7a-b). In the PDX, all cells also showed excessive and more widespread amplifications spanning both arms but these were localized more to the terminal ends of the arms (new Figure 4k – previously 5c). Similar to the diagnosis sample, the PDX cells also showed amplifications with cell-specific CN statuses, making almost each cell in this specimen unique.

5. In Fig 6, the authors suggest that the CK349 SC3 clone responsible for engraftment and clinical relapse is more sensitive to venetoclax. However, CK-AML and TP53-mutant AML generally speaking is known to be poorly responsive to venetoclax and HMA-containing regimens. What is the venetoclax sensitivity of CK-AML from the BeatAML cohort? Is this case an outlier?

Our response: We thank the reviewer for this comment. As shown in the figure below, in the BEAT AML cohort TP53 mutations or CK-AML status (defined as ≥ 3 SVs) do not result in markedly higher *ex vivo* venetoclax resistance. Importantly, the *ex vivo* drug screening in the BEAT AML study was done using bulk mononuclear cells, whereas we have previously shown the importance of assessing *ex vivo* sensitivities in disease-driving LSCs (Waclawiczek et al., 2023). This could explain the discordance between the BEAT AML *ex vivo* results and poor clinical response to venetoclax/HMA often observed in CK-AML patients. While comparing our sensitivities to the BEAT AML cohort would have been very interesting, unfortunately this was not possible as the cell viability assessment of our *ex vivo* screen was done after 24h, whereas in the BEAT AML study this was done after 72h, resulting in markedly lower IC50 values in the BEAT AML cohort.

Regarding the sensitivity of CK349-SC3 to venetoclax, as originally stated in the manuscript, the combination treatment of venetoclax and azacitidine was the most effective in CK349-SC3 compared to the other subclones in the same samples. However, for the revised manuscript we have included two additional patients (HIAML47 and P9) that were treated with venetoclax and azacitidine as first-line treatment (new Figure 5 and 6). This made it possible to compare the *ex vivo* venetoclax sensitivity of CK349-SC3 with sensitivities from two patients with known clinical response to venetoclax and azacitidine. Out of these two patients, patient HIAML47 showed only a short partial clinical response whereas patient P9 was refractory to the treatment. Compared to the cells from these two patient samples, SC3 cells from CK349 showed only slightly better efficacy to venetoclax monotherapy *ex vivo* (see figure below). Strikingly, CK349-SC3 and leukemic cells from HIAML47 (SC2 and SC3) showed considerable response to azacitidine alone which was also reflected in the response to the combination

treatment with azacitidine and venetoclax. While the combination treatment of venetoclax and azacitidine was the most effective in CK349-SC3 compared to the other subclones in the same sample, the killing efficacy was not better than in HIAML47. Thus, we have revised the text and removed the suggestion that this clone is likely to be sensitive to venetoclax. While we tested additional inhibitors in the PDX cells from CK349, only elesclomol showed some effect highlighting that a larger screening is needed to find alternative treatment options for this patient (see Figure R1.1 above).

Rebuttal Figure R1.5. a) The association of venetoclax *ex vivo* response with mutations from the BEAT AML data (Tyner et al., 2018). Increased sensitivity is indicated by red, increased resistance indicated by blue as determined by the sign of the Glass's Delta effect size (X-axis). TP53 mutations were not significantly associated with venetoclax resistance *ex vivo*. b) Area under the curve for venetoclax sensitivity between CK and non-CK AMLs. No difference between the groups was detected. c) Viabilities (fraction of viable cells compared to untreated control) of different populations in CK349, HIAML47 and P9D after 24h *ex vivo* exposure with indicated concentrations of venetoclax. d) Viabilities (fraction of viable cells compared to untreated control) of different populations in CK349, HIAML47 and P9D after 24h *ex vivo* exposure with indicated concentrations of azacitidine and venetoclax. All samples were largely resistant to venetoclax monotherapy whereas CK349 and HIAML47 showed moderate response to the combination therapy, likely due to azacitidine sensitivity.

6. In Fig 6, the pharmacologic analysis of CK282 is questionable. Review of the data (panel 6H, and S14c) suggests only minor differences in BCL-xL expression between the SC2 clone and the remainder of the leukemic blasts. Are the authors proposing that this difference drives preferential susceptibility of SC2 to BCL-xL inhibition? Is there specificity for SC2 clone, as the heat map in panel 6g suggest all clones are similarly susceptible. This data detracts from the main genetic studies in the manuscript and could be removed.

Our response:

We apologize for the unclear phrasing of the findings. Indeed, all subpopulations showed good response to BCL-xL inhibition with SC2-enriched cells showing the strongest effect, resulting in an almost complete eradication of these cells at 35nM and 100nM A-1331852 concentrations (revised Figure 5j-m, new Extended Data Fig. 13a-b). Based on protein expression, BCL-xL was also most highly expressed in the SC2-enriched cells (new Figure 5n). We do not mean to claim that only SC2-enriched cells are sensitive to BCL-xL inhibition, but rather that the engraftment-driving SC2 cells seem to be particularly sensitive to BCL-xL inhibition. We have added a line plot to new Figure 5l and Extended Data Fig. 13b to better show the sensitivity of the different subpopulations where it becomes more apparent that the SC2-enriched cells are the most sensitive.

Regarding the comment about the drug screening detracting from the main genetic studies, we respectfully disagree with this reviewer. While the genetic studies make a core of the story, the ability to assess the phenotype and function of genetically defined subclones is highly important. As CK-AMLs can consist of genetically distinct subclones, it is important to understand which subclones are more susceptible to which treatment and whether some subclones are unlikely to respond to planned treatment. Critically, the relapse driving clone and associated LSCs need to be targeted and this requires subclonal functional analyses. Our study showcases that single-cell multi-omics data can be leveraged and performed at the time of diagnosis to assess this aspect of precision medicine. Future larger screenings with low-cell-number input approaches will hopefully help uncover novel combination options that may be translated towards clinical applications.

7. The data on CK349 in Figure 5 and 6 could be combined into one figure as an example of how analysis of individual subclones yields clinically relevant findings.

Our response: Thank you for the constructive comment. With the addition of more patient samples, we have substantially revised and reorganized the entire manuscript. As suggested by the reviewer, we have made a completely new Figure 6 to demonstrate how analysis of individual subclones yields clinically relevant findings. Instead of combining the original Figure 5 and 6, in the new Figure 6, we dissected the intra-patient heterogeneity of two AML patient samples using newly added paired samples from diagnosis and relapse/refractory disease. This allowed us to assess the longitudinal genetic, epigenetic and transcriptomic evolution and further link this to *ex vivo* drug-response. We could show that in one patient, relapse was driven by the outgrowth of a subclone that had acquired a complex set of rearrangements at chromosome 6 that showed ongoing instability (new Figure 6b-c), as often seen in solid cancer metastasis and relapse (Burrell et al., 2013). Phenotypically, the relapse cells of this one patient were enriched in immature HSC-like cells, and thus displayed a shift from a more mature cell state observed at diagnosis to a more stem-like cell state at relapse (new Figure 6d). In the second patient, two subclones from diagnosis persisted at the refractory disease and showed subclone-specific resistance mechanisms. One subclone had gained a novel focal deletion at chromosome 17q arm, overlapping the tumor suppressor gene *NFI*, whereas the other subclone showed no genomic evolution but a shift in cell type abundance (new Figure 6j-k). We could further show that both subclones in the second patient were highly resistant to venetoclax therapy *ex vivo* already at diagnosis, reflecting the clinical response of the patient (new Figure 6p).

8. The Discussion is inadequate as many conclusions are stated as applicable to the broad class of CK-AML, but are derived from data from only 2, or even 1 sample, as is the case for the LSC, engraftment, and drug testing. Such studies with primary patient samples are difficult and expensive, but the

conclusions should be tempered based on these considerations. Similarly, the depth of genomic profiling and the number of cells analyzed is also very small, and care should be taken to acknowledge these limitations as well.

Our response: Guided by the comments of this reviewer we have doubled the analyzed CK-AML patients, extended the *ex vivo* drug screening and completely revised the entire manuscript including the discussion (see also our response above). We have additionally addressed the limitations of the study, including the number of cells analyzed and the inherent limitation of Strand-seq to only detect SVs larger than 200 kb in size. Altogether, we believe that the revised study showcases the biological and clinical relevance of single-cell multi-omics analysis in CK-AML.

Response to the comments made by Reviewer 2:

Reviewer #2 (expertise: haematopoietic malignancies malignancies, in vivo):

Remarks to the Author:

This manuscript presents a method for single multi-omic assessment of structural variants along with nucleosome occupancy and gene expression (merging existing scNOVA-seq from PMID 36424487 with CITE-seq) and applies it to a set of four AML patient samples with complex karyotype AML.

The manuscript presents detailed single cell resolution of structural variants in both AML patient samples and in PDX's derived from AML patient samples. While the genomic studies provide novel observations about clonality of structural variants and chromosomal instability in AML, **the data in Figures 5-6 which attempt to draw conclusions about relapse and therapy are underdeveloped.** These issues are as follows:

-Figure 5 is based on a single patient sample and the relapse from this patient was not subject to any type of genomic analysis (other conventional karyotyping). It is therefore unclear that the conclusions about clones which drive relapse are solidly supported by the data. The analyses in Figure 5 would require **more patient samples** evaluated via sc-Nova at diagnosis and relapse at a minimum, **and with PDX models if the authors want to draw conclusions about PDX-engrafting LSCs driving "resistance and relapse" as claimed.**

Our response: We thank this reviewer for acknowledging the novel observations about clonality of structural variants and chromosomal instability in AML. As discussed in the beginning, we have now substantially revised the manuscript to address the concerns of the reviewer. We have increased our cohort to size to include 8 CK-AMLs at initial sampling, 5 primary AML-PDX pairs and 2 paired samples from diagnosis and relapse/refractory disease (new Figure 1a). The new Figure 6 shows the analysis of the two paired samples pre- and post-therapy for which we have performed scNOVA and CITE-seq analysis. This allowed us to assess the genetic, epigenetic and transcriptomic evolution in a therapy setting. We could show that in one patient, relapse was driven by the outgrowth of a subclone that had acquired a complex set of rearrangements at chromosome 6 that showed ongoing instability, as often seen in solid cancer metastasis and relapse (Burrell et al., 2013). Phenotypically, the relapse cells of this patient were enriched in immature HSC-like cells, and thus displayed a shift from a more mature cell state observed at diagnosis to a more stem-like cell state at relapse (new Figure 6d). In the second patient, two subclones from diagnosis persisted at the refractory disease and showed subclone-specific resistance mechanisms. One subclone had gained a novel focal deletion at chromosome 17q arm, overlapping the tumor suppressor gene *NFI*, whereas the other subclone showed no genomic evolution but a shift in its cell type abundance (new Figure 6j-k). We could further show that both subclones from the second patient were highly resistant to venetoclax-based therapy *ex vivo* already at diagnosis, reflecting the clinical response of the patient.

Regarding the link between engraftment and relapse, we unfortunately lacked the material to perform this analysis in more samples. Others have shown that engrafted cells are skewed towards relapsed AML clones in PDX models generated from diagnosis AML cells (Kawashima et al., 2022; Shlush et al., 2017). But to avoid broad generalization based on one sample, we have toned down the text, and indicated the potential of engrafting cells to also drive disease progression, as already shown by others. Lines 452-456: "[...] we identified different clonal evolution fates and patterns during CK-AML reconstitution in mice [...]. Our data further indicate that the PDX engraftment-driving subclone can also drive relapse outgrowth in CK-AML patients, as in the case of CK349."

Despite our inability to further investigate the connection between engraftment and relapse, we embarked on a completely new longitudinal scNOVA-CITE analysis for two patients, described above, utilizing pre- and post-treatment sample pairs (new Figure 6). This allowed us to exemplify the biological and clinical relevance of single-cell clonal evolution analysis beyond PDX models.

-The therapeutic studies in the manuscript are quite modest in their findings. It is established from pivotal clinical trials of venetoclax and hypomethylating agents that patients with complex karyotype AML are less responsive to the regimen of azacitidine and venetoclax than patients with more

favorable/intermediate risk cytogenetics (see PMID 32786187 amongst other publications). The lack of favorable outcomes of patients with CK AML to this regimen is the basis for much ongoing clinical trials and preclinical studies in AML. Thus, the single patient *ex vivo* observation in Figure 6 is clearly not representative of responses to this regimen in actual patients. This questions the reliability of the claims made and/or the assay used in Figure 6.

Our response:

We appreciate the reviewer for bringing this to our attention and apologize for any shortcomings in the wording of our finding. As also outlined above and in the beginning, we have now included markedly more patients and revised the entire manuscript accordingly. The revised and extended manuscript further strengthened our initial conclusions and added intriguing new insights into CK-AML genomics and function.

More specifically, as originally stated in the manuscript, the combination treatment of venetoclax and azacitidine was the most effective in CK349-SC3 compared to the other subclones in the same sample (see Figure below), but our suggestion for its potential efficacy *in vivo* was too bold. For the revised manuscript we have included two additional patients (HIAML47 and P9) that were treated with venetoclax and azacitidine as first-line treatment. HIAML47 showed only a short partial clinical response and P9 was refractory to the treatment. Having samples from these two patients made it possible to compare the *ex vivo* venetoclax sensitivity of CK349-SC3 with sensitivities from two patients with known clinical response to venetoclax and azacytidine. As shown in the figure below, CK349-SC3 showed only slightly better efficacy to venetoclax monotherapy *ex vivo* compared to the cells from these two patient samples. The sensitivity to the combination treatment of azacitidine and venetoclax was largely driven by the effect of azacytidine with CK349-SC3 and HIAML47 leukemic cells (SC2 and SC3) showing similar killing effects at lower venetoclax concentrations. While the combination treatment of venetoclax and azacitidine was the most effective in CK349-SC3 compared to the other CK349 subclones, the killing efficacy was not better than in HIAML47. Thus, we have revised the text and removed the suggestion that the CK349-SC3 clone would likely be sensitive to venetoclax *in vivo*. To find alternative treatment options, we tested two additional inhibitors, revumenib and elesclomol, in the PDX cells from CK349. Revumenib is a Menin inhibitor that we hypothesized could show efficacy as CK349-SC3 had SVs affecting 11q and overlapping *KMT2A*. Elesclomol induces cellular death presumably through an increase in cellular oxidation, and has shown increased efficacy *in vitro* in TP53-rearranged AMLs (Nechiporuk et al., 2019). As shown in the figure below, PDX cells from CK349 were resistant to all other tested drugs except for elesclomol. While elesclomol was relatively toxic also in healthy HSPCs (see Figure below), its effectiveness against this otherwise pan-resistant CK-AML may justify further preclinical and clinical investigation. These data also encourage larger screenings to find alternative treatment options for this patient with even better efficacy than of elesclomol.

a

b

c

Rebuttal Figure R2.1. a) Viabilities (fraction of viable cells compared to untreated control) of different populations in CK349, HIAML47 and P9D after 24h *ex vivo* exposure with indicated concentrations of venetoclax (top) or azacytidine and venetoclax (bottom). All samples were largely resistant to venetoclax monotherapy whereas CK349 and HIAML47 showed moderate response to the combination therapy, likely due to azacytidine sensitivity. b) Viabilities (fraction of viable cells compared to untreated control) of CK349 PDX cells after 24h *ex vivo* exposure indicated treatments. c) Viabilities (fraction of viable cells compared to untreated control) of healthy BM HSPCs (CD34+ cells) after 24h *ex vivo* exposure with elesclomol.

-Related to the above, Bcl-xL (the other therapeutic highlighted in Figure 6) is a well-known therapeutic target in AML and the same Bcl-xL inhibitor as shown here has been recently suggested to overcome p53 mutant AML which lacks response to azacytidine and venetoclax (PMID 36508699).

Our response: We agree with the reviewer that Bcl-xL inhibition is not a novel therapeutic target and are happy to include the citation of the mentioned manuscript in the revised version: Kuusanmäki et al. 2023, Blood.

Response to the comments made by Reviewer 3:

Reviewer #3 (expertise: single-cell methods, AML):

Remarks to the Author:

In their manuscript “Identification of targetable relapse-driving leukemic stem cells in complex karyotype AML”, Leppä et al. set out to characterize intra-tumor heterogeneity in four cases of complex karyotype acute myeloid leukemia (CK-AML). Specifically, the authors generate Strand-seq data for a total of 278 cells from the primary patient samples, which is analyzed using the scNOVA computational approach to reveal somatic structural variation (SV) >200kb and nucleosome occupancy profiles with single-cell resolution. The authors reveal the existence of multiple subclones with distinct karyotypes in two of the four CK-AML cases, which they suggest are shaped by (linear and circular) breakage fusion bridge cycles. In addition, both chromothripsis-(like) patterns and ongoing karyotype remodeling of some cells is observed. On the most genetically heterogeneous cases, CK282 and CK349, CITE-seq was performed, providing additional high-throughput transcriptome and targeted cell surface proteome data for ~14k cells. The authors propose the scNOVA-CITE workflow for integration of these datasets, enabling them to couple (dynamic) genomic, transcriptomic and functional information at the single-cell level. Based on analysis of the integrated data, the authors propose that genomically unstable subclones carry a leukemic stem cell (LSC)-associated stemness signature. Strand-seq & scNOVA analysis of patient-derived xenograft (PDX) models indicates superior engraftment of the identified LSC-like subclone in these two cases, which is supported by karyotyping in a CK349 relapse sample. Finally, the authors perform drug sensitivity profiling using combinations of standard chemotherapy regimens and BH3 mimetics to target these relapse-driving genomic instability and stemness-associated phenotypes, highlighting a potential therapeutic options to eradicate relapse-driving LSC populations.

The manuscript is clear and well-structured but could be **more succinct (e.g., description of the subclones, SVs and potential mutational mechanisms in CK282 and CK349, lines 229–278). The Results contain several sections which are more hypothetical and would better benefit a Discussion, which is currently limited.**

Our response: We thank this reviewer for these positive and encouraging comments. We have substantially revised the manuscript as discussed in more detail already above and also below, including addition of more patient samples, a shortened description of the subclones, and significantly extended the discussion.

Overall, the main novelty presented lies in the combined scNOVA-CITE workflow and its integration of the different modalities generated (structural genomic, open chromatin, transcriptome and targeted surface proteome) on patient material and clinically relevant models. Specifically, the application to primary and PDX CK-AML samples provides a unique and intriguing insight into the evolution and intratumor genetic and phenotypic heterogeneity of the disease. The work may hence provide a blueprint for future (larger) studies into the interplay between genome and phenotype during tumor evolution.

Our response: We express gratitude to this reviewer for recognizing the novelty of our study and its provision of unique, intriguing insights into the evolution and intra-tumor genetic and phenotypic heterogeneity of CK-AML. The acknowledgment of its potential as a blueprint for future studies is greatly appreciated.

That said, the **number of samples is limited** (4 patients initially, of which only 2 are actually explored using scNOVA-CITE) and the core of the **integration pipeline** (linking subclonal populations identified using Strand-seq and scNOVA to cell populations detected in CITE-seq), which underpins all downstream analyses, **lacks assessment or benchmarking**. The limited number of cells for engrafting subclones, small effect size for the LSC signatures, and potential limitations of the SV burden and karyotype heterogeneity metric further limit the generalizability of claimed links between genomic instability, stemness, engraftment, and therapy resistance/relapse.

Our response: We thank the reviewer for this constructive feedback. As stated above and in the beginning, we have substantially revised the manuscript and increased the cohort size to a total of 8 CK-AML patients at initial sampling, 5 paired primary AML-PDX pairs and two paired diagnosis-relapse/refractory pairs (new Figure 1a). Overall, now 855 single cells analyzed with Strand-seq and 41,524 single cells analyzed with CITE-seq are included in our revised manuscript. As summarized in new Figure 1a, we have performed scNOVA and CITE-seq analysis on all primary patient samples and scNOVA on all PDX samples. We trust that these reviewer guided extensions have adequately addressed the issue of limited sample number. The specific concerns regarding the downstream analyses are each addressed below.

Specifically, I have the following main concerns:

1. Integration of the Strand- and CITE-seq data relies on the scNOVA-CITE workflow and accurate genotyping of subclone-specific copy number changes in the CITE-seq gene expression data. Copy number inference from scRNA-seq is noisy however, even when breakpoints are used as a prior, and **especially when going to a lower threshold of 10 expressed genes as is the case here**. In addition, methods for copy number calling from Strand-seq data have not been published (a manuscript by Christiansen et al. is said to be in preparation). While broad classes such as “gain” or “loss” may be easily picked up, specific copy number states will be harder to designate. Furthermore, while CITE-seq is performed directly on flow sorted cells from the thawed cryopreserved samples, the Strand-seq data derives from cultured patient material. As such, correspondence between the two modalities may not be guaranteed, as the passaging may induce biases. Considering the above and in the absence of an in-depth assessment or benchmark (for instance, **using a ground truth single-cell genome + transcriptome dataset**), it is hard for this reviewer to assess the accuracy of the integration workflow and its downstream results.

Our response: Thank you for giving us the opportunity to fully address the technical concerns raised. Regarding the benchmarking of integration workflow, in Jeong et al. (Nature Biotechnology, 2022) we previously performed extensive benchmarking of the CONICSmat computational method in terms of its performance for the integration of scNOVA data with scRNA-seq data. In Jeong et al. (Nature Biotechnology, 2022), we also compared the performance of CONICSmat to other established tools for somatic copy number alteration discovery from scRNA-seq data, including inferCNV and HoneyBADGER. During the revision of the current manuscript, we have also compared targeted copy number re-calling using CONICSmat with copy number discovery using inferCNV which resulted in very similar SV detection as illustrated in the figure below.

Based on our observations, we do agree that CONICSmat – similar to other CNV calling methods from scRNA-seq data – struggles to distinguish specific copy number states. For our analyses it was, however, sufficient to call whether a region was deleted or amplified without knowing the exact copy number status. Most amplifications in our cohort resulted in three copies. In the few cases where more copies were present (as in the case of the seismic amplification), other amplifications with fewer copies affecting the same segment were not detected in other subclones. The same was true for deletions, which were typically hemizygous resulting in a copy-number state of 1. Thus, for this study a more detailed copy number state assessment from CITE-seq data – thanks to the prior evaluation of the respective regions at high resolution with scNOVA – was unnecessary. We anticipate that in the future when CNV calling methods from scRNA-seq further improve, it will become possible to distinguish specific copy number states also for this modality.

We also concur that although Strand-seq excels at identifying SVs in single cells on a scalable basis, a remaining limitation of this technique is the requirement for cells to undergo one round of division in culture. We note in this regard that classic karyotyping of metaphases also relies on dividing cells and is routinely used in the clinic for risk classification and is part of the ELN-2022 recommendations for diagnosis and management of AML. That being said, we have compared the subclone percentages between Strand-seq and CITE-seq which we found to be well-correlated (Spearman’s $R=0.7$, p -value =

0.0003; see new Extended Data Fig. 8b), suggesting that (short term) cell culturing for one cell division does not induce significant biases. We have added a sentence about this to the manuscript. Lines 327-330: “We observed a marked correlation between Strand-seq and CITE-seq subclone detection (Spearman’s $R=0.7$, p -value = 0.0003, Extended Data Fig. 8b), suggesting that both single cell techniques give a similar representation of subclonal frequencies.”

Rebuttal Figure R3.1. Inference of somatic copy number alterations (SCNAs) using CITE-seq data from CK349 and P9D. a) InferCNV analysis of 3,514 high quality CK349 AML cells, and 267 control cells (cells sequenced by CITE-seq not originating from the myeloid lineage), profiled by CITE-seq. No additional SVs were detected on top of the SVs discovered by Strand-seq. b) CONICSmat-based targeted SCNA recalling of the subclone-defining SVs in CK349 using the high-resolution breakpoints derived from Strand-seq. Use of these SV breakpoints allowed CONICSmat to confidently call SC1 in 2,556, SC2 in 163 and SC3 in 270 single-cells from the CITE-seq data. c) InferCNV analysis of 4,575 high quality P9D AML cells, and 673 control cells (cells sequenced by CITE-seq not originating from the myeloid lineage), profiled by CITE-seq. No additional SVs were detected on top of the SVs discovered by Strand-seq. d) CONICSmat-based targeted SCNA recalling of the subclone-defining SVs in P9D using the high-resolution breakpoints derived from Strand-seq. Use of these SV breakpoints allowed CONICSmat to confidently call SC1 in 681, SC2 in 183 and SC3 in 3,133 single-cells from the CITE-seq data. Unassigned cells in CK349 and P9D fell below the posterior probability cutoff of 0.8 used for a confident SCNA assignment. (Note that the high variability observed on the 6p-arm in both InferCNV analyses, likely arose from the presence of MHC genes in this locus, as discussed at the InferCNV github vignette).

2. The authors use SV numbers and their variability across cells to assess karyotype heterogeneity. There are several caveats to this: (1) SVs <200kb and potentially also larger simple copy number gains are missed by scNOVA. **The latter is most evident within amplified regions as in Figures 2a or 5c, where distinct coverage peaks support the existence of multiple missed segments within the larger regions.** (2) SV detection and so these metrics are likely affected by sequencing depth per cell. This confounder is not explored. (3) By modifying expression of potentially hundreds or thousands of genes in-cis, SVs that alter copy number are likely to have more drastic effects on cell phenotypes and hence functionally relevant karyotype heterogeneity than balanced rearrangements such as inversions or translocations which may affect just a handful of genes near their breakpoints. Nevertheless, these classes are counted equally, potentially limiting the utility and comparability of the metrics. **Did the authors consider other metrics for functionally relevant karyotype heterogeneity?**

Our response: We thank the reviewer for these constructive comments. We agree that Strand-seq has remaining limitations with respect to its SV detection resolution, which we now have acknowledged in the discussion:

Lines 581-584: “[...] plate-based single-cell methods have an inherent limitation when it comes to cell numbers (Griffiths 2018) [...] Thus, the extent of genetic heterogeneity reported in this study could be underestimate as small subclones may have been missed [...]”

Regarding SV detection and the potential link to sequencing depth, we only included good quality cells with adequate coverage in our analysis. And while we acknowledge the 200 kb detection resolution of Strand-seq, as is evident from the figure below, the sequencing depth does not significantly affect the number of detected SVs per cell. We also agree with the Reviewer that SVs that alter the copy number are often more likely to have drastic effects on cell phenotypes, with the seismic amplification being a good example. However, as shown in Figure 1f “simple” inversions, too, can promote the overexpression of oncogenic drivers, potentially resulting in extensive downstream impacts on gene regulation within the genome. Consequently, we faced ambiguity regarding the prioritization of certain SVs over others, and we therefore decided to adopt the total count of SV-altered segments as a practical operational metric. We note that with a larger cohort containing more diagnosis as well as paired relapse/refractory samples with clinical information about treatment response, disease-free survival and overall survival, it will likely be possible to further improve this metric and assess which SVs are clinically relevant and should be given more weight. Likewise, a larger cohort will also allow the exploration of other metrics for functionally relevant karyotype heterogeneity.

Rebuttal Figure R3.2. Strand-seq coverage versus number of SV-altered segments detected. Higher SV numbers are not caused by higher sequencing depth per cell.

3. While this reviewer appreciates the amount of work performed, **sample sizes** ($n=4$ for Strand-seq, but only 2 for scNOVA-CITE) **and cell counts do not always support the general claims** made in the manuscript. For instance, based on perceived differences in cell type fractions assigned using correlation in nucleosome occupancy patterns for subclones represented by only 5 and 4 cells (Fig 4a) and small differences in expression signatures (Fig 4b), authors conclude on lines 362–364: “These data suggest an association between the genetic profile of a cell and its ability to generate more differentiated leukemic cells, arguing that complex SVs may drive stem-like features.” The authors also **assume direction of causality here** (as with their example of ALDH1A1), **while it may be equally likely that stemness allows cells to survive accumulation of more complex SVs**. These claims are then **repeated as proven further in the text: e.g., lines 379–380**: “... high intra-patient karyotypic heterogeneity mediated through seismic amplification and the resulting high stemness ...”. The **discussion contains similarly general statements**, e.g., line 521: “In CK-AML, minor subclones with complex SVs and continued karyotypic instability generate such disease-driving and therapy resistant LSCs.”

Our response: We appreciate the reviewer’s comments. As mentioned above, we increased our cohort size substantially, with our new cohort now containing 6 patients with linear or branched polyclonal growth (each bearing multiple subclones) at initial sampling and 2 patients with monoclonal growth (one major clone each). For the 6 patients with multiple subclones we pursued the scNOVA-CITE framework to assess differences between subclones, expanding the number of assessed single cells to 35,577. As shown in the new Figure 3, in 5 of 6 of the patient samples with multiple subclones, we were able to confidently assign cells from the CITE-seq data to the corresponding subclones defined by scNOVA. Additionally, as shown in the new Figure 6, we included a relapse and a refractory sample for two newly added patients and assessed the longitudinal disease evolution using scNOVA-CITE.

We agree that our cell counts in the Strand-seq data are low and have acknowledged this as a limitation in the discussion. Nevertheless, in Grimes et al. (Nature Genetics, 2024, in press) we have shown that cell type enrichment is abundant in conjunction with SV mosaicism seen in normal CD34+ cells. In that study we detected cell type enrichments from as low as 5 cells, and verified these with orthogonal methods.

We included a comparison between the abundance of primitive cell types (HSC-like and CMPs) estimated with scNOVA and the LSC-associated scores based on CITE-seq, which appear well correlated (Spearman's $R = 0.71$, $p = 0.058$; new Extended Data Fig. 10b). Additionally, we identified a significant correlation between subclone-specific LSC-associated scores and subclone-specific SV burden ($R = 0.77$, $p = 0.00018$; new Extended Data Fig. 10c). But as the reviewer rightfully points out, at this point we cannot say whether stemness is caused by the underlying genetics or whether stemness allows cells to survive high SV burdens. We have revised the text accordingly to clarify this point in our general claims, including causality of SV acquisition and stemness.

Lines 399-401: "While high leukemic stem cell (LSC) scores generally showed association with primitive cell types and high SV burden (Extended Data Fig. 10a-c), they were not restricted to one cell type and showed extensive heterogeneity within subclones (Extended Data Fig. 10d)."

Minor comments:

4. Authors could perform CITE-seq or explore further cells with Strand-seq from the remaining two cases as they do appear to show some genetic heterogeneity, which may have become more evident with increased cell numbers. This might also help support or dismiss some of the claims highlighted above.

Our response: In the extended and completely revised manuscript we have included CITE-seq analysis for all primary samples including 8 patients at initial sampling as well as 2 matched relapse/refractory samples. From the 8 patients at initial sampling, 2 showed monoclonal growth and thus subclonal differences could not be assessed. We did include these samples in analyses where subclones were not needed, such as overall stemness assessment (new Extended Data Fig. 10a). The remaining 6 patients showed linear growth or polyclonal branched growth patterns at initial sampling (new Figure 2) and in 5 patients we were able to confidently assign cells from the CITE-seq data to the corresponding subclones defined by scNOVA (new Figure 3a). For the one sample for which we were unable re-call the subclones from CITE-seq data using CONICSmatrix, the subclone-defining SV (Dup 8q) passed the initial filtering of uninformative, noisy regions (BIC difference > 200 and likelihood ratio test adj. p -val < 0.01) but it did not reach our posterior probability cutoff criteria of 0.8, making it not possible to confidently call or reject the presence of the SV using CITE-seq. The limited resolution of CITE-seq based copy-number calling in this case likely stems from the fact that the subclone-specific duplication SVs in the sample do not lead to marked gene expression changes. Consequently, confidently re-calling the subclones was not feasible from CITE-seq data based on our set cutoff criteria.

5. A more **fine-grained and algorithmic phylogenetic reconstruction** for the CK-AML cases would improve the impact and robustness of the findings and proposed methodology. The authors could explore using their copy number estimates/karyotypes and recently published tools such as SCARLET, Medalt, MEDICC2, among others.

Our response: We agree with the reviewer that a more fine-grained and algorithmic phylogenetic reconstruction would be advantageous. We have used hierarchical clustering based on Ward's method to arrange SVs, which works well to reveal the subclones (revised Figure 1b and new Extended Data Fig. 5c), but is not ideal for phylogeny interpretation. Regarding the published tools mentioned by the Reviewer, SCARLET is based on SNVs from scDNA-seq data which we are lacking from Strand-seq data, MEDALT detects focal (gene resolution) and broad (chromosomal-arm resolution) CNAs but not more fine-grained SVs that are prevalent in our Strand-seq data, and lastly, while MEDICC2 is able to detect also more fine-grained SVs, it only considers genomic alterations that change copy number, thus missing SVs such as inversions and translocations present in our Strand-seq data. Nevertheless, we tried out MEDICC2 on our samples using just CNAs as input. When using 8 cores for parallelization, MEDICC2 was not able finish the run within the 100h run time limit that we set. Long runtimes are a known issue based on the MEDICC2 bitbucket and a suggested workaround for this is to run MEDICC2 with multiple cores. As we were already using multiple cores the next suggestion is to remove duplicate cells. When we limited the input to up to 5 cells per subclone and increased the parallelization to 12

cores, we ran into memory limitation issues and MEDICC2 was not able finish the run within the 300 GB run memory limit that we set (see figure below for the job report). As requesting more memory blocks the cluster for other users, jobs with > 200 GB are not recommended by our Omics IT and Data Management Core Facility. Thus, we anxiously wait for better methods tailored for Strand-seq data to perform algorithmic phylogenetic reconstruction.

Rebuttal Figure R3.3. MEDICC2 job results. a) Job report of the run for patient sample CK282. b) CPU (top) and memory (bottom) consumption of the job.

6. It is unclear whether any non-leukemic diploid cells were captured in the Strand-seq and CITE-seq analyses, if/how they were identified and excluded from the analyses (e.g., to avoid biasing karyotype heterogeneity measures or assignments to clones).

Our response: In the initial cohort all captured cells had many SVs making it extremely unlikely that they were normal cells. However, with the addition of the new samples we did capture one good quality cell in one sample without any detectable SVs (HIAML47 in new Figure 2c). As the patient progressed from MPN/CMML to AML, we hypothesized that this cell was reminiscent of pre-LSCs rather than healthy HSPCs. In the CITE-seq data we only assigned cells to subclones if we could confidently call subclone-defining SVs. Thus, healthy diploid cells were excluded, with the exception of HIAML47 where we had evidence that pre-LSCs were present. In the CITE-seq data we identified these pre-LSCs by the absence of clonal trisomy 8 and subclonal Del at 12p (new Figure 2c). As healthy monocytes can also be present and lack SVs, we assigned each cell to their corresponding healthy counterparts using automatic cell type annotation with SingleR, and only considered cells resembling HSPCs as pre-LSCs (see Methods).

7. Line 38: suggest modifying “characterized by highest level” to ““characterized by the highest level”

Our response: We have revised the abstract and removed this sentence.

8. Quality metrics are only provided for the Strand-seq data on the original 278 cells derived from the primary patient samples. I could not find those for the PDX-derived Strand-seq data, nor for the CITE-Seq experiments. (Partially) deidentified data supporting the analyses in the form of gene or binned-read counts were not included or referred to within the manuscript and EGA identifiers are not included.

Our response: We apologize for this and have now included the quality metrics for all Strand-seq and CITE-seq data (new Supplementary Table 2, new Extended Data Fig. 1a-c). The EGA accession number has now been included.

9. A number of Extended Data Figure panels are of low quality and hard to read. Specifically, Extended Data Figures 2a-c would be improved if vectorized, 7 has the scale bars mirrored and pasted in, and in 10b the right M-FISH panel is of low quality.

Our response: We have improved the quality of the figures when possible.

10. Extended Data Figure 2d – it is unclear what the brown and blue sequences indicate.

Our response: The brownish red and blue sequences indicate the excerpt sequence around the breakpoint. We have added this to the figure legend.

11. Figure panels 2e and 2h are unnecessarily repeated in Fig 4a. An indication of the number of cells in each subclone ($n = \dots$) would be more helpful to the reader to assess effects.

Our response: We have extensively revised our figures, and this includes removing the repeated figure panels from Figure 4a.

12. Lines 136–139: I would not describe the H1-specific reduced NO (and potentially H1-specific expression) as being “driven by epigenetic dysregulation”, as this is likely caused by the rearrangement and enhancer hijacking. Suggest rephrasing.

Our response: We have rephrased the sentence to “[..]. suggesting that epigenetic dysregulation *reflects* aberrant haplotype-specific gene expression of the *MECOM* oncogene [...]”

13. It is unclear whether whole-chromosome gains are counted towards the “SV burden”.

Our response: Yes, whole-chromosome gains are counted towards the SV burden.

14. The authors refer to the identified complex amplification in CK349 SC3 as a seismic amplification and refer to Rosswog et al., Nat Genet, 2021. Is this not the same mechanism proposed earlier in Garsed et al, Cancer Cell, 2014? Especially since the authors do indeed observe large ring chromosomes and linearized versions in their PDX M-FISH analysis.

Our response: Yes, we believe it is the same mechanism as proposed by Garsed et al. with Rosswog et al. being the first to refer to this type of an amplification as seismic amplification. Rosswog et al. also mention that this is likely the same mechanism as reported by Garsed et al. We have added Garsed et al. as a reference.

15. Lines 195–199: This part is largely hypothetical and may be shortened or moved to the discussion.

Our response: We have shortened this part.

16. Lines 202–206: Do the circular and linearized chr11 derivatives occasionally co-occur or are they always present in distinct cells?

Our response: We do see that they can also co-occur in some cells. Below an example cell with both circular and linearized chr11 derivatives which we have also included in new Extended Data Fig. 11a.

PDX-CK349
Chromosome 11

Rebuttal Figure R.3.4. Multiplex fluorescence in situ hybridization (M-FISH) of a cell with normal chromosome 11, a ring chromosome of chromosome 11 and a linearized marker chromosome containing segments from chromosome 11 and chromosome 21 obtained from the PDX of CK349.

17. Line 234: What is the evidence supporting the statement that these SVs are “serially acquired”?

Our response: The event in question is depicted in the new Extended Data Fig. 5a and Extended Data Fig. 7d. The complex rearrangement consists of duplications and deletions spanning through the chromosome with only three short disomic segments present, as mapped out below in a simplified manner. The event distribution suggests that some deletions likely came before the duplication and others after. This is especially evident in the new deletion found in the second cell. Further, the switch from $cc > ww$ at ~ 47 Mb indicates that there is an inversion. Importantly the switch happens to both copies of the duplication (i.e $cc > ww$ - if we only consider the ‘mutated’ H2 strand). This suggests that it more likely arose after the bigger duplication. A classical inverted duplication would go, e.g. from $w > wc$, indicating a single duplication event arose in the inverted orientation from the original copy. Thus, a $cc > ww$ signal suggests the duplication arose and then a segment was inverted afterwards.

Rebuttal Figure R.3.5. Strand-specific read depth of two representative single cells from CK349 showing serially acquired SVs at chromosome 13. Reads denoting somatic SVs, discovered using scTRIP, mapped to the W (Watson, orange) or C (Crick, green) strand. A schematic depiction of the strand-specific SVs is shown on top of the Strand-seq plots.

18. Can the authors expand their discussion of the remarkable ongoing instability observed for specific chromosomes in several subclones? E.g., the frequency of distinct chr13 and chr17 rearrangements in CK349 and of chr20 in CK282? What might be the driving forces?

Our response: We have included new plots to better describe the frequency of distinct chr13 and chr17 rearrangements in CK349 (new Extended Data Fig 5b and 6d, respectively). A plot for chr20 SVs in CK282 could already be found Fig. 2j in the initial manuscript which we have now moved to new

Extended Data Fig. 6b. As for the driving forces, unfortunately we do not know what causes this ongoing instability and we can only speculate. Both samples in question harbored a deletion at chromosome 17 overlapping *TP53* and CK349 additionally carried a mutation in *TP53* (Supplementary Table 1), likely enabling the propagation of chromosomal instability (Sansregret et al., 2018). In CK349 chr17 and in CK282 chr20 contained a terminal deletion with a potential loss of a telomere, which can result in sister chromatid fusion and prolonged BFB cycles. BFB cycles can terminate when the unstable chromosome acquires a new telomere, often by translocation of the ends of other chromosomes (Cosenza et al., 2022). Thus, the instability may be passed on from one chromosome to another and result in ongoing instability affecting multiple chromosomes. In our case, all the chromosomes in question were involved in (iso)dicentric chromosomes or translocations. However, as evident in new Extended Data Fig 5b and 6d, we also observe stabilization of some of the instability. We now cover this topic in the revised discussion.

19. Figure 1b and Extended Data Figure 6 are clustered by their SV genotype likelihoods. It is however unclear how these likelihoods are derived. A separate heatmap akin to these figures but showing the actual underlying likelihoods would also be helpful to assess confidence in the genotype assignments.

Our response: We apologize for the unclarity in the original manuscript, and have added more detail to the methods regarding the SV calling. As explained in the scTRIP paper by Sanders et al. (2019, Nature Biotechnology), SV discovery is performed using a joint calling framework that first aligns, normalizes and places reads into genomic bins to assign template strand states and builds chromosome-length haplotypes. It then infers SVs in the segmented data by employing a Bayesian model that estimates the genotype likelihoods for each segment and each cell. Using this Bayesian model, the most probable SV type is assigned to each segment. We then manually inspected and verified each SV. We have rephrased the heatmap legend and changed it from “SV genotype likelihoods” to “SV genotypes” due to the additional manual inspection step.

20. Lines 274–278: “... CK-AML is not only characterized by complex genomic rearrangements leading to extensive SV density ...” and “... this makes almost every leukemia cell unique ...”. Given the limited number of samples and cells explored, authors may wish to rephrase these general statements.

Our response: We have toned down our general statements, including removing this sentence.

21. How robust are the differential NO findings for small subclones, which were captured with just 4 or 5 cells (e.g., CK282, SC2/5)? For instance, for SC2, a disproportionate amount of differential NO signal localizes to (clusters on) chr8, which is subclonally aberrated in these cells. A similar pattern is visible for chr7 in SC5. While some of this may indeed be true signal driven by rearrangements (nice to show if the case), it is unclear to this reviewer how much missed rearrangements, (small) CN changes not normalized away, read binning, or breakend position approximation bleed into these results?

Our response: In the Figure S6d of our scNOVA manuscript (Jeong et al., 2023) (see panel a in figure below), downsampling analysis showed that even if we decrease the number of cells down to 5 cells or to a single-cell, scNOVA’s infer expressed gene module gives the AUC at around 0.8. Moreover, according to the Extended Data Figure 4 of scNOVA manuscript (see panel b in figure below), small cell fraction of subclone down to 1.3% still gives reasonable AUC (above 0.7 in case of CNN+PLS-DA) for the task of inferring differential gene expression. This benchmarking analysis shows the robustness of differential NO analysis for small subclones.

To further illustrate that that the NO signal in small subclones in our cohort is not driven by CN changes not normalized away, we assessed the copy number states for each cell in chromosomes affected by SVs. In the figure below, we show for chromosomes 8 and 7 the single cell copy number assignments inferred by PloidyassignR using 1Mb bins and 500kb sliding window (GitHub.com/lysfyg/PloidyAssignR). The genes showing differential NO in the subclones affected by rearrangements are shown on top of the heatmap. The cells affected by the rearrangements cluster

together and show a very clear pattern of copy number states through the chromosome. Thus, we believe that the DE genes from NO data represent actual NO changes.

Rebuttal Figure R.3.6. scNOVA performance evaluation in Jeong et al. 2022. a) AUC values for each chromosome, estimated in downsampled aggregated ('pseudo-bulk') Strand-seq data, as well as in single cells, using scNOVA's CNN. The overall AUC was computed as the weighted average over 23 chromosome pairs, scaled by the number of genes. b) In silico cell mixing of RPE-1 and HG01573 cells to simulate application of scNOVA to different cell fractions (CFs). In this analysis six different CF ranges were considered (20, 10, 5, 3.3, 2, and 1.3). For each in silico cell mixing experiment, a total of 150 single cells were randomly subsampled for the major pseudo-clone (containing RPE-1 cells) and the minor pseudo-clone (HG01573 cells), by controlling the minor pseudo-clone CF at 20, 10, 5, 3.3, 2, and 1.3%, respectively. For each CF, random subsampling of single-cell libraries 10 times was performed, with the respective mean AUC depicted in the plot. Two different analysis modes - default (dashed lines, CNN with negative binomial generalized linear model), and alternative (solid lines, CNN with PLS-DA) are depicted. When the CF is larger than 10%, the default mode performs better, whereas for CFs smaller than 10%, the alternative mode outperforms the default mode.

Genes showing differential NO in CK282 subclones and the corresponding single-cell copy number assignments inferred by PloidyassignR

Rebuttal Figure R.3.7. Single cell copy number assignments inferred by PloidyassignR. Using 1Mb bins and 500kb sliding window the heatmap shows the copy number status for each bin along chromosomes 8 (top) and 7 (bottom) in each CK282 cell, depicted as rows. The cells harboring rearrangements on the plotted chromosomes are annotated on the right with the corresponding subclone. The ideogram on top of the heatmap shows the genes with differential NO in the subclone affected by rearrangement, with decreased NO shown in blue and increased NO in red.

22. For the scNOVA-CITE integration, authors state on lines 311–313 that “integration of the data showed clustering of the cells mostly along genetic subclones (Fig. 3d-e)”. This reasoning is somewhat circular, since the RNA data itself was directly used to assign these subclones.

Our response: We agree with the reviewer that this statement may appear somewhat circular. However, many subclone-specific differentially expressed genes constitute genes not hit by subclone-defining SVs. Thus, the phenotype of the subclones is not only a result of increased/decreased number of gene copies.

23. To support the correspondence of culture-derived Strand-seq and direct CITE-seq, validity of the integration, and derived results, authors could explore the genome-wide (or for the identified genes) correlation between NO and gene expression changes. For instance, by plotting delta-NO vs log-fold change in gene expression for all genes per subclone.

Our response: We have shown in new Extended Data Fig. 8b that Strand-seq and CITE-seq subclone detection shows good correlation (Spearman's $R=0.7$, p -value = 0.0003), suggesting that both single cell techniques give a similar representation of subclonal frequencies. As for benchmarking, in our scNOVA manuscript (Jeong et al., 2023) we previously performed extensive benchmarking of the scNOVA approach and showed that NO at gene bodies shows an inverse correlation with bulk RNA-seq gene expression values (see below Figure S4 of our scNOVA manuscript (Jeong et al., 2023).

Importantly, we perform copy-number aware NO analysis to accurately assess epigenetic changes. This means that the NO reads have been normalized to the copy number state to assess changes driven by the occupancy and not by the number of copies of a specific gene. Copy number changes have a direct effect on the read densities that are not reflective of true changes in nucleosome occupancy and can therefore bias the differential NO results (Robinson et al., 2012). Thus, we considered the copy number effect as an 'unwanted' effect, and adjusted for it during the normalization step to gain an accurate view of the epigenetic changes. This also means that the NO changes are not necessarily reflected in the gene expression changes, and vice versa, as different subclones can have multiple chromosomes with differing copy number states. For example, in a scenario where a subclone has a copy number gain, gene expression in this region tends to be higher compared to a subclone without the gain. However, copy-number aware NO may not show reduced NO as the gain does not necessarily result in increased accessibility. This makes the comparison of NO and gene expression difficult in a setting where subclones have different copy number states.

Rebuttal Figure R.3.8. Inverse correlation between NO at the body of genes and bulk RNA-seq gene expression values in Jeong et al. 2022. Inverse correlation shown for three RPE-1 derived cell lines: a) the original RPE-1 cell line, b) BM510 and c) C7. NO was calculated for 101 bins spanning -2kb to +2kb of gene bodies as a read count per million mapped reads using ngsplot software¹⁴. For each of the bins, genome-wide correlation between NO and gene expression level from bulk RNA-seq data was measured using Spearman's rho. Inverse correlation between NO and gene expression was apparent along the entire gene body (see gray dots), with the most pronounced inverse correlation measured at the TSS.

24. Lines 328–330: is the observation of PI3K-Akt-mTOR activity and other pathways in multiple subclones not in disagreement with the NO-based results, where this was specific to SC2?

Our response: We do not believe that this observation disagrees with the NO-based results in general, as we do not expect NO-based results and gene expression results to show perfect agreement. As discussed above, we performed the NO analysis in a copy-number aware manner to assess actual

epigenetic changes. In contrast, gene expression results are affected by multiple factors, including other epigenetic changes (methylation, histone modifications etc.) and gene dosage. With this in mind, our data shows that PI3K-Akt-mTOR shows activity in SC2 based on NO. Our data also shows, that PI3K-Akt-mTOR activity based on gene expression is enriched in SC1 and SC2, which may not only be due to NO alone as other epigenetic changes (e.g. methylation, histone modifications) and non-epigenetic changes (e.g. gene dosage, mRNA stability, non-coding RNAs) can also affect gene expression.

25. Line 334: Cells of different subclones occasionally seem to occupy similar cell states, as the authors indicate. Could the authors shed light onto what these cell states might be, e.g., via marker gene expression or pathway analysis or considering Gavish et al., Nature, 2023?

Our response: The shared cell states largely reflect lineage differentiation. As shown below for patient sample P9D, the patient has two larger clusters that broadly represent primitive and differentiated cells. These clusters generally upregulate expression of genes associated with a more primitive state (e.g. *CD34*, *EGR1*, *PROM1*, *MEIS1*) or with a more differentiated state (e.g. *MPO*, *AZU1* and *LYZ*). A more fine-grained clustering reveals additional clusters again largely reflecting differentiation. For example, marker genes for the primitive clusters 5 and 8 include genes associated with stemness (e.g. *EGR1* and *BCAT1*) and erythroid differentiation (e.g. *CD38*, *GATA2*, *GATA1*), respectively. In contrast, marker genes for the more differentiated clusters 2 and 3 include genes associated with monocytic (e.g. *SI00A9*, *SI00A8*, *CD14*) and dendritic cells (e.g. *FCGR2B*, *CD1C*), respectively. While SC1 cells are enriched in cluster 1 some cells can also be found in cluster 8 together with SC3 cells. This is even more clear for the more differentiated cluster where SC1 and SC3 cells occupy similar cell states, largely reflecting monocytic and dendritic differentiation. We hope this sheds some light to the similar cell states between subclones.

Rebuttal Figure R3.9. Similar cell states between subclones in P9D. a) UMAP annotated based on the subclones called using CONICSmat. b) UMAP annotated based on the identified clusters. c) Expression of primitive markers *CD34* and *HOPX*. d) Expression of myeloid differentiation marker *LYZ*. e) Expression of cluster 5 (top) and cluster 8 (bottom) marker genes *EGR1* and *BCAT1*, and *CD38* and *GATA2*, respectively. f) Expression of cluster 2 (top) and cluster 3 (bottom) marker genes *S100A9* and *FCGR2B*, respectively.

26. Lines 266–271 and 391–392. It is unclear to this reviewer how chr20 in CK282 may show a classical and shared BFB pattern (with the DelTer and adjacent InvDup) where the initiating DelTers are of different sizes. It would also be helpful to include a figure akin to Extended Data Figure 5 showing the different chr20 patterns in CK282 and its PDX model.

Our response: With the ‘classical BFB pattern’ (discussed extensively in Sanders et al. Nature Biotechnology 2020 with respect to Strand-seq data) we refer to the presence of an inverted duplication adjacent to a terminal deletion at chromosome 20 which was present (and thus shared) in most SC1, SC2, and SC3 cells. As evident in new Extended Data Fig. 5a-b (previous Fig. 2i-j), the breakpoints for the terminal deletions are, however, not the same for all cells. We have now included in new Extended

Data Fig. 11d examples of cells from CK282 and PDX-CK282 showing different chr20 patterns. All the detected patterns are summarized in new Extended Data Fig. 5b and 11c.

27. Lines 410–411: “... only some CK-AML subclones display high engraftment potential ...”. The observation that CK282 SC2 and CK349 SC3 are the major subclones in the PDXs, does not mean the other subclones do not have high engraftment potential. They might engraft efficiently on their own. “...which is associated with their karyotypic instability and stemness activity ...”. As indicated above, the evidence supporting this here remains weak at best.

Our response: We agree with the reviewer that the other subclones may also have engraftment potential if transplanted on their own. However, when multiple subclones are present (as in our case), we only observe engraftment of LSCs from one or two subclones with one subclone showing superior engraftment (as assessed by the fraction of engrafted cells). However, with the addition of new samples, we also see engraftment of major and minor subclones and have focused on describing the insight gained by our approach into the subclonal repertoire of LSCs during CK-AML reconstitution in mice.

28. Lines 427–429: “The majority of these genes reside in the genomic segments affected by the DNA rearrangements at chromosome 11, and were likely upregulated due to higher CN levels in the seismic amplification.” It would be helpful if the location of these genes and the CN profile of this chromosome were visualized together.

Our response: We have substantially revised the manuscript and due to limited space removed the above part from the manuscript. However, for this reviewer, we have included below a plot showing the top 20 upregulated genes located at chromosome 11 together with the CN profile.

Rebuttal Figure R3.10. Upregulation of genes residing in the genomic segments affected by the DNA rearrangements at chromosome 11. The top 20 upregulated genes at chromosome 11 in CK349 engraftment-driving cells are shown at the top with three representative single cells from CK349 showing seismic amplifications shown below. Reads denoting somatic SVs, discovered using scTRIP, mapped to the W (Watson, orange) or C (Crick, green) strand.

29. Figures 6b and 6e are overplotted, making it hard to observe the distribution for each subclone.

Our response: In the revised Figure 5c-e, we have now only highlighted the engraftment-driving cells. We have included density plots next to the scatter plots for better visualization in new Extended Data Fig. 12c-e.

30. Lines 506–507: “This suggested a high dependency of these cells for BCL-xL, which is in agreement with their high sensitivity to the BCL-xL-specific inhibitor and their elimination by this treatment in culture.” Both the expression and drug sensitivity differences between the CK349 populations appear relatively minor.

Our response: We believe that the reviewer refers to CK282 and not CK349. All subpopulations in CK282 showed good response to the inhibition with SC2 simply showing the strongest response. We have rephrased the text to make this more clear. To better see the differences between the subclones, we have also included a line plot in addition to the heatmap (new Figure 5I).

31. Lines 518–524: “Our data highlights the clinical relevance of intra-patient karyotype heterogeneity with ongoing chromosomal instability and punctuated evolution-like rearrangements. Such clonal diversification at the individual cell level may allow the selection of cancer stem cell-driven subclones with increased fitness. In CK-AML, minor subclones with complex SVs and continued karyotypic instability generate such disease-driving and therapy resistant LSCs. Functionally, these cells show characteristics associated with aggressive disease, namely ability to regenerate leukemia in vivo”
a. As highlighted above, this reviewer remains unconvinced that the current sample and cell numbers as well as effect sizes support these general statements.

Our response: As discussed above, we have substantially revised the manuscript, including adding more samples and thus completely revised and extended the results and discussion sections. With the expanded cohort size, we reaffirm that our high-resolution data combining single-cell genomics with single-cell transcriptomics provide intriguing insights into the subclonal patterns in CK-AML and uncover an even higher intra-patient complexity than previously reported for CK-AML. Furthermore, our results provide, to our knowledge, the first insights into the clonal evolution fates and patterns during CK-AML reconstitution in mice and in patients after therapy, revealed by single-cell-based SV discovery. Importantly, with these expanded data we discovered continued genomic evolution in 4/5 mice and in 2/2 treated patients with paired samples, demonstrating that engraftment in mice and disease progression in patients is often accompanied by spontaneous generation of karyotype diversity. Together, our data strongly suggest that ongoing chromosomal instability in CK-AML fosters intra-tumor heterogeneity, giving rise to cells with distinct characteristics.

Our data also show that CK-AMLs not only have a high degree of intra-patient heterogeneity but they also show extensive heterogeneity between CK-AML patients. To assess this inter-patient heterogeneity, in-depth analysis of large cohorts with hundreds of samples need to be tested, for which we present now a conceptual framework.

32. Line 593: “SV segments”, suggest rephrasing.

Our response: Thank you for the suggestion. We have changed SV segments to SV-altered segments.

33. Lines 611–612: Suggest rephrasing to: “To infer differentially active genes for each subclone, the single-cells in certain subclones were compared to all other single-cells in the same tumor using an alternative mode of scNOVA based on partial least squares-discriminant analysis.”

Our response: Thank you for this suggestion. We have included this change.

34. In the Weighted Nearest Neighbor analysis of CITE-seq data, 18 dimensions were used for the protein modality. Given that there are just under 40 proteins measured, this number seems rather high compared to the RNA where 30 dimensions are used for the thousands of genes measured. How was this parameter determined?

Our response: To determine the significant dimension, we plotted the standard deviations of the principle components which showed flattening at the picked parameters (see figure below).

Rebuttal Figure R3.11. Standard deviation of principle components of RNA (left) and ADT (right) modalities. Dashed line indicates the used parameters.

35. For the “Reference-based” mapping of leukemic cells onto a multimodal BM atlas, the authors use Anchors (Seurat’s FindTransferAnchors function). This implies that both the CK-AML and the reference bone marrow data contain at least some cells of the same type, i.e., normal bone marrow cells. However, it is not clear what fraction of normal cells is still present in the CK-AML CITE-seq data, and so whether the anchor-based mapping is reliable.

Our response: We thank the reviewer for the comment. We agree that “reference-based” mapping between healthy bone marrow and AML is not perfect and that label transfer methods need to be further improved. Considering the similarities of differentiation hierarchies between AML and normal cells (van Galen et al., 2019) we believe that annotating cells based on healthy BM cells can, however, be used to get an idea of the differentiation state of AML cells based on shared transcriptomic profiles. For the revised manuscript we changed the “Reference-based” *mapping* to “Reference-based” *annotation*. We assigned each cell from the AML samples to their corresponding healthy counterparts using singleR. It determines the similarity of each single-cell to reference BM cells based on Spearman correlation (see Methods).

36. Could the authors use the high-coverage (300x) OGM data to refine breakpoint positions in the Strand-seq data, potentially even in the subclones? This may be especially helpful in SV-dense amplified regions which are currently undersegmented in the Strand-seq.

Our response: Unfortunately, we tried to perform OGM from the PDX of CK349 but were not able to get good quality high molecular density DNA to perform the mapping.

37. Lines 969–972/Figure 2: Parts of the abbreviation list are duplicated.

Our response: We have removed the duplicated abbreviations.

38. Line 1012: add “testing” to “followed by Benjamini-Hochberg multiple correction.”

Our response: We have added “testing”.

39. Extended Data Figure 9c-d: it would be helpful if the order of samples on the horizontal axes was consistent between panels.

Our response: We agree, and have included more samples to the figure and made the order of the samples on the horizontal axes consistent between the panels (new Figure 4a and new Extended Data Fig. 10a).

40. Extended Data Figure 10b: How representative are the karyotypes of these two cells? The right M-FISH panel appears to indicate a homozygous loss of both chr16 and chr5q, and is unlikely to be viable.

Our response: Thank you for pointing this out. We have removed the plot as the metaphase spreads seem to not have been complete due to the metaphase spread preparation, and replaced it with another example cell from a complete metaphase spread.

41. Extended Data Figure 14a: Is the indicated CD45dim/SSClow/SSCmid gating strategy for singlets correct (i.e., only side and no forward scatter was used)?

Our response: Yes, the gating strategy is correct. As shown in new Extended Data Fig. 11a (previously Extended Data Figure 14a), doublets were removed based on forward scatter (Height vs Area), followed by removal of dead cells and lineage positive cells. Finally, leukemic blasts were gated from these live lineage-negative singlets within the CD45dim/SSClow/SSCmid gate.

Rebuttal References:

- Bandyopadhyay, S., Duffy, M.P., Ahn, K.J., Sussman, J.H., Pang, M., Smith, D., Duncan, G., Zhang, I., Huang, J., Lin, Y., *et al.* (2024). Mapping the cellular biogeography of human bone marrow niches using single-cell transcriptomics and proteomic imaging. *Cell*.
- Bollen, Y., Stelloo, E., van Leenen, P., van den Bos, M., Ponsioen, B., Lu, B., van Roosmalen, M.J., Bolhaqueiro, A.C.F., Kimberley, C., Mossner, M., *et al.* (2021). Reconstructing single-cell karyotype alterations in colorectal cancer identifies punctuated and gradual diversification patterns. *Nat Genet* *53*, 1187-1195.
- Burrell, R.A., McGranahan, N., Bartek, J., and Swanton, C. (2013). The causes and consequences of genetic heterogeneity in cancer evolution. *Nature* *501*, 338-345.
- Cosenza, M.R., Rodriguez-Martin, B., and Korbel, J.O. (2022). Structural Variation in Cancer: Role, Prevalence, and Mechanisms. *Annual review of genomics and human genetics* *23*, 123-152.
- Griffiths, J.A., Scialdone, A., and Marioni, J.C. (2018). Using single-cell genomics to understand developmental processes and cell fate decisions. *Molecular systems biology* *14*, e8046.
- Jeong, H., Grimes, K., Rauwolf, K.K., Bruch, P.M., Rausch, T., Hasenfeld, P., Benito, E., Roider, T., Sabarinathan, R., Porubsky, D., *et al.* (2023). Functional analysis of structural variants in single cells using Strand-seq. *Nat Biotechnol* *41*, 832-844.
- Kawashima, N., Ishikawa, Y., Kim, J.H., Ushijima, Y., Akashi, A., Yamaguchi, Y., Hattori, H., Nakashima, M., Ikeno, S., Kihara, R., *et al.* (2022). Comparison of clonal architecture between primary and immunodeficient mouse-engrafted acute myeloid leukemia cells. *Nat Commun* *13*, 1624.
- Klco, J.M., Spencer, D.H., Miller, C.A., Griffith, M., Lamprecht, T.L., O'Laughlin, M., Fronick, C., Magrini, V., Demeter, R.T., Fulton, R.S., *et al.* (2014). Functional heterogeneity of genetically defined subclones in acute myeloid leukemia. *Cancer Cell* *25*, 379-392.
- Layer, R.M., Chiang, C., Quinlan, A.R., and Hall, I.M. (2014). LUMPY: a probabilistic framework for structural variant discovery. *Genome Biol* *15*, R84.
- Morita, K., Wang, F., Jahn, K., Hu, T., Tanaka, T., Sasaki, Y., Kuipers, J., Loghavi, S., Wang, S.A., Yan, Y., *et al.* (2020). Clonal evolution of acute myeloid leukemia revealed by high-throughput single-cell genomics. *Nat Commun* *11*, 5327.
- Nechiporuk, T., Kurtz, S.E., Nikolova, O., Liu, T., Jones, C.L., D'Alessandro, A., Culp-Hill, R., d'Almeida, A., Joshi, S.K., Rosenberg, M., *et al.* (2019). The TP53 Apoptotic Network Is a Primary Mediator of Resistance to BCL2 Inhibition in AML Cells. *Cancer Discov* *9*, 910-925.
- Priestley, P., Baber, J., Lolkema, M.P., Steeghs, N., de Bruijn, E., Shale, C., Duyvesteyn, K., Haidari, S., van Hoeck, A., Onstenk, W., *et al.* (2019). Pan-cancer whole-genome analyses of metastatic solid tumours. *Nature* *575*, 210-216.
- Robinson, M.D., Strbenac, D., Stirzaker, C., Statham, A.L., Song, J., Speed, T.P., and Clark, S.J. (2012). Copy-number-aware differential analysis of quantitative DNA sequencing data. *Genome Res* *22*, 2489-2496.
- Rücker, F.G., Schlenk, R.F., Bullinger, L., Kayser, S., Teleanu, V., Kett, H., Habdank, M., Kugler, C.M., Holzmann, K., Gaidzik, V.I., *et al.* (2012). TP53 alterations in acute myeloid leukemia with complex karyotype correlate with specific copy number alterations, monosomal karyotype, and dismal outcome. *Blood* *119*, 2114-2121.
- Sanders, A.D., Falconer, E., Hills, M., Spierings, D.C.J., and Lansdorp, P.M. (2017). Single-cell template strand sequencing by Strand-seq enables the characterization of individual homologs. *Nat Protoc* *12*, 1151-1176.
- Sanders, A.D., Meiers, S., Ghareghani, M., Porubsky, D., Jeong, H., van Vliet, M., Rausch, T., Richter-Pechanska, P., Kunz, J.B., Jenni, S., *et al.* (2020). Single-cell analysis of structural variations and complex rearrangements with tri-channel processing. *Nat Biotechnol* *38*, 343-354.

- Sansregret, L., Vanhaesebroeck, B., and Swanton, C. (2018). Determinants and clinical implications of chromosomal instability in cancer. *Nature reviews Clinical oncology* *15*, 139-150.
- Shlush, L.I., Mitchell, A., Heisler, L., Abelson, S., Ng, S.W.K., Trotman-Grant, A., Medeiros, J.J.F., Rao-Bhatia, A., Jaciw-Zurakowsky, I., Marke, R., *et al.* (2017). Tracing the origins of relapse in acute myeloid leukaemia to stem cells. *Nature* *547*, 104-108.
- Tyner, J.W., Tognon, C.E., Bottomly, D., Wilmot, B., Kurtz, S.E., Savage, S.L., Long, N., Schultz, A.R., Traer, E., Abel, M., *et al.* (2018). Functional genomic landscape of acute myeloid leukaemia. *Nature* *562*, 526-531.
- van Galen, P., Hovestadt, V., Wadsworth Ii, M.H., Hughes, T.K., Griffin, G.K., Battaglia, S., Verga, J.A., Stephansky, J., Pastika, T.J., Lombardi Story, J., *et al.* (2019). Single-Cell RNA-Seq Reveals AML Hierarchies Relevant to Disease Progression and Immunity. *Cell* *176*, 1265-1281 e1224.
- Waclawiczek, A., Leppä, A.M., Renders, S., Stumpf, K., Reyneri, C., Betz, B., Janssen, M., Shahswar, R., Donato, E., Karpova, D., *et al.* (2023). Combinatorial BCL2 Family Expression in Acute Myeloid Leukemia Stem Cells Predicts Clinical Response to Azacitidine/Venetoclax. *Cancer Discov* *13*, 1408-1427.
- Watkins, T.B.K., Lim, E.L., Petkovic, M., Elizalde, S., Birkbak, N.J., Wilson, G.A., Moore, D.A., Grönroos, E., Rowan, A., Dewhurst, S.M., *et al.* (2020). Pervasive chromosomal instability and karyotype order in tumour evolution. *Nature* *587*, 126-132.
- Zhang, Q., Riley-Gillis, B., Han, L., Jia, Y., Lodi, A., Zhang, H., Ganesan, S., Pan, R., Konoplev, S.N., Sweeney, S.R., *et al.* (2022). Activation of RAS/MAPK pathway confers MCL-1 mediated acquired resistance to BCL-2 inhibitor venetoclax in acute myeloid leukemia. *Signal transduction and targeted therapy* *7*, 51.

Point-by-point response to the comments made by the three reviewers to our manuscript by Leppä*, Grimes*, Jeong* et al. Korbels and Trumpp§ (NG-A62852R1):

We would like to thank all three reviewers for their positive comments.

Response to the comments made by Reviewer 1:

Reviewer #1:

Remarks to the Author:

I thank the authors for their thoughtful and comprehensive response to my concerns, which they have addressed to my satisfaction. I have a few minor comments:

Our response: We are glad that the reviewer's concerns were satisfactorily addressed.

- Should "278 single-cell genomes" be updated to 855 genomes as they report increasing their sample size?

Our response: It is unclear to us where the "278 single-cell genomes" is written as we could not find it in the main text or in the figure legends. On line #107 we state that "855 single-cell genomes" were sequenced. We also show in Figure 1a how the cells are distributed throughout the cohort (diagnosis/salvage samples, patient-derived xenografts and relapse/refractory samples) with the sum being 855 single cells for Strand-seq.

- Please include response duration and overall survival data in Table 1a.

Our response: We have included event-free survival and overall survival data in Supplementary Table 1a. For 7 patients for which data was available, we have also included a plot showing the percentage of peripheral blasts in response to given treatment to better visualize complete and partial responses (Supplementary Figure 1).

- If space permits, please incorporate the discussion provided in the rebuttal in the final manuscript, perhaps as supplemental information. Specifically, Rebuttal Figure R1.5, and comparison between single cell and bulk approaches (Comment 2).

Our response: We have extended the discussion about *ex vivo* drug screening and included a discussion of the differences between scTRIP and bulk genomic analyses as supplemental note.

Reviewer #2:

Remarks to the Author:

The authors have substantially revised the manuscript and notably increased the number of patient samples analyzed throughout. The manuscript is greatly improved.

Our response: We appreciate the recognition of the improvements and the additional efforts undertaken.

Reviewer #3:

Remarks to the Author:

In this revised manuscript, the authors have significantly improved the quality and impact of their work. This reviewer greatly appreciates the extra work done. In general, the comments raised by the reviewers have been addressed by the authors.

I do however still struggle to assess the robustness of the Strand- and CITE-seq integration workflow for assigning CITE-Seq cells to clones identified in the Strand-seq. In contrast to what the authors suggest, in Jeong et al., Nature Biotech 2022, this reviewer could only find benchmarking of the NO to gene expression models and comparison of RNA-based copy number inference tools, both of which do not address the issue of accurate clone mapping between the two methods. In fact, the relatively poor correlation in subclonal fractions between Strand- and CITE-Seq (new Extended Data Figure 8b, driven mostly by some very big and some very small subclones) and the inability of assigning cells in one of the polyclonal patients (CK295, despite large duplications on 8q and chr16 – I do appreciate the conservative approach here), demonstrates that this mapping is more problematic than apparent from the manuscript or even the literature (Jeong et al.). While performing extensive validation might be outside the scope of the current manuscript, more insights into the copy number calls or state probabilities (on both the Strand- and CITE-seq side) may already be helpful to demonstrate that this integration is in fact robust. A joint plot, combining Rebuttal Figure R3.1 (targeted CONICSmats CNA calling results) and Rebuttal Figure R.3.7 (PloidyAssignR CNA calls) where clones are aligned and indicated in both might already help, at least visually.

Our response: We thank the reviewer for the constructive feedback on our revised manuscript. We appreciate the recognition of the improvements and the additional efforts undertaken. Concerning the robustness of the Strand- and CITE-seq integration workflow for assigning CITE-seq cells to clones identified in Strand-seq, we would like to highlight that our recent publication, Grimes et al. Nat Genetics 2024 (Cell-type-specific consequences of mosaic structural variants in hematopoietic stem and progenitor cells), already addresses the benchmarking requested by this reviewer. Specifically, we validated our integration methodology by comparing single-cell RNA-seq based copy number inference with Strand-seq, assessing model performance for cell typing (Fig. 3b and Supplementary Fig. 8 from Grimes et al. 2024), and the targeted recalling of copy-number alterations for subclonal characterization (Fig. 5i-k from Grimes et al. 2024). These benchmarks, along with the work published in Jeong et al. Nat Biotechnol 2022, provide a strong foundation for this current manuscript on CK-AML.

To further demonstrate the robustness of our integration, we have added a joint plot of subclonal assignments from Strand-seq (based on scNOVA) and CITE-seq (based on CONICSmats) to the Extended Data Fig. 8c. This plot visualizes the strong correlation between CITE-seq and Strand-seq based assessments of subclonal heterogeneity in CK-AML, thus providing additional insights into the reliability and validity of our subclone assignments. For the reviewer, we have also included a plot below showing the CNA calling results from PloidyAssignR in addition to the scNOVA results included in Extended Data Fig. 8c. This plot shows that CNA inference from CITE-seq reflects well those from both PloidyAssignR and scNOVA.

Additionally, it is important to reiterate the primary motivation for integrating Strand-seq and CITE-seq/scRNA-seq in this current manuscript as well as in our prior works. Strand-seq offers superior resolution for detecting structural variants such as copy-number alterations as well as copy-neutral events in single cells. These events cannot always be fully captured by CITE-seq/scRNA-seq data alone due to its limited resolution (Table S4 from Jeong et al., Nat Biotechnol; Grimes et al., Nat Genetics 2024). We note that these difference in resolution are quite dramatic, over an order in magnitude (>10Mb in the case of CITE-seq/scRNA-seq in discovery mode, versus >200kb for Strand-seq). However, when copy-number alterations are identified by CITE-seq/scRNA-seq through targeted recalling using CONICSmats, the single-cell expression measurements provide more detailed insights into the transcriptomic state of the respective subclone than Strand-seq would provide alone. We have added a statement into the discussion about the motivation and gains for integrating Strand-seq and CITE-seq.

CK349

Rebuttal Figure 1. Subclonal heterogeneity between Strand-seq and CITE-seq-based assessment in patient CK349 (male, XY). Somatic copy number alteration (SCNA) discovery from Strand-seq data based on PloidyAssignR (left) and filtered scNOVA calls (middle) show good correlation with targeted SCNA calling based on CONICSmag from CITE-seq data (right). Subclone assignments are shown on the right of each heatmap with corresponding cell numbers. Centromeres for each chromosome are shown on top of the PloidyAssignR heatmap (left) for easier assessment of poorly mappable regions.

Point-by-point response to the comments made by the reviewers to our manuscript by Leppä*, Grimes*, Jeong* et al. Korbel§ and Trumpp§ (NG-A62852R2):

Reviewers' Comments:

Reviewer #3:

Remarks to the Author:

The Strand- and CITE-seq integration method used in the current paper relies on copy number inference from each modality separately, followed by clone inference and assignment (CNA-based approach). In contrast, the integration method in Grimes et al. relies on matching nucleosome occupancy patterns at marker genes (Strand-Seq) to their expression in scRNA (NO-based approach). As such, the benchmarks in Grimes et al. and Jeong et al. do not provide insights into the performance of the approach used here. Fig 5i-k from Grimes et al. only shows the results of CNA-based integration without a measure of reliability beyond mere similarity of clonal frequencies between Strand-Seq and scRNA. In fact, 17q-Del containing cells remain unresolved in the CNA-based approach.

That being said, this reviewer appreciates the conservative approach of not making assignments, e.g. in the 17-Del case above and CK295 in the current manuscript, where CNA-based matching seems to fail entirely. In addition, the co-occurrence analysis in Grimes et al., Supplementary Figure 26, does provide some degree of performance measure, even if only for a single case.

The new Extended Data Fig. 8c further also helps assuage this reviewer. It would have been nice had the authors included these side-by-side plots for all patients, as opposed to a single cherry-picked example. This would help readers gauge the robustness of assignments across cases profiled with Strand-seq plus CITE-/scRNA-seq (importantly also for the CK295 case where assignments fails) and add confidence the technique is generally applicable.

Finally, this reviewer fully agrees with the authors that integrating Strand-Seq with CITE-/scRNA-seq can be highly informative and is of general interest, as evident from the findings presented here and in previous work. Supporting its importance is also precisely why an assessment of integration performance, robustness, and pitfalls would have been helpful.

Our response: We appreciate the thorough evaluation of our manuscript and the insightful comments. We thank the reviewer for recognizing the challenge in resolving cells in particular with small SVs using our CNA-based integration approach. Our conservative approach, as the reviewer noted, was aimed at avoiding overinterpretation in cases where the data were ambiguous. We have highlighted this now also more clearly in the methods and further clarified our stringent filtering for confident SCNA re-calling.

We thank the reviewer for the constructive suggestion of side-by-side plots not only for one example, but for all patients. Following this suggestion, we have now included these as Supplementary Figure 2 (see also below). We are confident that these additional plots add confidence to the integration performance and its robustness.

Rebuttal Figure 1. Subclonal heterogeneity between Strand-seq and CITE-seq-based assessment. Somatic copy number alteration (SCNA) discovery from Strand-seq data based on filtered scNOVA calls (left) show good accordance with targeted SCNA calling based on CONICSmats from CITE-seq data (right). Subclone assignments are shown on the right of each heatmap with corresponding cell numbers. In patient CK295, cells could not be confidently assigned to all subclones based on posterior probability cutoff criteria of 0.8. See Extended Data Fig. 8c for patient CK349.